# Advancing regulatory variant effect prediction with AlphaGenome

Žiga Avsec[1,2 ✉], Natasha Latysheva[1,2], Jun Cheng[1,2], Guido Novati[1,2], Kyle R. Taylor[1,2], Tom Ward[1,2], Clare Bycroft[1,2], Lauren Nicolaisen[1,2], Eirini Arvaniti[1,2], Joshua Pan[1,2], Raina Thomas[1], Vincent Dutordoir[1], Matteo Perino[1], Soham De[1], Alexander Karollus[1], Adam Gayoso[1], Toby Sargeant[1], Anne Mottram[1], Lai Hong Wong[1], Pavol Drotár[1], Adam Kosiorek[1], Andrew Senior[1], Richard Tanburn[1], Taylor Applebaum[1], Souradeep Basu[1], Demis Hassabis[1] & Pushmeet Kohli[1 ✉]

Deep learning models that predict functional genomic measurements from DNA sequences are powerful tools for deciphering the genetic regulatory code. Existing methods involve a trade-off between input sequence length and prediction resolution, thereby limiting their modality scope and performance[1–5]. We present AlphaGenome, a unified DNA sequence model, which takes as input 1 Mb of DNA sequence and predicts thousands of functional genomic tracks up to single-base-pair resolution across diverse modalities. The modalities include gene expression, transcription initiation, chromatin accessibility, histone modifications, transcription factor binding, chromatin contact maps, splice site usage and splice junction coordinates and strength. Trained on human and mouse genomes, AlphaGenome matches or exceeds the strongest available external models in 25 of 26 evaluations of variant effect prediction. The ability of AlphaGenome to simultaneously score variant effects across all modalities accurately recapitulates the mechanisms of clinically relevant variants near the *TAL1* oncogene[6]. To facilitate broader use, we provide tools for making genome track and variant effect predictions from sequence.

Interpreting the impact of genome sequence variation remains a central biological challenge. Non-coding variants, which reside outside of protein-coding regions, are particularly challenging to interpret because of the diverse molecular consequences they can elicit. For example, non-coding variants can modulate genome properties such as chromatin accessibility, epigenetic modifications and three-dimensional chromatin conformation. Variants can further influence messenger RNA (mRNA) availability by altering expression levels or modifying sequence composition through splicing changes. Additionally, variants can exhibit cell-type-specific or tissue-specific effects. Given that more than 98% of observed genetic variation in humans is non-coding[7], global characterization of the complex effects of this vast majority of variants remains intractable without computational predictions.

Computational methods can learn patterns from experimental data to predict and explain variant effects. One class of methods, sequence-to-function models[1–5], takes a DNA sequence as input and predicts genome tracks, a data format associating each DNA base pair with a value (representing read coverage, count or signal) derived from experimental assays performed in cell lines or tissues. Genome tracks span various data modalities measuring gene expression (with output types comprising RNA sequencing (RNA-seq), cap analysis of gene expression (CAGE) sequencing, and precision nuclear run-on analysis of capped RNA (PRO-cap)), splicing (splice sites, splice site usage and splice junctions), DNA accessibility (DNase I hypersensitive site

sequencing (DNase-seq)) and assay for transposase-accessible chromatin sequencing (ATAC-seq)), histone modification (chromatin immunoprecipitation sequencing (ChIP-seq)), transcription factor binding or chromatin conformation (high-throughput chromosome (Hi-C) or micrococcal nuclease-based (Micro-C) conformation capture). Successfully trained sequence-to-function models accurately predict experimental measurements from input sequences. Furthermore, by comparing genome track predictions from an alternative sequence versus a reference sequence, these models can predict the molecular effects of variants.

Currently, deep learning-based sequence-to-function models face two fundamental trade-offs constraining their ability to predict how variants affect diverse modes of biological regulation. First, often owing to computational limitations, models must trade off between capturing long-range genomic interactions and achieving nucleotide-level predictive resolution. Although models such as SpliceAI[4], BPNet[8] and ProCapNet[9] provide base-resolution predictions, they are restricted to short input sequences (for example, 10 kb or less), and thus may miss the influence of distal regulatory elements. Models such as Enformer[1] and Borzoi[2] can process longer sequences (approximately 200–500 kb) to capture broader context but at the cost of reducing output resolution (128-bp or 32-bp bins), which can blur fine-scale regulatory features such as splice sites, transcription factor footprints or polyadenylation sites.

A second trade-off exists between capturing diverse modalities versus specializing in one or a few. Several state-of-the-art (SOTA) models

[1]Google DeepMind, London, UK. [2]These authors contributed equally: Žiga Avsec, Natasha Latysheva, Jun Cheng, Guido Novati, Kyle R. Taylor, Tom Ward, Clare Bycroft, Lauren Nicolaisen, Eirini Arvaniti, Joshua Pan. ✉e-mail: avsec@google.com; pushmeet@google.com

are highly specialized for single modalities, such as SpliceAI[4] for splice site prediction, ChromBPNet[10] for local chromatin accessibility and Orca[3] for three-dimensional genome architecture. However, specialized models alone are insufficient for capturing the diverse molecular consequences of variants across modalities. Even within a single modality like splicing, specialized models such as SpliceAI[4] or Pangolin[11] predict certain aspects (such as splice site prediction) while omitting others (such as splice junction prediction or competition between splice sites). Models like DeepSEA, Basenji, Enformer, Sei and Borzoi have demonstrated the utility and practicality of multimodal models. They allow users to use a single model for several modalities, instead of requiring several specialized models. Furthermore, their learned general sequence representation enables them to be readily fine-tuned for new tasks. However, these more generalist models can lag behind their specialized counterparts on certain tasks, such as splicing, or may lack particular modalities, such as contact maps.

Here we present AlphaGenome, a model that unifies multimodal prediction, long-sequence context and base-pair resolution into a single framework. The model takes 1 Mb of DNA sequence as input and predicts a diverse range of genome tracks across numerous cell types. The splicing predictions of AlphaGenome include a new splice junction prediction approach alongside splice site usage prediction. We evaluated the performance of AlphaGenome using a comprehensive set of benchmarks, covering both its ability to accurately predict genome tracks on previously unseen DNA sequences and its effectiveness in variant effect prediction tasks. AlphaGenome achieved SOTA performance on 22 of 24 genome track prediction tasks and 25 of 26 variant effect prediction tasks. We performed extensive ablations of target resolution, sequence length, distillation and modality combinations to explain the performance of AlphaGenome and inform design choices for future sequence-to-function models. We envisage that AlphaGenome will provide a powerful and extensible foundation for analysing the regulatory code within the genome.

We first present key technical details of the AlphaGenome data and training procedure, alongside a high-level summary of our evaluations (Fig. 1). We then demonstrate high-fidelity genome track prediction performance, a prerequisite for variant effect prediction (Fig. 2). Next, we focus on variant effect prediction with modality-specific deep dives into splicing (Fig. 3), gene expression (Fig. 4) and chromatin accessibility (Fig. 5). Finally, we highlight the model's utility in cross-modality variant interpretation (Fig. 6) and dissect the impact of modelling choices on the performance of AlphaGenome (Fig. 7).

## Unifying DNA sequence-to-function model

AlphaGenome is a deep learning model designed to learn the sequence basis of diverse molecular phenotypes from human and mouse DNA (Fig. 1a). It simultaneously predicts 5,930 human or 1,128 mouse genome tracks across 11 modalities covering gene expression (RNA-seq, CAGE and PRO-cap), detailed splicing patterns (splice sites, splice site usage and splice junctions), chromatin state (DNase, ATAC-seq, histone modifications and transcription factor binding) and chromatin contact maps. These span a variety of biological contexts, such as different tissue types, cell types and cell lines (see Supplementary Table 1 for the summary and Supplementary Table 2 for the complete metadata). These predictions are made on the basis of 1-Mb of DNA sequence, a context length designed to encompass a substantial portion of the relevant distal regulatory landscape. For instance, 99% (465 of 471) of validated enhancer–gene pairs fall within 1 Mb (ref. 12).

AlphaGenome uses a U-Net-inspired[2,13] backbone architecture (Fig. 1a and Extended Data Fig. 1a) to efficiently process input sequences into two types of sequence representations: one-dimensional embeddings (at 1-bp and 128-bp resolutions), which correspond to representations of the linear genome, and two-dimensional embeddings (2,048-bp resolution), which correspond to representations of spatial interactions between genomic segments. The one-dimensional embeddings serve as the basis for genomic track predictions, whereas the two-dimensional embeddings are the basis for predicting pairwise interactions (contact maps). Within the architecture, convolutional layers model local sequence patterns necessary for fine-grained predictions, whereas transformer blocks model coarser but longer-range dependencies in the sequence, such as enhancer–promoter interactions. Base-pair-resolution training on the full 1-Mb sequence is enabled through sequence parallelism across eight interconnected tensor processing unit (v3) devices. Genomic track predictions are linear transformations of these sequence embeddings, aside from splice junction count prediction, which uses a separate mechanism that captures interactions between one-dimensional embeddings of donor–acceptor pairs (Extended Data Fig. 1).

We trained the model using a two-stage process: pretraining and distillation. The pretraining phase (Fig. 1b) used the observed experimental data to produce two types of models. Fold-specific models were trained using a 4-fold cross-validation scheme (Methods), with three fourths of the reference genome used for training and the remaining one fourth held out for validation and testing. These models were then used to evaluate the generalization of AlphaGenome by predicting genomic tracks on unseen (test) reference genome intervals (Fig. 1b). Additionally, all-fold models were trained on all available intervals of the reference genome and served as teachers in the second stage (distillation; Fig. 1c). In the distillation phase, a single student model, sharing the pretrained architecture, was trained to predict the output of an ensemble of all-fold teachers using randomly augmented input sequences (Methods). This distilled student model, as shown previously[14], achieved improved robustness and variant effect prediction accuracy in a single model instance, making predictions across all modelled modalities and cell types with a single device call per variant. Taking less than 1 s on an NVIDIA H100 GPU, the student model is highly efficient for large-scale variant effect prediction relative to the alternative approach of ensembling several independently trained models.

## Performance overview

To characterize the model performance of AlphaGenome, we first assessed its generalization to unseen genome intervals, a prerequisite for high-quality variant effect prediction. We conducted 24 genome track evaluations, encompassing all 11 predicted modalities (Methods and Supplementary Table 3). For out-of-fold evaluations, pretrained, fold-specific AlphaGenome models were used and compared with the strongest available external model for each respective task. AlphaGenome outperformed these external models on 22 of 24 evaluations (Fig. 1d, Extended Data Fig. 3 and Supplementary Table 3). Notably, AlphaGenome exhibited a +14.7% relative improvement in cell-type-specific gene-level expression log-fold change prediction compared with Borzoi[2], another multimodal sequence model (Fig. 1e and stratified metrics in Extended Data Fig. 3e). AlphaGenome also outperformed specialized single-modality models on their respective tasks, such as Orca[3] on contact maps (contact map Pearson r +6.3%; cell-type-specific differences +42.3%; Fig. 1d and Extended Data Fig. 4), ProCapNet[9] on transcription initiation tracks (+15% total counts Pearson r; Extended Data Fig. 3f) and ChromBPNet[10] on accessibility (+1.6% for ATAC; +9.5% for DNase profile Jensen–Shannon divergence; Extended Data Fig. 3g).

We next evaluated the model's performance on predicting variant effects. We assembled a second set of 26 variant effect prediction benchmarks across gene expression, splicing, polyadenylation, enhancer–gene linking, DNA accessibility and transcription factor binding. Again, we compared with the strongest externally available model on each task (Methods and Supplementary Table 4). For variant

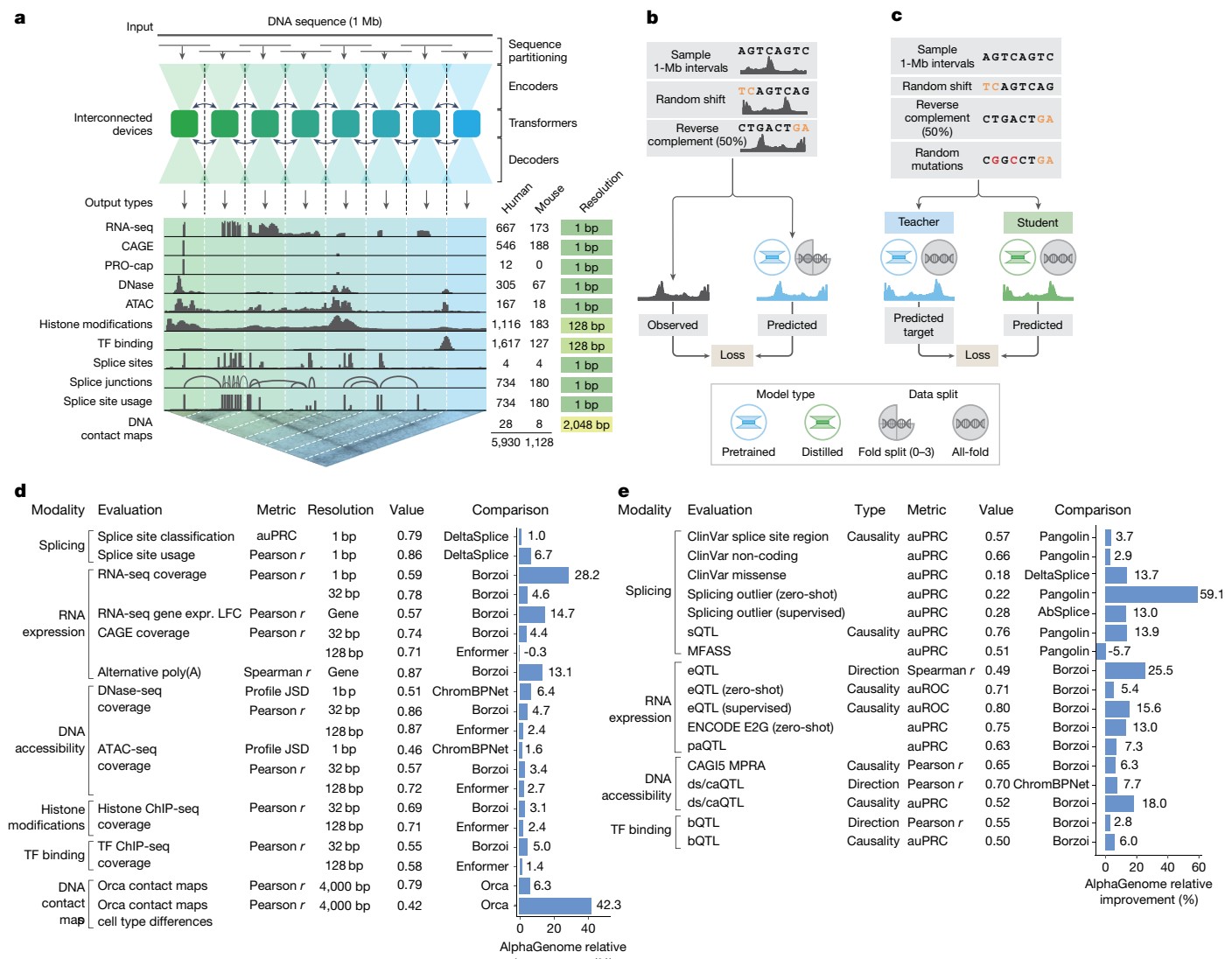

**Fig. 1 | AlphaGenome model architecture, training regimes and comprehensive evaluation performance. a**, Model overview. AlphaGenome processes 1 Mb of DNA sequences and species identity (human/mouse) to predict 5,930 human or 1,128 mouse genome tracks across diverse cell types and 11 output types at specific resolutions (far right). Computation leverages sequence parallelism, breaking the 1 Mb of DNA sequence into 131-kb chunks processed across devices. The core architecture features a U-Net-style design comprising an encoder (downsampling the sequence), transformers with inter-device communication and a decoder (upsampling), which feed into task-specific output heads at their respective resolutions (detailed in Extended Data Fig. 1). **b**, The pretraining process, in which 1-Mb DNA intervals are sampled from cross-validation folds, augmented (shifted and reverse complemented) and used to train the model against experimental targets, yields fold-specific and all-fold teacher models. **c**, The distillation process, in which a student model learns to reproduce predictions from frozen all-fold teacher models using augmented and mutationally perturbed input sequences, yields a single model suitable for variant effect prediction. **d**, Track prediction: pretrained

fold-split model. Relative performance improvement (%) of AlphaGenome over the best competing model for a selection of genome track prediction tasks across modalities and resolutions (Supplementary Table 3). The 'value' column represents the absolute performance of AlphaGenome. For all tasks shown, a value of 1.0 indicates perfect performance, with the exception of 'profile JSD', for which the ideal value is 0. Both competing models and AlphaGenome pretrained fold-split models were evaluated on held-out genome regions unseen during model training. For classification tasks, we adjusted the relative improvement to account for the performance of a random classifier (Methods). **e**, Variant effect prediction: distilled all-fold model. Relative performance improvement of AlphaGenome over the best competing model for a subset of variant effect prediction tasks (Supplementary Table 4). The distilled student AlphaGenome model is used for these evaluations. The ds/caQTL direction (causality) rows represent the average relative improvement across several similar datasets (Methods). ds, DNase sensitivity; ca, chromatin accessibility; JSD, Jensen–Shannon divergence.

effect prediction, we used the distilled student model. AlphaGenome matched or outperformed the external models on 25 of 26 evaluations (Fig. 1e and Supplementary Table 4). This included strong performance in quantitative trait locus (QTL) evaluations, such as sign prediction for expression QTLs (eQTLs; +25.5% versus Borzoi[2]) and accessibility QTL (+8.0% versus ChromBPNet[10], averaged across five datasets; Methods), demonstrating its strength against both multimodal and specialized single-modality baselines. Collectively, these results demonstrate

that AlphaGenome more accurately models both genome tracks and variant effects.

## Improved track prediction performance

Given the strong performance of AlphaGenome on genome track evaluations, we investigated its track predictions in more detail. Fold-specific, pretrained AlphaGenome models demonstrated high

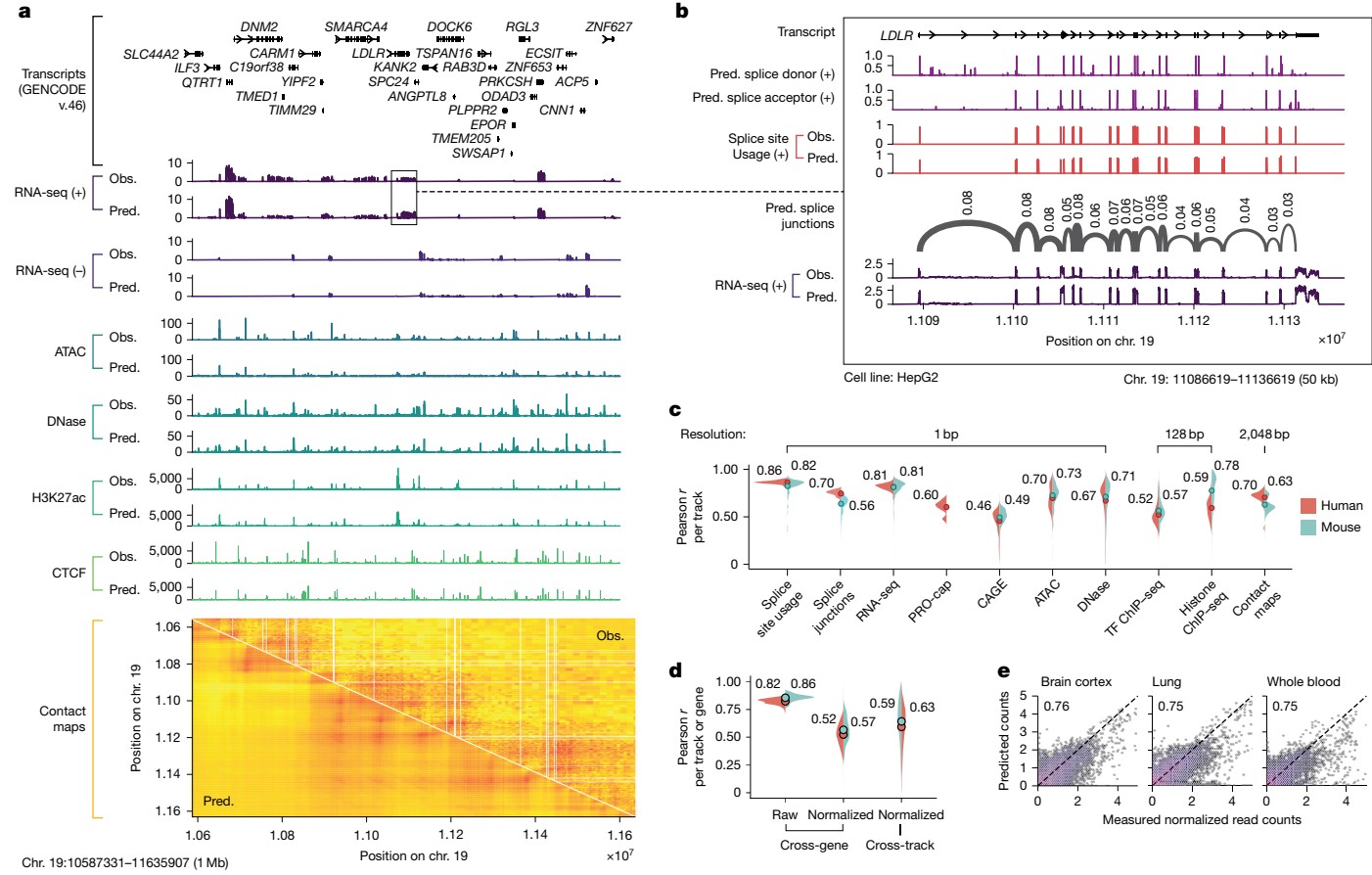

**Fig. 2 | Example of AlphaGenome track predictions and detailed performance evaluations. a**, Observed and AlphaGenome-predicted genome tracks within a 1-Mb held-out region on human chr. 19 (0-based coordinates: 10587331–11635907) in the HepG2 cell line. The *y*-axis scales for each assay are defined in the Methods section. Strand-specific tracks are denoted as positive (+) or negative (−), whereas strand-agnostic tracks are shown without a strand symbol. Contact maps are pairwise interaction matrices; therefore, both *x* and *y* axes display genome coordinate positions. RNA-seq, ATAC-seq and DNase-seq track predictions are at 1-bp resolution; H3K27ac and CTCF ChIP-seq are at 128-bp resolution; and contact maps are at 2,048-bp resolution. **b**, Example predictions with splicing. Base-pair-resolution AlphaGenome predictions for a 50-kb region highlighting detailed splicing (donor/acceptor sites, splice site usage and splice junctions) and RNA-seq predictions around the *LDLR* gene. **c**, Track prediction performance evaluation across different modalities. Violin plots display the distribution of Pearson correlations between predicted and observed tracks evaluated on held-out test intervals. Each violin plot is grouped by modality and split by organism (human in red; mouse in blue). Filled circles with accompanying

numerical values indicate the mean Pearson *r* per assay group and organism. Splice junction, RNA-seq, PRO-cap, CAGE and ChIP-seq tracks were log(1 + *x*) transformed, whereas the remainder were untransformed. **d**, Evaluation of RNA-seq gene log-expression prediction on held-out test intervals. The leftmost panel assesses the Pearson correlation between predicted and observed log-expression values across all genes within individual tracks. The middle and rightmost panels evaluate the prediction of tissue or cell-type specificity using quantile-normalized expression values (detailed in Methods); correlations are computed either across genes per track (middle) or across tracks per gene (right). **e**, Splice junction count prediction. Predicted versus observed splice junction read counts (log(1 + *x*) transformed; *n* = 1,344,738) and Pearson *r* between them in selected human tissues known for having distinct splicing patterns[49]. Each hexagonal bin is coloured by the density of the data points in that bin, with warmer colours corresponding to higher density. The diagonal dotted line indicates perfect agreement (predicted = observed). More tissues are shown in Extended Data Fig. 2d. Obs., observed; Pred., predicted.

concordance between predicted and observed read coverage on unseen genome intervals (Fig. 2a). As an example, predicted HepG2 genome tracks over the *LDLR* gene showcased strand-specific, base-pair-resolution RNA-seq coverage over exons, along with predicted splice sites, splice site usage and splice junction read coverage (Fig. 2b). More examples illustrating splicing, gene expression and chromatin track predictions are provided in Supplementary Figs. 1–3, and finer delineation of genomic features such as exon boundaries is highlighted in Supplementary Fig. 4.

Quantitatively, we observed strong Pearson correlations (*r*) between predicted and observed signals for functional genomics tracks in both human and mouse genomes (Fig. 2c), both across all tracks and when subsetting by biosample types or data sources (Supplementary Fig. 5). Although overall expression levels are predicted well, accurately capturing cell-type-specific expression deviations remains a challenging task (Fig. 2d and Supplementary Fig. 2j).

On splicing (Extended Data Fig. 2a), AlphaGenome accurately predicts splice sites (Extended Data Fig. 2b) and splice site usage (Extended Data Fig. 2b,c). It also accurately predicts quantitative splice junction read coverage and PSI5 and PSI3 within various tissues, achieving high correlation with experimental measurements (Fig. 2e, Extended Data Fig. 2b,d,e and Methods). Although AlphaGenome demonstrates the ability to predict tissue-specific alternative splicing in some instances (Supplementary Fig. 1), further improvements are needed to precisely predict intermediate splicing efficiencies and to capture tissue-specific nuances (Extended Data Fig. 2c,e).

## Improved splicing variant predictions

One of the main ways genetic variants cause disease is by disrupting splicing[15], a process that produces mature RNA sequences by excising

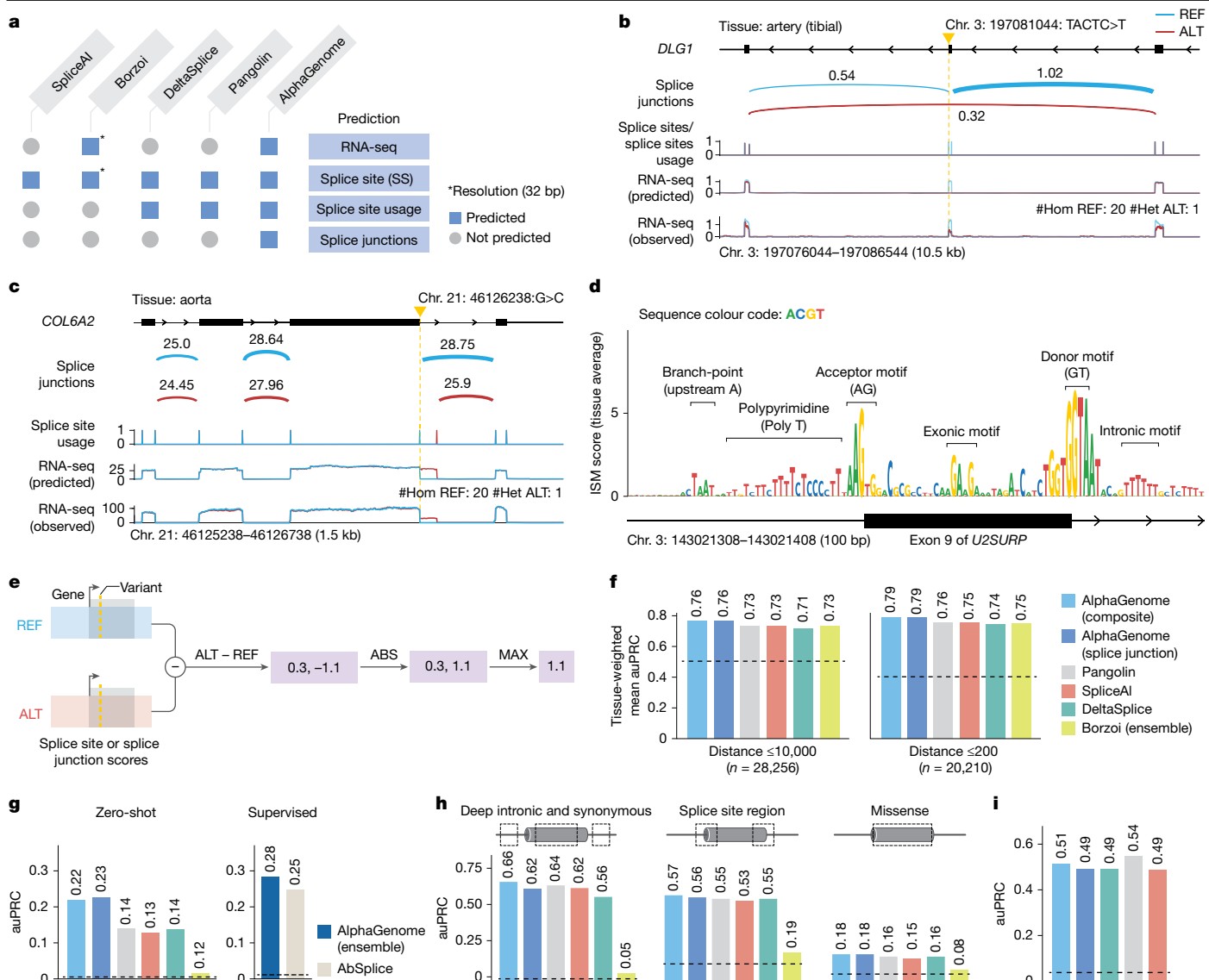

**Fig. 3 | AlphaGenome is a SOTA splicing variant effect prediction model.**
**a**, Comparison of prediction outputs across deep learning models. All models predict at 1-bp resolution, except Borzoi (32 bp). Borzoi predicts splice sites implicitly through RNA-seq coverage, whereas others produce explicit predictions. **b**, Variant causing exon skipping in *DLG1* (GTEx artery tibial tissue). Predicted splice junction, site usage and RNA-seq coverage are shown alongside observed coverage for reference (REF; blue) and alternative (ALT; red) alleles. **c**, New splice junction variant in *COL6A2* (aorta), creating a new splicing donor and disrupting the extant one. **d**, ISM of *U2SURP* exon 9 and flanking introns using the mean splice junction score across tissues. Splicing-related motifs are highlighted. **e**, Schema of splice variant effect prediction with AlphaGenome. The maximum difference between REF and ALT predictions across splice sites or splice junctions is used to score variants (Methods). **f**, Comparison of AlphaGenome composite and splice junction scorers versus other methods for classifying fine-mapped sQTL variants. Variants are stratified into two groups by distance to the splice site, as done in Borzoi[2]. Tissue-specific auPRCs were averaged and weighted by variant count per tissue. **g**, Prediction of rare variants associated with splicing outliers. AlphaGenome was evaluated in both zero-shot and supervised settings (training an ensemble model similar to AbSplice[50]). **h**, Classifying pathogenic versus benign ClinVar variants on the basis of splicing effects for deep intronic (more than 6 bp from splice sites) and synonymous (more than 3 bp from splice sites) variants, variants in the splice site region (within 6 bp intronic or 3 bp exonic) and missense variants predicted as 'likely_benign' by AlphaMissense[51]. **i**, MFASS splicing variant classification (MPRA-tested variants). auPRC on the classification of experimentally validated splice-disrupting variants (data from Chong et al.[22]). #Hom/#Het, number of homozygous/heterozygous samples in GTEx.

introns and ligating exons at splice junctions. Splicing outcomes can be modelled at three levels: the probability that any given nucleotide acts as a splice donor or acceptor (splice site prediction)[4,11,16], competitive selection among potential splice sites (splice site usage prediction)[11,16] and prediction of specific introns (splice junction prediction). AlphaGenome predicts all three of these quantities alongside direct RNA-seq coverage prediction, thereby providing a more comprehensive view of the splicing-related molecular consequences of variants (Fig. 3a).

To illustrate the capacity of AlphaGenome to simultaneously predict several relevant splicing variant effects, we first probed its ability to recapitulate known biological outcomes. We interrogated a 4-bp deletion (chr. 3:197081044: TACTC>T), a variant empirically observed to cause exon skipping in tibial artery tissue in a sample from the Genotype–Tissue Expression (GTEx)[17] project (Fig. 3b). AlphaGenome accurately predicted this established consequence across all levels: a substantial reduction in the predicted usage of the affected exon's splice site, loss of predicted junctions linking the skipped exon edges, emergence of a

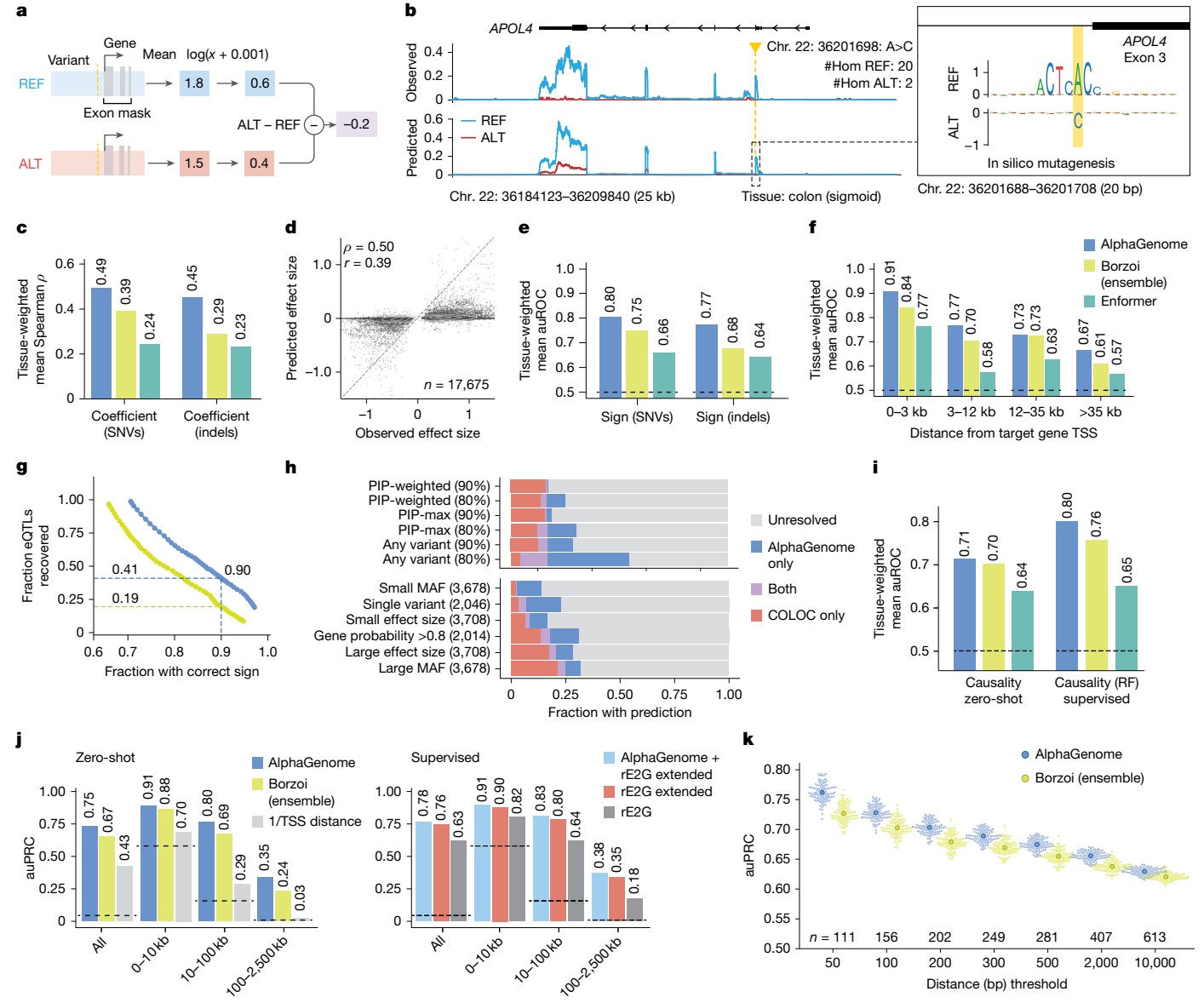

**Fig. 4 | AlphaGenome predicts the effect of variants on gene expression.**
**a**, RNA-seq variant scoring. Variant scoring strategy for predicting the effect of a genetic variant on the expression of a target gene (Methods). **b**, Example predictions for a known eQTL (chr. 22: 36201698: A>C) in GTEx colon (sigmoid) tissue. The observed RNA-seq coverage is the average across GTEx samples homozygous for either allele. Inset, comparative ISM on reference and alternative sequences over a 20-bp window centred on the variant (Methods). **c**, Comparison of performance (Spearman's $\rho$) at predicting the effect size of eQTLs across 49 GTEx tissues ('coefficient') for different models and variant sets. **d**, Comparison of AlphaGenome-predicted variant scores and observed effect sizes (SuSiE $\beta$ posterior) for 17,675 fine-mapped GTEx eQTLs (SNVs). Each point is a unique variant/gene/tissue combination. Spearman's $\rho$ (signed) = 0.50; Spearman's $\rho$ (unsigned; absolute values) = 0.10. Pearson's $r$ (signed) = 0.39; Pearson's $r$ (unsigned; absolute values) = 0.20. **e**, Comparison of performance (auROC) at predicting the direction of effect of eQTLs ('sign') for different models and variant sets. **f**, eQTL sign prediction performance stratified by different variant-to-TSS distance bins (SNVs only). **g**, Relationship between sign accuracy and eQTL recall. For a series of variant score thresholds, we plotted the fraction of GTEx eQTLs with a score above the threshold ($y$ axis) and sign accuracy achieved ($x$ axis) on those variants. **h**, Coverage of predictions across GWAS loci. Fraction of GWAS credible sets (from Open Targets[52]) with a predicted direction of effect for a plausible target gene, comparing AlphaGenome predictions to the eQTL co-localization approach. Top, each bar represents a different strategy for summarizing AlphaGenome scores, and two different score thresholds that yielded a given sign accuracy on eQTL of 80% or 90% (Methods). For COLOC, we counted a credible set as resolved if $H_4 > 0.95$. Bottom, using the AlphaGenome strategy of PIP-weighting (80%), credible sets were further stratified by different properties (Methods). **i**, Comparison of performance (auROC) at distinguishing causal from non-causal eQTLs ('causality') using both zero-shot and supervised approaches (Methods). **j**, Enhancer–gene linking performance (ENCODE–rE2G CRISPRi dataset[12]). Zero-shot evaluation: performance (auPRC) comparison stratified by enhancer-to-TSS distance. Supervised evaluation: AlphaGenome input gradient score integrated into ENCODE–rE2G extended and ENCODE–rE2G models. **k**, Performance (auPRC) of paQTL variant effect prediction, thresholded by distance from the polyadenylation site. Each swarm plot represents 100 permutations of randomly matching each positive SNP with one of its distance and expression-matched negatives (Methods). Larger dots are the mean. RF, random forest.

putative junction bypassing the exon and strong decrease in predicted RNA-seq coverage of the exon. Similarly, the predictions of Alpha-Genome accurately captured the new splice junction and extended exon induced by the chr. 21: 46126238: G>C variant, an effect observed in a heterozygous GTEx RNA-seq sample (Fig. 3c). Finally, in silico mutagenesis (ISM), which systematically predicts effects of all possible

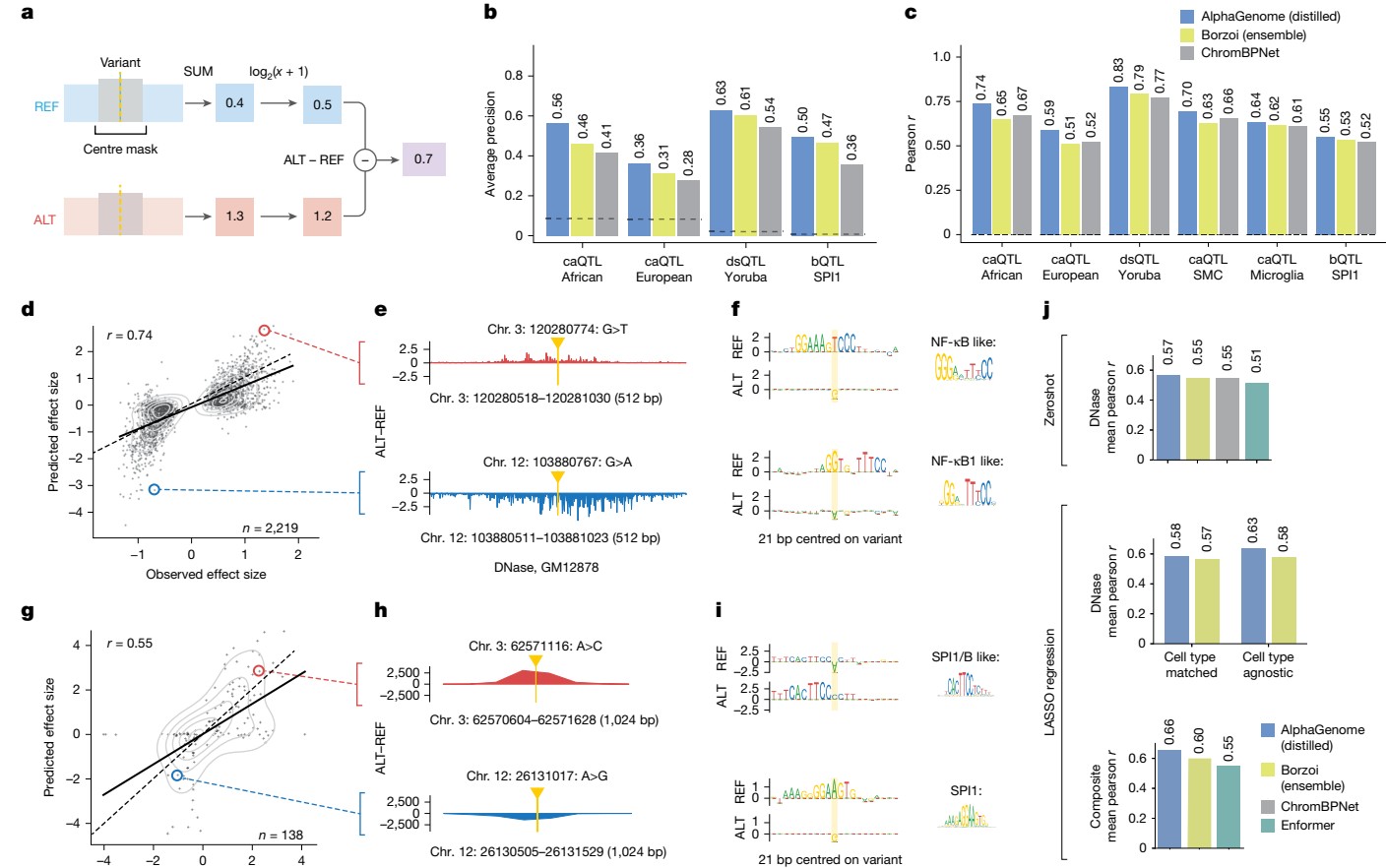

**Fig. 5 | AlphaGenome accurately predicts variant effects on chromatin accessibility and SPI1 transcription factor binding. a**, Schematic of the centre-mask variant scoring strategy used for accessibility and ChIP-seq predictions (Methods). **b,c**, Performance comparison of AlphaGenome, Borzoi and ChromBPNet on QTL causality (**b**; average precision) and QTL effect size (**c**; Pearson *r*) across QTL types and ancestries. **d**, Predicted versus observed effect sizes for causal caQTLs (African ancestry). The scatterplot displays GM12878 cell line DNase predictions. Signed Pearson *r* = 0.74; unsigned Pearson *r* = 0.45. Signed Pearson *r* correlation uses raw values; unsigned Pearson *r* uses absolute values. Red and blue circles highlight variants in **e** and **f**. **e**, Example ALT−REF differences in predicted DNase (GM12878) for variants in **d. f**, ISM-derived sequence logos for REF/ALT alleles from **e**, suggesting variant disruption or modulation of transcription factor binding motifs. Putative binding factors and JASPAR[53] matrix IDs (MA0105.1 and MA0105.3) are indicated on the right.

**g**, Predicted versus observed effect sizes for causal SPI1 bQTLs using the GM12878 SPI1 ChIP-seq track. Signed Pearson *r* = 0.55; unsigned Pearson *r* = 0.12. Red and blue circles highlight variants in **h** and **i. h**, Example AlphaGenome predictions for selected SPI1 bQTLs. Shown are ALT−REF differences in predicted SPI1 ChIP-seq track (GM12878) around the variants highlighted in **g. i**, ISM-derived sequence logos for REF and ALT alleles of example SPI1 bQTLs from **h**, suggesting potential impacts such as creation or disruption of SPI1 or related motifs. The putative binding factors and JASPAR matrix IDs (MA0081.2 and MA0080.5) are indicated on the right. **j**, CAGI5 MPRA challenge performance (average across loci; mean Pearson *r*). Top, zero-shot using cell-type-matched DNase; middle, LASSO regression using cell-type-matched or agnostic DNase; bottom, LASSO regression using multimodal features (DNase + RNA + histone ChIP-seq output types for AlphaGenome and Borzoi; DNase + CAGE output types for Enformer) and all cell types. TF, transcription factor.

single nucleotide variations in a sequence region (Methods), revealed the sequence determinants of the splicing predictions. For example, ISM analysis of exon 9 of the *U2SURP* gene and its flanking introns highlighted recognizable splicing-related sequence motifs[18,19] (Fig. 3d). Further examples of experimentally validated splice-disrupting variants identified in individuals with autism spectrum disorder[4] are shown in Supplementary Fig. 6.

Building on the multifaceted splicing predictions of AlphaGenome, we developed a unified splicing variant scorer to systematically detect splice-disrupting variants. Specifically, we designed a custom variant scoring strategy for each prediction modality (Fig. 3e and Methods) and summed the individual scores to provide a composite measure of a variant's predicted effect. We benchmarked this composite scorer against existing methods on a wide range of splicing-related variant effect prediction tasks. AlphaGenome performed best on fine-mapped splicing QTL (sQTL) classification[2,20], including both single-nucleotide polymorphisms (SNPs) within 10 kb of the closest splice site and proximal variants within 200 bp of splice sites

(Fig. 3f). Furthermore, it achieved the highest performance at predicting rare SNPs and indels associated with GTEx splicing outliers in both supervised and unsupervised settings (Fig. 3g and Methods). We also evaluated the performance of AlphaGenome in distinguishing pathogenic variants from benign variants in ClinVar[21], in which its composite scores outperformed the existing best method in each category across all three variant categories: deep intronic and synonymous (area under the precision–recall curve (auPRC) 0.66 versus 0.64 by Pangolin), splice region (auPRC 0.57 versus 0.55 by Pangolin) and missense (auPRC 0.18 versus 0.16 by DeltaSplice and Pangolin; Fig. 3h and Supplementary Fig. 7). When assessing the ability of AlphaGenome to predict whether rare variants disrupt splicing using data from a massively parallel splicing minigene reporter assay (multiplexed functional assay of splicing using SORT-seq (MFASS))[22], it was outperformed by Pangolin (auPRC 0.54 versus 0.51) but surpassed SpliceAI and DeltaSplice (both auPRC 0.49; Fig. 3i). Notably, the splice junction scorer alone outperformed previous approaches on all benchmarks, except 'deep intronic and synonymous' ClinVar

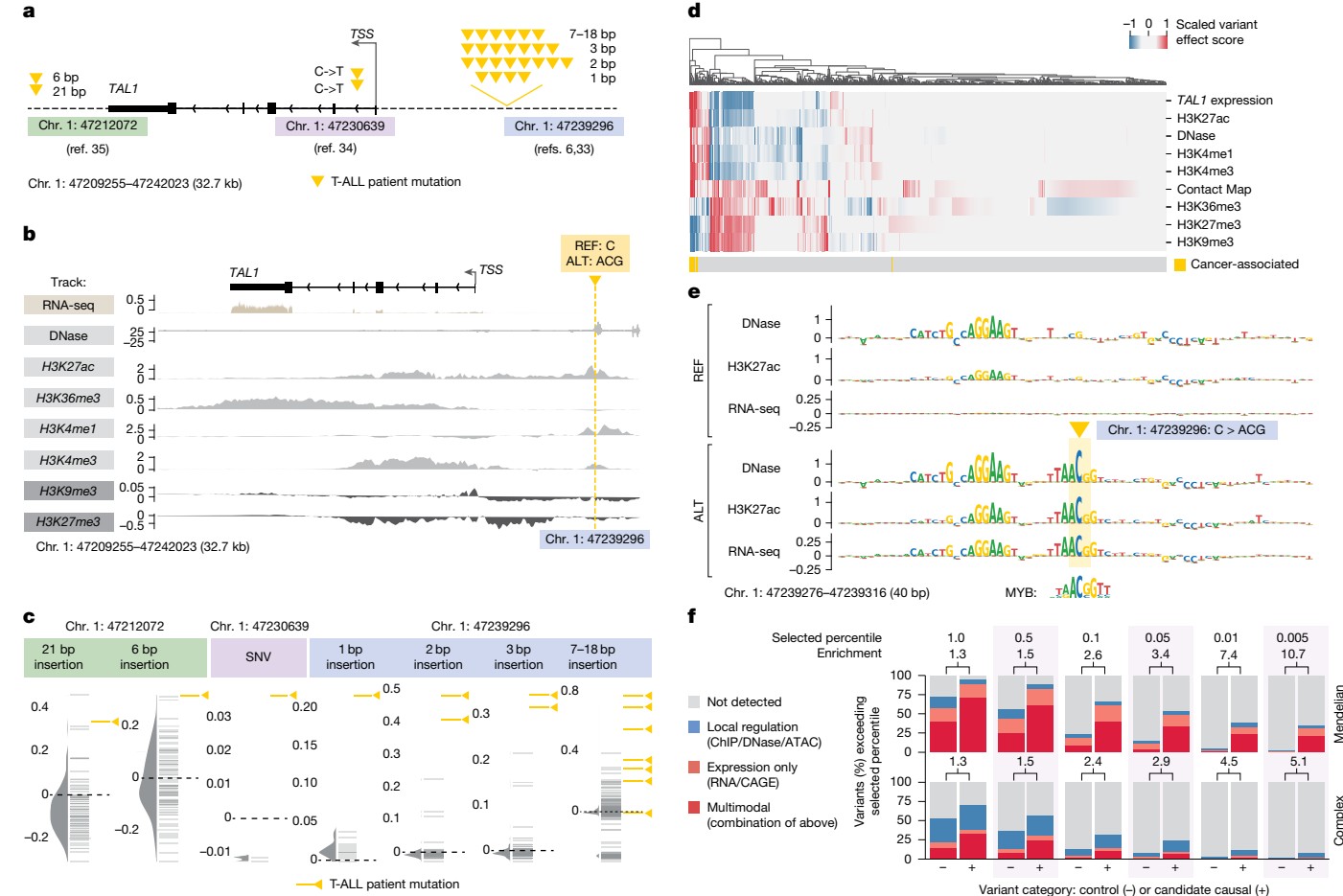

**Fig. 6 | Interpreting variant effects across modalities with AlphaGenome.**
**a**, Non-coding cancer mutations in T-ALL. Overview of groups of mutations affecting *TAL1* in patients with T-ALL. **b**, Detailed ALT–REF predictions for an oncogenic insertion (chr. 1: 47239296: C>ACG) characterized in ref. 6. Shown are differences between AlphaGenome predictions between the ALT and REF sequences of the variant in CD34+ CMP tracks. The ALT sequence increases expression of the *TAL1* gene 7.5 kb away. **c**, Predicted TAL1 expression change (ALT–REF) in CD34+ CMPs. RNA-seq variant scores for *TAL1* expression in CD34+ CMPs. Oncogenic mutations (orange) are compared with randomly sampled, length-matched indels (grey). **d**, Multimodal heat map of predicted variant effects. Each column is a distinct variant from **c**. Each row is a variant effect score associated with a genome track in CD34+ CMPs, except for contact map variant effect scores, which were averaged across tissues (as there is no CD34+ CMP contact map in our data). Background mutations are included alongside oncogenic mutations. Variants were grouped by their insertion length and position (as displayed in Fig. 6c), and scores were min-max scaled. **e**, ISM results

for DNase, H3K27ac and *TAL1* RNA-seq expression prediction by AlphaGenome in CD34+ CMPs. Top, ISM on the reference sequence; bottom, ISM on the oncogenic insertion sequence (chr. 1: 47239296: C>ACG). Myb motif from a previous study[6], originally from UniPROBE[54]. **f**, Multimodality in trait-altering non-coding variants. Fraction of trait-affecting variants[55] ('candidate causal'; 338 for Mendelian and 1,140 for complex traits), as well as matched control variants[55] ('control'; 3,042 and 10,260, respectively), which exceed varying quantile-score thresholds in at least one predicted track. Here, surpassing a quantile-score threshold of 1.0 implies a predicted effect in excess of 99% of common variants (Methods). Variants are categorized depending on the tracks where the threshold was passed: 'local regulation' (ChIP/DNase/ATAC), 'expression only' (RNA/CAGE) and 'multimodal' (combination of the above). Numbers above the bars indicate the relative enrichment of detected variants (sum of the three categories) among candidate causal variants compared with the control variants. The enrichment increases with stricter thresholds, with a reduction in recall (*x* axis).

and MFASS variants, underscoring the importance of modelling splicing at the junction level. In summary, AlphaGenome achieves SOTA splicing variant effect prediction on six of seven benchmarks, providing a more comprehensive view of altered splicing events and transcript structure.

## Performance across gene expression tasks

Beyond impacting isoform composition through splicing modulation, non-coding variants can influence traits and cause diseases by altering gene expression[23,24]. We evaluated the ability of AlphaGenome to predict the impact of variants on gene expression across a range of regulatory mechanisms, including eQTLs, enhancer–gene interactions and alternative polyadenylation (APA).

## Improved prediction of eQTL effects

We first evaluated the ability of AlphaGenome to predict the impact of eQTLs, which are variants associated with gene expression variation. We developed a variant scoring strategy to quantify a variant's predicted effect on a gene's expression level (Fig. 4a and Methods). An illustrative example of a known eQTL/sQTL locus (rs9610445; chr. 22: 36201698: A>C) is shown in Fig. 4b. The alternative 'C' allele is associated with lower expression of *APOL4* in GTEx data (sum of single effects (SuSiE) '$\beta$ posterior' = −0.709; posterior inclusion probability (PIP) > 0.9), and both the RNA-seq coverage and direction of effect are recapitulated by the predictions of AlphaGenome (variant score = −1.52; quantile score = −1.00; Methods). Furthermore, ISM around the variant indicates a possible nucleotide splice donor sequence motif,

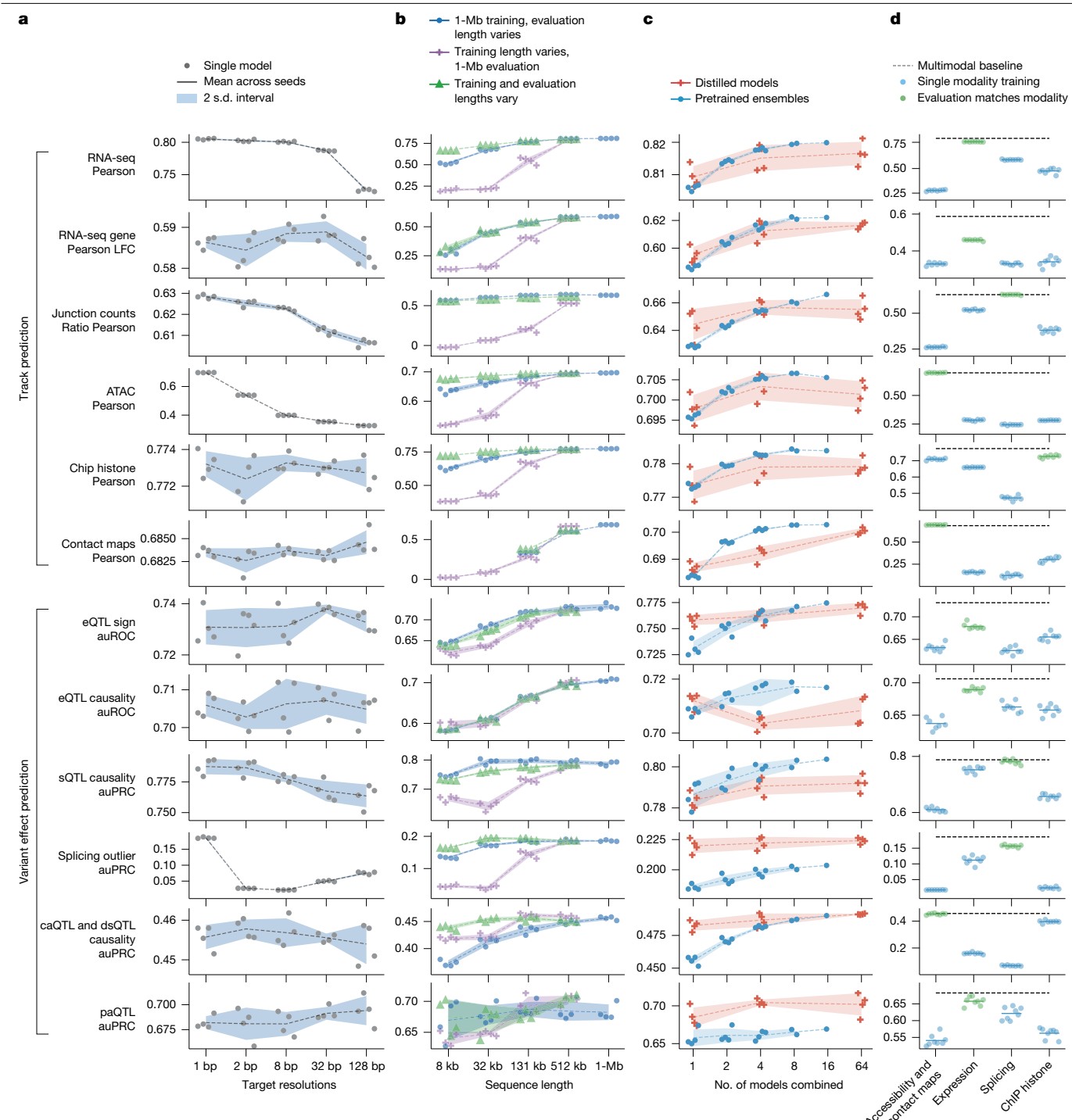

**Fig. 7 | Impact of resolution, sequence length, ensembling, distillation and multimodal training on AlphaGenome performance.** Ablation studies evaluating key model design choices across various performance metrics (*y* axis). For all panels, lines represent the mean over replicate training runs with different random seeds (*n* = 4 unless otherwise stated), and shaded contours denote the uncertainty interval (two standard deviations). **a**, Impact of target resolution. Performance comparison across models trained to predict targets (DNA accessibility, gene expression and splicing) at varying resolutions (*x* axis; 1–128 bp). **b**, Impact of sequence length during training and inference. Blue dots represent a single set of models trained with 1-Mb input, evaluated using varying input sequence lengths (*x* axis). Purple crosses represent models trained at the sequence length indicated on the *x* axis but evaluated at a fixed 1-Mb input length. Green triangles represent models trained and evaluated using the same matched sequence length (*x* axis). **c**, Impact of the number of sub-models in

ensembling and distillation. Performance comparison for mean ensembles of pretrained models (blue dots/contours; *x* axis indicates ensemble size) versus single models produced by distillation using 1, 4 or 64 teacher models (orange crosses/contours; *x* axis indicates number of teachers). **d**, Impact of multimodal learning. Performance comparison evaluating models trained only on specific modality groups (blue dots; *n* = 8 seeds per group, highlighted in green if the modality matches the evaluation metric) against the full multimodal model (black dashed line; *n* = 4 seeds average). During training for these models, we ensured that only the target modality group's prediction heads contributed updates to the shared representations, allowing assessment of that modality group's contribution to overall model performance. Groups shown (*x* axis) include models trained using gradients only from accessibility (ATAC, DNase and contact maps), expression (RNA-seq, CAGE and PRO-cap), splicing (sites, usage and junctions) or histone ChIP-seq.

which is disrupted by the variant, thereby leading to an aberrant transcript and reduced expression (Fig. 4b; inset). More examples of eQTL variant prediction and mechanistic interpretation by ISM are shown in Supplementary Fig. 8.

Using fine-mapped GTEx eQTL as ground truth, we benchmarked AlphaGenome against the SOTA models Borzoi[2] and Enformer[1]. AlphaGenome demonstrated improved prediction of both the magnitude ('coefficient'; Spearman $\rho$ with SuSiE[25] $\beta$ posterior) and direction ('sign'; area under the receiver operating characteristic (auROC) curve) of eQTL effects compared with the previous SOTA (Borzoi) (Fig. 4c–f). AlphaGenome improved the tissue-weighted mean Spearman $\rho$ from 0.39 to 0.49 and the mean sign auROC from 0.75 to 0.80. These improvements in coefficient and sign prediction were observed broadly across most of GTEx tissues, variant-to-transcription start site (TSS) distance bins and variant functional annotation classes (Extended Data Fig. 5a–c).

As previously reported[26], performance decays with distance to the target gene across all eQTL tasks (Extended Data Fig. 5f). However, AlphaGenome exhibited mild improvement on sign/coefficient prediction for distal variants (greater than 35 kb; Fig. 4f and Extended Data Fig. 5b), and top-scoring predictions exhibited high sign accuracy across distance categories (Extended Data Fig. 5g). Additionally, AlphaGenome outperformed Borzoi on coefficient and sign prediction for indel (insertion or deletion) eQTLs (Fig. 4c,e and Extended Data Fig. 5d). Notably, the effect size predictions of AlphaGenome for high-confidence eQTLs (scores greater than 99th percentile of common variant effects) highly correlate with observed effects (Spearman $\rho = 0.73$; Extended Data Fig. 5h) and consistently surpassed Borzoi's performance across various quantile score thresholds (such as Spearman $\rho$ 0.73 versus 0.61 at approximately 99th percentile threshold; Extended Data Fig. 5i).

The overall improvement in prediction accuracy, particularly for the sign of a variant's effect, translates to substantial gains in practical applications. At a score threshold yielding 90% sign prediction accuracy, AlphaGenome recovered over twice as many GTEx eQTLs (41%) as Borzoi (19%; Fig. 4g). Applying this improved sign prediction capability to genome-wide association study (GWAS) interpretation, we evaluated the ability of AlphaGenome to assign a direction of effect to candidate target genes for 18,537 GWAS credible sets (Methods). Using a threshold calibrated to 80% accuracy on eQTLs (Fig. 4g), AlphaGenome assigned a confident sign prediction for at least one variant in 49% of GWAS credible sets (11% using a conservative PIP-weighted scoring approach; Fig. 4h). AlphaGenome and a widely used co-localization method for sign prediction (COLOC $H_4 > 0.95$)[27] resolved the direction of effect for largely non-overlapping sets of loci (Fig. 4h). This indicates their complementary utility, collectively increasing the total yield of loci with determined effect directions. Furthermore, AlphaGenome resolved approximately 4-fold more credible sets in the lowest minor allele frequency quintile compared with COLOC, probably reflecting its reduced dependence on population genetics parameters that affect power to detect associations (Fig. 4h, stratified bars). Thus, AlphaGenome expands our ability to generate functional hypotheses about the direction of GWAS signals, particularly for low-frequency variants.

The performance of AlphaGenome on distinguishing fine-mapped eQTLs from distance-matched variants ('causality'; auROC) was comparable with Borzoi (Fig. 4i). However, leveraging the predictions of AlphaGenome within a supervised framework boosted performance on the causality task; training a random forest model using the scores of AlphaGenome from several modalities increased the mean auROC from 0.68 to 0.75, also surpassing previous SOTA performance (mean auROC 0.71 for Borzoi; Fig. 4i). Notably, using features derived from variant scores across all predicted modalities provided a performance uplift in this supervised setting compared with using RNA-seq-derived scores alone (Extended Data Fig. 5e), highlighting the practical benefit of the multimodal predictions of AlphaGenome for identifying causal expression-modulating variants.

## Competitive enhancer–gene linking

We then assessed whether AlphaGenome can link enhancer elements to their target genes, given that tissue-specific gene expression is modulated by enhancer–promoter interactions, often involving enhancers in distal genomic regions. For this task, we leveraged an independent CRISPRi perturbation dataset from the ENCODE–rE2G study[12]. Evaluated zero-shot, AlphaGenome outperformed Borzoi in identifying validated enhancer–promoter links, particularly for enhancers located beyond 10 kb from the TSS of the target gene (Fig. 4j, Extended Data Fig. 7a and Methods), although both models still underestimate the impact of very distal enhancers (Extended Data Fig. 7d). Furthermore, the zero-shot performance of AlphaGenome was comparable (within 1% auPRC) with the ENCODE–rE2G (extended) model, which was explicitly trained on this task and cell line data (Fig. 4j). It also strongly outperformed the simpler DNase-based ENCODE–rE2G model and a distance-to-TSS baseline. Beyond their stand-alone predictive power, features derived from AlphaGenome improved the supervised enhancer–promoter linking models. Incorporating AlphaGenome predictions into the ENCODE–rE2G (extended) model yielded a new SOTA performance across all distance-to-TSS categories (Fig. 4j, Extended Data Fig. 7b,c and Methods).

Altogether, these results demonstrate the improved capacity of AlphaGenome to capture long-range functional regulatory connections directly from sequence, which is vital for interpreting distal genetic variants.

## Improved prediction of 3′ polyadenylation QTL effects

We further evaluated AlphaGenome on variants affecting APA. APA is a process that generates transcript diversity by varying the 3′ untranslated region (3′ UTR) of mRNA molecules, often with concomitant impacts on RNA half-life and tissue specificity[28]. By predicting RNA-seq coverage, AlphaGenome intrinsically models the competition between proximal and distal polyadenylation signals (PASs) and thus can be used to detect APA. We found that AlphaGenome achieves SOTA performance in predicting APA (Spearman $\rho = 0.894$ versus Borzoi's 0.790; Extended Data Fig. 6a and Methods)[2]. Given this, we next examined its ability to distinguish 3′ polyadenylation QTLs (paQTLs) from expression-matched negatives (Fig. 4k and Methods). AlphaGenome outperformed Borzoi on this task (all paQTLs within 10 kb of PAS, auPRC 0.629 versus Borzoi's 0.621; proximal paQTLs within 50 bp of PAS, auPRC 0.762 versus Borzoi's 0.727). AlphaGenome outperformed Borzoi at all considered distances, although the predictive accuracy of both models declined with distance to the polyadenylation site (Fig. 4k and Supplementary Table 5). ISM revealed that AlphaGenome learned the relevance of the canonical polyadenylation motif and could detect variants that disrupt or create this motif (Extended Data Fig. 6c–h), although it occasionally underpredicted the expression impact (Extended Data Fig. 6i,j). Overall, these results demonstrate the improved accuracy of AlphaGenome in predicting genetic variant effects on 3′ UTR processing, a distinct regulatory mechanism, using only RNA-seq predictions and without any explicit training on polyadenylation data or associated variants.

## Improved prediction of chromatin accessibility, DNase sensitivity and binding QTLs

We next assessed the ability of AlphaGenome to predict the effects of variants on chromatin states, focusing on QTLs associated with chromatin accessibility (caQTLs), DNase sensitivity (dsQTLs) and transcription factor binding (bQTLs). Variant effects were scored by comparing model predictions for reference and alternative alleles in a local window around the variant (Fig. 5a and Methods). Benchmarking performance using fine-mapped QTLs from diverse ancestries (using benchmarks from ChromBPNet[29]), AlphaGenome consistently achieved SOTA performance compared with both Borzoi and the specialized

ChromBPNet model. This was observed across QTL types and ancestries, both in predicting QTL causality (average precision; Fig. 5b and Supplementary Fig. 9) and in correlating with experimentally determined effect sizes ('coefficient'; Pearson $r$; Fig. 5c and Supplementary Fig. 9). This strong performance generalized across more datasets, including European ancestry caQTLs, Yoruba ancestry DNase sensitivity QTLs and predictions within specific cell types such as microglia and cardiac smooth muscle cells. To illustrate this performance and gain mechanistic insight, we examined specific examples, including fine-mapped caQTLs (African ancestry) and SPI1 bQTLs[30]. In both cases, AlphaGenome outperformed alternative models in causality prediction (precision–recall curves; Supplementary Fig. 9e,h), and its predicted effect sizes correlated with observed values (Fig. 5d,g; Pearson $r = 0.74$ for caQTLs and $r = 0.55$ for SPI1 bQTLs). Furthermore, ISM applied to high-impact variants revealed that predicted changes in accessibility or transcription factor binding scores generally corresponded to altered motifs for transcription factors known to modulate chromatin accessibility, such as NF-κB (Fig. 5e,f). For SPI1-specific variants, ISM further highlighted altered SPI1 motifs within the local sequence context (Fig. 5h,i).

Beyond population-level QTLs, understanding the impact of local sequence context on gene regulation is crucial. We therefore assessed the performance of AlphaGenome on the 5th Critical Assessment of Genome Interpretation (CAGI5) saturation mutagenesis massively parallel reporter assay (MPRA) challenge[31,32]. MPRAs measure the regulatory activity of short DNA sequences (typically through reporter gene expression), a process closely linked to local chromatin accessibility and transcription factor binding. We therefore evaluated this benchmark using DNase, RNA-seq and ChIP output types, comparing with Enformer, Borzoi and ChromBPNet (Methods and Extended Data Fig. 8). Notably, the cell-type-matched DNase predictions of AlphaGenome achieved performance comparable to both ChromBPNet and the Borzoi Ensemble (Fig. 5j, top; Pearson $r = 0.57$). Furthermore, aggregating features from DNase predictions across all modelled cell types using least absolute shrinkage and selection operator (LASSO) regression further improved performance over cell-type-matched DNase features and the analogous approach with the Borzoi Ensemble model (Fig. 5j, middle; Pearson $r = 0.63$). Finally, by integrating features across several modalities and all cell types with LASSO, AlphaGenome achieved SOTA performance on the CAGI5 benchmark (Fig. 5j, bottom; Pearson $r = 0.65$). Overall, these analyses highlight the strong performance of AlphaGenome in predicting variant effects on chromatin states in QTL benchmarks, as well as its ability to leverage accessibility predictions to model the regulatory activity of sequences as measured by MPRAs.

## Multimodal view of variant effects

We next investigated how the unified modelling approach of AlphaGenome can be used to virtually screen a locus. We evaluated three groups of gain-of-function mutations affecting the *TAL1* gene in T cell acute lymphoblastic leukaemia (T-ALL): a cluster of 5′ neo-enhancer mutations upstream of the *TAL1* TSS[6,33], an intronic single-nucleotide variant (SNV)[34] and a 3′ neo-enhancer[35]. All three variant groups were shown to converge on a common mechanism—upregulation of the *TAL1* oncogene (Fig. 6a,b).

To assess AlphaGenome, we analysed predictions in CD34[+] common myeloid progenitor (CMP) data, the closest available match to the T-ALL cell of origin. For the oncogenic mutation chr. 1: 47239296: C>ACG[6], AlphaGenome predicted increases in the activating histone marks H3K27ac and H3K4me1 at the variant, consistent with experimentally observed neo-enhancer formation at that position[6,33] (Fig. 6b). It also predicted decreased levels of the repressive histone marks H3K9me3 and H3K27me3 near the *TAL1* TSS and elevated active transcription mark H3K36me3 across the *TAL1* gene body, both of which are concordant with its predicted increase in *TAL1* mRNA expression levels (Fig. 6b).

We expanded our analysis to include all oncogenic variants across studies[6,33–35], comparing each variant to a background set of length-matched and sequence-shuffled control variants. For the sole intronic SNV, we compared it with other possible SNVs at that site. Oncogenic mutations were predicted to increase *TAL1* expression in CD34[+] CMPs more than shuffled controls (Fig. 6c). Across modalities, oncogenic variants exhibited a distinct predicted mechanism compared with shuffled controls, as shown through unsupervised clustering (Fig. 6d). Finally, these observations are tissue-specific. By reproducing this analysis across all RNA-seq tracks, we found that the difference in oncogenic versus shuffled mutations is most pronounced in T-ALL-relevant tracks such as 'thymus', 'CMP' and 'haematopoietic multipotent progenitor cells' (Supplementary Fig. 10).

To understand the sequence determinants of the predictions of AlphaGenome in CD34[+] CMP, we performed ISM on the reference and alternative sequences of the oncogenic chr. 1: 47239296: C>ACG variant. No mutations within 40 bp of the variant had a predicted effect on *TAL1* expression in the reference sequence (Fig. 6e). By contrast, the alternative sequence introduced a MYB motif at the variant position predicted to increase *TAL1* expression, chromatin accessibility and H3K27ac chromatin marks (Fig. 6e), as discovered previously[6]. The model additionally identified a second ETS-like motif nearby, which affected *TAL1* expression in the alternative sequence but not in the reference sequence. The role of this motif is currently unknown.

We then quantitatively evaluated the utility of AlphaGenome for analysing trait-altering non-coding variants. Although models of molecular effect do not directly predict phenotypic consequences (Supplementary Note (trait-altering variants) and Supplementary Fig. 11), the predictions of AlphaGenome could nevertheless prove useful for interpreting the gene-regulatory implications of such variants. Setting a high threshold on the quantitative scores of AlphaGenome (for example, predicted expression change) strongly enriched for causal variant candidates among matched controls (Fig. 6f), although at the cost of low recall, particularly for GWAS variants. Inspection of some of these predictions for Mendelian disease variants (Extended Data Fig. 9a–h) illustrates how the multimodal outputs of the model can be used to simultaneously elucidate the effects of a variant on transcription factor binding, accessibility and gene expression while also predicting the direction of variant-induced expression changes. This makes AlphaGenome a useful complement to conservation-based measures of deleteriousness, which are agnostic to the mechanism of action of a variant.

## Model and data ablations

To understand the contributions of key design and training decisions to the performance of AlphaGenome, we conducted ablation studies evaluating the impact of target resolution, sequence length, ensembling, distillation and multimodal learning (Fig. 7). First, we found that base-pair resolution is important for achieving the highest performance. Training with target tracks at 1-bp resolution consistently yielded the best results, particularly for tasks requiring fine-scale detail, such as splicing (PSI5 and PSI3) and accessibility (ATAC), in which performance generally declined as target resolution became coarser (Fig. 7a). By contrast, performance on tracks with coarser assay-specific resolution, such as contact map correlation or histone ChIP-seq correlation, was relatively insensitive to the target resolution used during training. Similarly, variant effect prediction metrics, which often aggregate effects over larger regions (such as gene bodies or exons), were also robust to resolution changes.

We next evaluated the performance of AlphaGenome across various combinations of training sequence lengths and inference context lengths, finding that training with 1-Mb input sequences and subsequently performing inference using the full 1-Mb context yielded the best overall results (Fig. 7b). First, models trained on longer DNA

sequences (up to the full 1 Mb) outperformed those trained on shorter sequences (32 kb or less), even when the latter were evaluated using 1-Mb contexts (Fig. 7b (purple crosses) and Methods), underscoring the benefits of providing more sequence context during training. Second, inference also benefited from longer context. The 1-Mb-trained AlphaGenome model performed optimally when evaluated on the full 1-Mb sequence, with its performance progressively declining as shorter segments were used for inference input (Fig. 7b (blue dots)). Finally, the 1-Mb-trained model, even when assessed on shorter inference contexts, often achieved comparable performance to models that were specifically trained and evaluated at those same matched shorter lengths (Fig. 7b (green triangles)). This implies that the same model trained on the full 1-Mb sequence can also be used with a more limited context at inference time, providing an option for further increasing prediction speed if needed.

We compared distillation (Fig. 1c) with standard ensembling as a strategy to efficiently achieve high performance (Fig. 7c). Distillation using many teacher models (for example, 64 (orange crosses)) produced single models with performance that was competitive with, and sometimes surpassed, mean ensembles of several independently pretrained models (for example, the four-model ensemble (blue dots)). Distillation of even just a single teacher model was beneficial for some variant effect prediction tasks, including caQTL, splicing outlier and eQTL sign prediction tasks. Finally, distillation without randomly mutating the input sequence resulted in a drop in student model performance (eQTL sign, −0.06; eQTL causality, −0.01; sQTL causality, −0.01; splicing outlier, −0.015), affirming the impact of our input perturbation strategy during distillation. Overall, distillation offers a way to achieve strong performance with reduced inference cost compared with evaluating large ensembles, facilitated by scalable training strategies such as assigning one unique teacher per training device (Methods).

Lastly, we assessed the importance of multimodal learning (Fig. 7d and Supplementary Fig. 12). The fully multimodal model generally outperformed models trained on single modality groups, confirming the overall benefit of integrating diverse data types for learning shared representations (Fig. 7d). This benefit was task-dependent; for example, predicting accessibility variants was effective using only accessibility data, whereas predicting eQTLs benefited from the full multimodal input (a more complete set of experiments is shown in Supplementary Fig. 12). Complementary experiments showed that excluding single modality groups during training typically resulted in only modest performance decreases, suggesting redundancy between modalities (Supplementary Fig. 12, left panel). Training on single modalities was often detrimental to variant effect prediction performance (Supplementary Fig. 12, middle panel). Cumulative addition experiments showed that although track predictions improved most when their relevant data types were added, variant effect tasks benefited most from the initial inclusion of expression and accessibility data, with diminishing returns from subsequent modality additions (Supplementary Fig. 12, right panel). Together, these results demonstrate the value of the unified multimodal approach for achieving broad and high performance, particularly for tasks integrating diverse regulatory signals.

## Discussion

AlphaGenome advances efforts to decipher the regulatory code of the genome, offering a unified deep learning model that simultaneously predicts diverse functional genomic signals from megabase-scale DNA sequences. It matches or surpasses specialized SOTA models in regulatory variant effect prediction, underscoring the model's relatively robust grasp of fundamental DNA regulatory principles and its value for mechanistically interpretating non-coding variation. A core strength is its efficient multimodal variant effect prediction, which simultaneously scores variant impacts across all predicted modalities in a single inference pass. This integrated capability is crucial for understanding variants with complex mechanisms, as illustrated by the recapitulation of oncogenic *TAL1* variant effects, and could power large-scale analyses that dissect regulatory sequence elements genome-wide. Furthermore, the new capability of AlphaGenome to directly model splice junctions enables a more holistic view of splice-disrupting variants.

Going forward, AlphaGenome holds promise across diverse biological disciplines. For molecular biology, AlphaGenome can serve as an engine for in silico experimentation, enabling rapid hypothesis generation and prioritization of resource-intensive wet-lab experiments. For rare disease diagnostic research, the improved variant effect predictions of AlphaGenome could provide further functional evidence to current annotation pipelines, offering new diagnostic avenues for non-coding variants of uncertain significance. The improved splicing, expression and accessibility predictions of AlphaGenome could be used 'in the loop' to accelerate sequence design applications, such as therapeutic antisense oligonucleotides[36] and tissue-specific enhancers[37,38]. Finally, AlphaGenome can complement the capabilities of generative models trained on DNA sequences by predicting functional properties of newly generated sequences[39]. To widely enable these broad applications, we provide accessible tooling to access AlphaGenome through a hosted model and API.

Despite its advances, AlphaGenome shares challenges common to current sequence-based models and has specific scope limitations. Accurately capturing the influence of distal regulatory elements (more than 100 kb away) remains a continuing objective. Moreover, although the model predicts tissue-specific and cell-type-specific genome tracks with some success, accurately recapitulating tissue-specific patterns across cellular contexts and predicting condition-specific variant effects remain challenging. Both our training data and evaluations are heavily focused on protein-coding genes, and future work can be done to improve coverage of non-coding genes such as microRNAs. Our species coverage remains limited to human and mouse, and the evaluations in this study are primarily human-focused. We have not yet benchmarked the model on personal genome prediction, which is a known weakness of models in this space[40,41]. Finally, application to complex trait analysis is limited because AlphaGenome predicts molecular consequences of variants, whereas these phenotypes involve broader biological processes (including gene function, development and environmental factors) and gene-to-disease effects[42] beyond the direct sequence-to-function scope of the model.

Addressing these limitations motivates several future research directions. Data generation efforts could increase input genome diversity by assaying more species or by perturbing non-coding regulatory elements at scale to help build the next generation of variant effect prediction models. Key computational avenues include refining variant prediction accuracy and utility (such as through task-specific calibration, fine-tuning on perturbational datasets or integration of single-cell data[43,44]), incorporating a broader range of data modalities (such as DNA methylation and RNA structural features) and pursuing foundational model improvements (such as leveraging DNA language models[45–47], expanding multi-species capabilities and developing robust methods for assay bias correction[29,48]). Moreover, integrating AlphaGenome predictions with other measures of variant effect, such as conservation-based scores, as well as existing data on gene function and biological pathways could prove useful in advancing common and rare-variant analysis. Finally, estimates of model certainty would aid in better interpreting predictions.

In summary, AlphaGenome provides a powerful and unified model for analysing the regulatory genome. It advances our ability to predict molecular functions and variant effects from DNA, offering valuable tools for biological discovery and enabling applications in biotechnology. Ultimately, AlphaGenome serves as a foundational step towards the broader scientific goal of deciphering the complex cellular processes encoded in DNA sequences.

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

# Methods

For detailed descriptions of data acquisition and processing, model architecture, training procedures (pretraining and distillation) and benchmarking protocols against existing methods, refer to the Supplementary Information.

## Reporting summary

Further information on research design is available in the Nature Portfolio Reporting Summary linked to this article.

## Data availability

All primary experimental datasets used for the training and evaluation of AlphaGenome in this study were obtained from publicly accessible sources. A comprehensive manifest detailing these data sources, including specific repositories (such as ENCODE portal, GTEx portal, 4D Nucleome portal, ClinVar and gnomAD), individual accession numbers, relevant version information and direct URLs where applicable, is provided in Supplementary Table 2. This study did not generate new primary experimental data requiring deposition.

## Code availability

AlphaGenome is available for non-commercial use through an online API at http://deepmind.google.com/science/alphagenome, with an accompanying Python software development kit provided to interact with the model. We also provide a genome interpretation suite to facilitate the exploration and interpretation of AlphaGenome. This offers a range of functionalities, such as streamlined variant scoring with quantile calibration and identification of critical sequence regions through contribution scores from ISM-based experiments. The model source code, weights, variant scoring implementations and a selection of variant evaluation datasets and predictions are available at https://github.com/google-deepmind/alphagenome_research.

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

**Acknowledgements** We thank D. Hariharan, C. Taylor and O. Bertolli for product and program management support, and M. Beck and U. Okereke for legal counsel. We are grateful to Y. Assael and A. Botev for their contributions to modelling and engineering and to A. Trostanetski, L. Tenório, V. Johnston, R. Green and S. Chakera for their work on the API release. We also thank K. Tunyasuvunakool for feedback on the paper; R. Tremlett for assistance with graphic design and figure preparation; and former interns I. I. Taskiran, A.-A. Muşat, R. Khan and R. Yi for their contributions to the earlier stages of this research. Gemini (Google) was used for assistance with language editing and improving the clarity of the paper. This study extensively used publicly available datasets. We extend our sincere gratitude to the numerous research consortia (including, but not limited to, ENCODE, GTEx, FANTOM5, ACMG, OMIM and the 4D Nucleome Project), their members, data contributors and all study participants whose efforts in generating and openly sharing these foundational genomic resources were essential for this study. This study was funded by Google DeepMind.

**Author contributions** Z.A. conceptualized the study with input from N.L., J.C., C.B., T.W., G.N., K.R.T., P.K. and D.H. Z.A. led the project, with N.L. leading manuscript preparation, G.N. leading overall model development, J.C. conceiving and leading the splicing-related research and K.R.T. and T.W. leading engineering. Z.A. and G.N. conceived the device-distributed model architecture with input from R.T. G.N. conceived the contact-map pair stack. J.C. and G.N. conceived and developed the splice junction prediction head, with contributions from Z.A. Z.A. and G.N. conceived model distillation with input from J.C., V.D. and K.R.T. G.N. developed the model with contributions from Z.A., J.C., K.R.T., T.W., E.A., V.D., S.D., A.G., A.M., A. Kosiorek and A.S. Z.A., J.C. and G.N. curated, processed and prepared the training data with contributions from L.N., K.R.T., T.W., J.P., M.P., T.S. and A.M. T.W. developed the data pipeline with contributions from Z.A., J.C., G.N., K.R.T. and L.N. Z.A., J.C., K.R.T., T.W., L.N. and E.A. designed the variant scoring infrastructure with contributions from N.L., C.B., R.T., T.A. and G.N. G.N., K.R.T., T.W., L.N., E.A. and V.D. optimized the model and variant scoring runtime. C.B. conceived and developed the variant score normalization with contributions from Z.A. and T.W. T.W. conceived and developed the model serialization and inference infrastructure with contributions from Z.A., K.R.T. and L.N. L.N. developed the evaluation infrastructure with contributions from N.L., K.R.T., T.W., R.T., V.D. and A.M. Z.A., N.L., J.C., G.N., C.B., L.N., E.A., J.P., R.T., V.D., M.P., A. Kosiorek, A.G., A.M. and P.D. evaluated the model. Specific evaluation areas were covered as follows: Z.A. and V.D. covered track evaluation; J.C. and E.A. covered splicing track and variant evaluation; N.L. and C.B. covered eQTL; E.A. and J.P. covered enhancer-to-gene linking; A. Karollus covered eQTL indel effects; N.L. covered accessibility QTL; J.P. covered multimodal examples and ACMG case studies; N.L. and G.N. covered contact maps; C.B. covered GWAS and complex trait analysis with help from A. Karollus, J.P. and T.A.; R.T. and M.P. covered paQTL and CAGI5; A. Karollus covered TraitGym. Z.A., N.L., J.C., G.N., K.R.T., T.W., C.B., L.N., E.A., J.P., R.T., V.D., M.P. and A. Karollus analysed data and prepared figures. Z.A., N.L., J.C., G.N., K.R.T., T.W., C.B., L.N., E.A., J.P., R.T., V.D., M.P., A. Karollus, A.G. and S.B. wrote the paper. L.H.W. managed and coordinated the project planning and execution. Z.A. and P.K. managed the project. All authors reviewed the paper.

**Competing interests** This study was conducted in the course of employment at Google DeepMind, with no competing financial interests.

**Additional information**
**Correspondence and requests for materials** should be addressed to Žiga Avsec or Pushmeet Kohli.

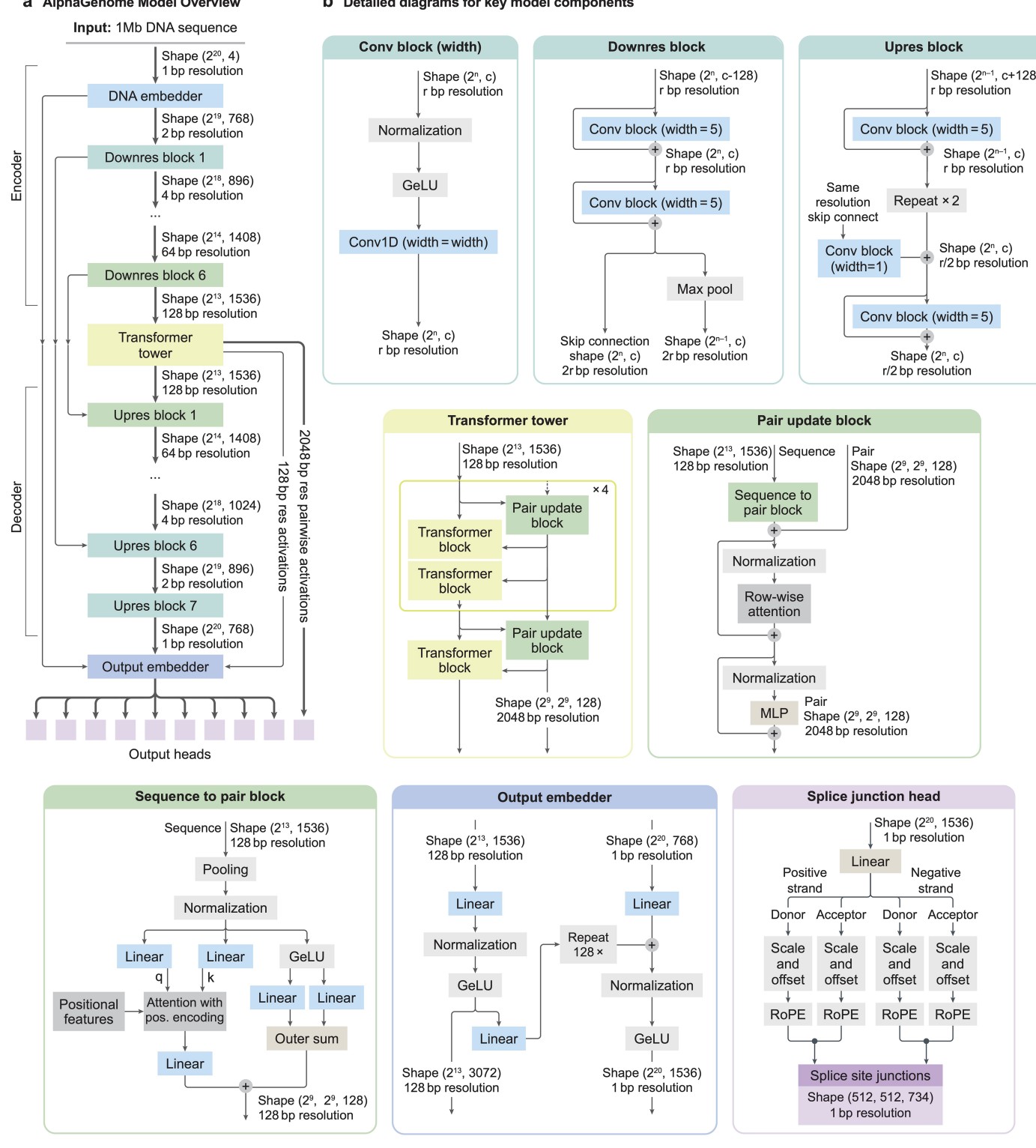

**Extended Data Fig. 1 | AlphaGenome model architecture. (a)** The architecture follows a U-Net-like structure with an Encoder, a central Transformer Tower, and a Decoder processing a 1-Mb DNA input sequence. The Encoder uses convolutional blocks and max pooling to progressively downsample the sequence resolution (from 1 bp to 128 bp) while increasing feature channels. The Transformer Tower operates at 128 bp resolution, iteratively refining sequence representations and generating pairwise (2D) representations. The Decoder uses convolutional blocks and upsampling, incorporating skip connections (dashed lines) from corresponding Encoder stages, to restore sequence resolution up to 1 bp.

An Output Embedder performs final processing before feeding representations to task-specific output heads. **(b)** Internal structure of key component blocks used repeatedly within the architecture overview shown in (**a**). Diagrams detail the layers within the convolutional blocks (Conv block, Upres block), the Transformer blocks, and the blocks responsible for generating and updating pairwise representations (Pair update block, Sequence to pair block). Tensor shapes are shown excluding the batch dimension. Abbreviations: r = log-resolution, c = channels.

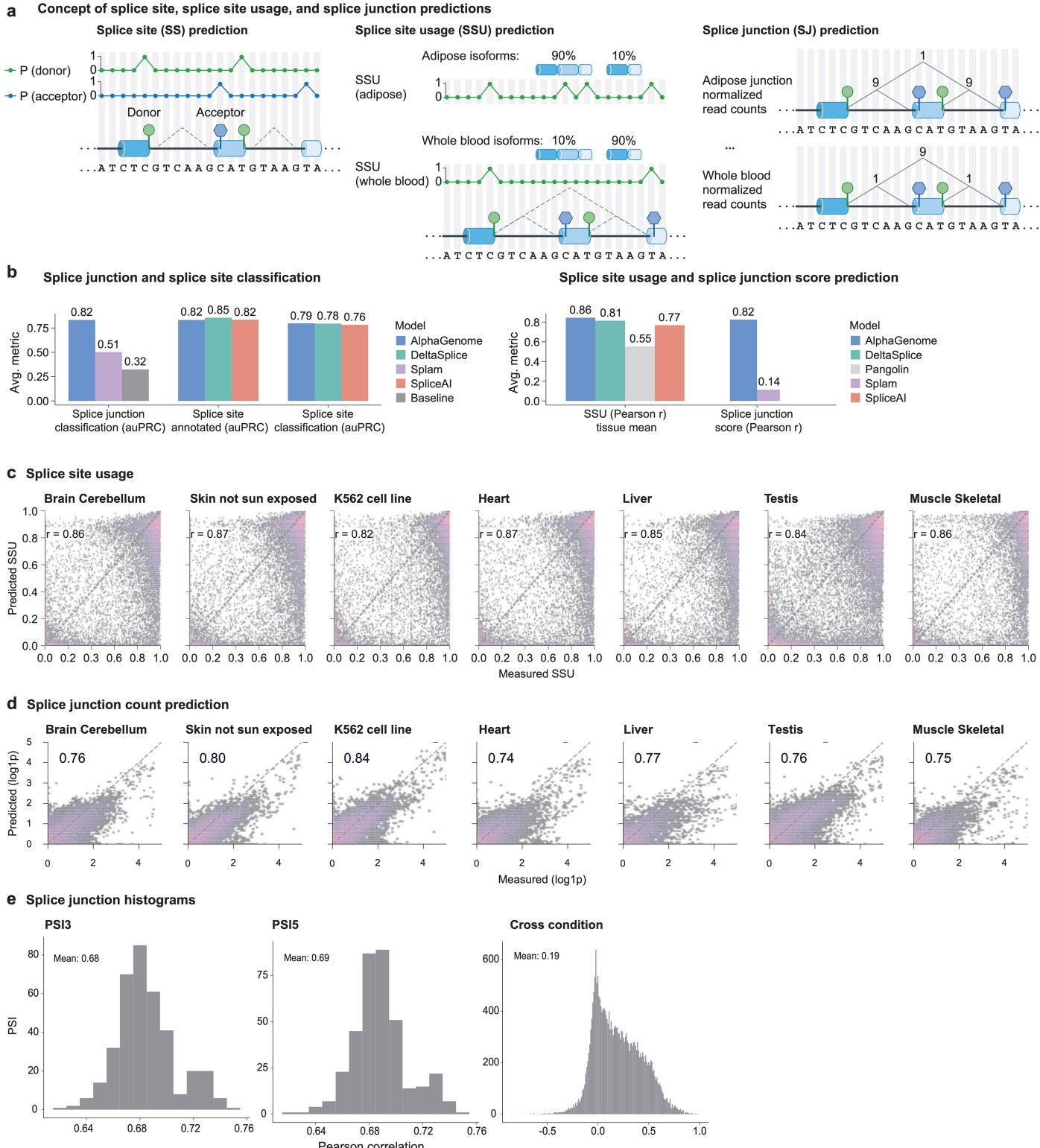

**Extended Data Fig. 2 | Splicing track performance.** (**a**) Schematic overview of splice site (SS) classification, splice site usage (SSU) prediction, and splice junction (SJ) read count prediction tasks. (**b**) (left) Performance comparison (AUPRC) of SS classification and SJ classification against reference methods. 'Baseline' means the fraction of positive splice junctions in the evaluated data. Splice site classification is evaluated with both GTF (GENCODE v46) annotated splice sites only and also splice sites derived from GTEx RNA-seq data (Methods). Splice junction classification discriminates between true splice junctions observed from RNA-seq data versus false junctions not observed from RNA-seq (but where the splice sites are observed). Splice junction classification was evaluated per tissue and then the mean AUPRC across tissues were reported. (right) Performance comparison (Pearson r) of predicted vs. measured SSU and SJ counts (log(1+x) transformed). (**c**) Scatter plot between predicted and measured donor SSU across seven example human tissues (from GTEx). Pearson r in each tissue is displayed as text. (**d**) Scatter plot between predicted and measured splice junction counts across seven human tissues (from GTEx). Pearson r in each tissue is displayed as text. (**e**) Distribution of Pearson correlation coefficients between predicted and measured PSI3 per tissue (left), PSI5 per tissue (middle), and junction counts across tissues (measuring tissue specificity of the splice junction predictions).

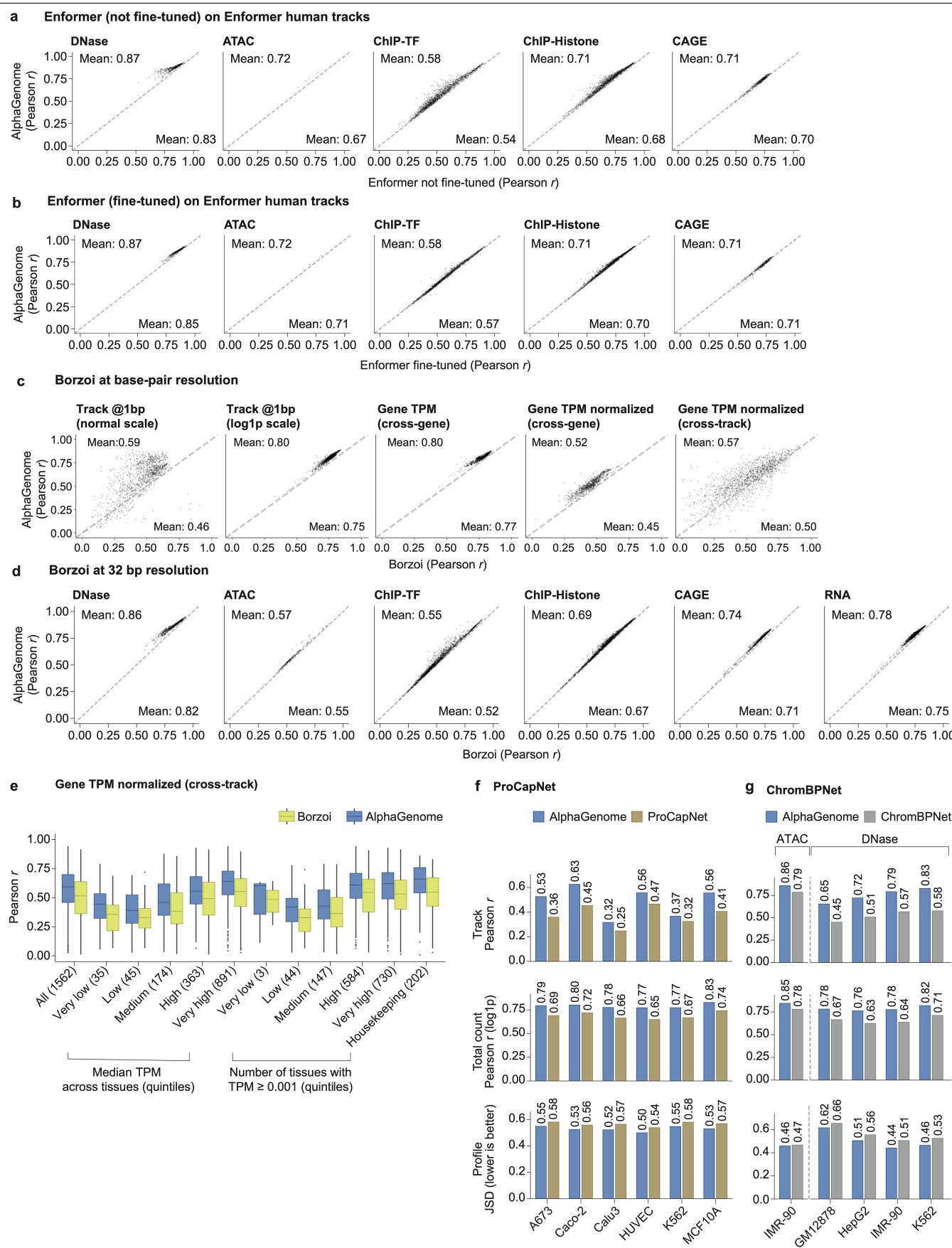

**Extended Data Fig. 3** | See next page for caption.

**Extended Data Fig. 3 | Track-level performance benchmarking.** Performance comparison of AlphaGenome with Enformer and Borzoi on held-out genomic track prediction. (**a, b**) Comparison of AlphaGenome test set performance on Enformer human tracks (each dot is one track) against Enformer models either (**a**) not fine-tuned or (**b**) fine-tuned on human data (the main released Enformer version). AlphaGenome model was re-trained for direct comparability using matched training intervals and an additional Enformer prediction head (Methods). (**c**) Evaluation of RNA-seq prediction performance at base and gene resolution using the same source of RNA-seq data as Borzoi, but processed at base-resolution and not scaled (Methods). Borzoi's 32 bp RNA-seq predictions were upsampled and unscaled to the original scale for comparison. The larger performance difference observed on the normal scale (first column) likely reflects resolution differences at exon-intron boundaries. This difference decreased when using log(1+x) transformed values (second column), suggesting better agreement on overall gene expression levels. A similar trend was observed when aggregating expression per gene (average exon coverage, third column). Cell-type specificity was evaluated by correlating quantile-normalized, mean-subtracted expression profiles across genes (fourth column) and across tracks (fifth column). (**d**) Test set performance comparison of AlphaGenome against Borzoi (fold 1) on Borzoi track data at 32 bp resolution (each dot is one track).

AlphaGenome was fine-tuned with an additional Borzoi head at matched resolution (Methods). (**e**) Stratification of cell-type specific prediction accuracy. The per-gene log-fold change correlation performance (from panel c, fourth column) was stratified by gene characteristics: median expression level across tissues (Median TPM; quintile breakpoints: $5.5 \times 10^{-9}$, $4.1 \times 10^{-4}$, $8.1 \times 10^{-4}$, 0.17, 4.1, $3.6 \times 10^{4}$ TPM), number of tissues with the gene expressed (TPM $\geq$ 0.001; quintile breakpoints: $9.4 \times 10^{-8}$, $9.4 \times 10^{-4}$, 8.0, 52, 54, 54 tissues), and housekeeping gene status. Sample sizes in brackets are the number of genes in each category. Box plots display the median (center line), the 25th and 75th percentiles (box bounds), and the whiskers extend to 1.5 times the interquartile range from the box bounds; points beyond whiskers indicate outliers. (**f, g**) Performance comparison of AlphaGenome against (**f**) ProCapNet (on PRO-Cap data) and (**g**) ChromBPNet (on ATAC and DNase). Evaluation was performed on ProCapNet fold 5 and ChromBPNet fold 0 test peak regions, respectively, where regions overlapping with AlphaGenome fold 0 training intervals were excluded. Performance is quantified by track Pearson r, Pearson r on the log total count, and Jensen-Shannon distance (JSD; lower indicates better performance). AlphaGenome outperforms the baselines across all metrics, modalities and cell-lines. For (**g**), only tracks with matching experiment accessions between AlphaGenome and ChromBPNet training sets were considered.

## a  Example contact map predictions

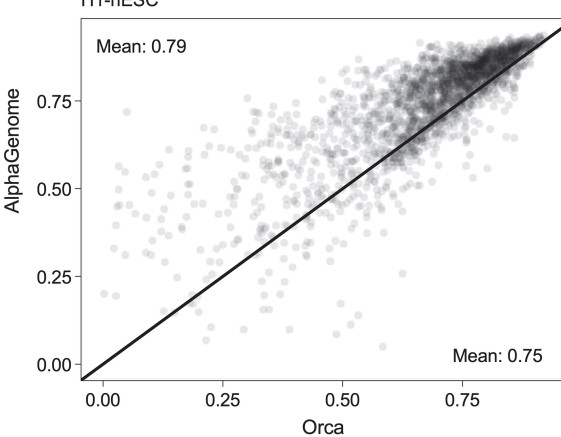

**Ground Truth (H1-hESC)**

chr10:120714877-121714877 | chr10:104438614-105438614 | chr9:26702627-27702627 | chr9:14753588-15753588 | chr9:2165300-3165300

**Orca**

r = 0.718 | r = 0.794 | r = 0.798 | r = 0.596 | r = 0.730

**AlphaGenome**

r = 0.856 | r = 0.823 | r = 0.834 | r = 0.708 | r = 0.854

## b  Contact map prediction performance

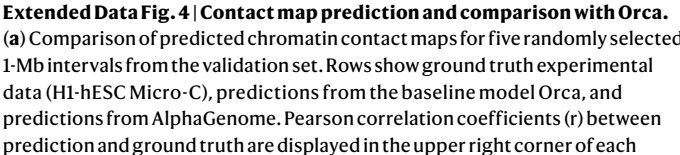
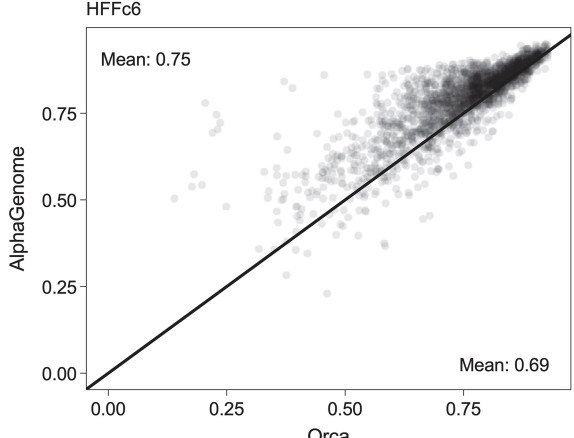

H1-hESC — AlphaGenome (y-axis) vs Orca (x-axis); Mean: 0.79, Mean: 0.75

HFFc6 — AlphaGenome (y-axis) vs Orca (x-axis); Mean: 0.75, Mean: 0.69

**Extended Data Fig. 4 | Contact map prediction and comparison with Orca.**
(**a**) Comparison of predicted chromatin contact maps for five randomly selected 1-Mb intervals from the validation set. Rows show ground truth experimental data (H1-hESC Micro-C), predictions from the baseline model Orca, and predictions from AlphaGenome. Pearson correlation coefficients (r) between prediction and ground truth are displayed in the upper right corner of each corresponding predicted map. (**b**) Scatterplot comparing overall contact map prediction performance of AlphaGenome and Orca across all 1-Mb test intervals, for the two cell-types predicted by Orca. Each point represents an interval, plotting the Pearson R achieved by AlphaGenome (y-axis) versus Orca (x-axis) when comparing predictions to the ground truth. Points above the diagonal indicate intervals where AlphaGenome achieved higher correlation than Orca.

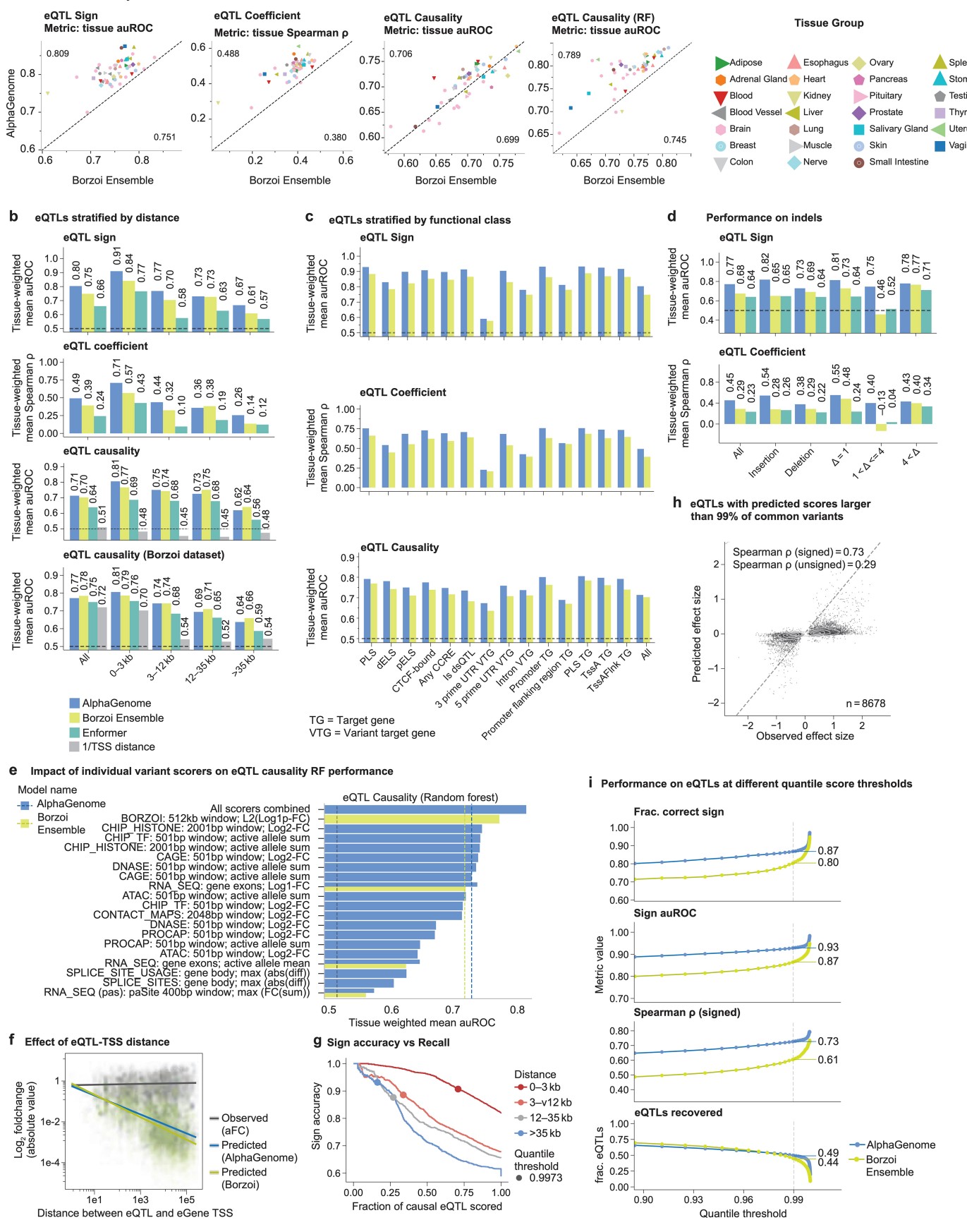

**Extended Data Fig. 5 |** See next page for caption.

**Extended Data Fig. 5 | Additional eQTL analysis.** Further characterization and stratification of AlphaGenome's eQTL prediction performance. (**a**) Performance comparison on eQTL tasks (Coefficient Spearman R, Sign auROC, Causality auROC zero-shot, Causality RF auROC) across individual GTEx tissues (see Methods for specific scorer configurations). For zero-shot tasks, AlphaGenome's default variant scorer is used (Methods; Fig. 4a). For the supervised causality task ('eQTL Causality (RF)'), results use the best-performing feature set ('All scorers combined'; **e**). For Borzoi comparison, the published L2_DIFF scorer across all output tracks is used. (**b**) Performance on eQTL tasks (Coefficient, Sign, Causality) stratified by distance from variant to target gene TSS, compared to Borzoi and a 1/distance baseline for the causality task. (**c**) Performance on eQTL tasks stratified by broad functional classes based on variant location. TG = Target Gene; VTG = Variant target gene. (**d**) Performance on indel eQTLs for the Sign and Coefficient tasks stratified by indel size, comparing AlphaGenome and Borzoi. (**e**) Random forest performance on the eQTL causality task using different feature sets. Horizontal bar plot compares RF performance (auROC from Fig. 4g) when trained using features derived from all scorers combined or individual modality scorers (also comparing AlphaGenome to Borzoi outputs). (**f**) Observed versus predicted eQTL log2 fold change plotted against variant-TSS distance (within 262 kb, ensuring target TSS is within receptive field context). Observed values are allelic log-fold changes from Mohammadi et al.[56], which were used to allow direct comparison with the scale of AlphaGenome's predicted log-fold changes (Methods). (**g**) Sign accuracy as well as the fraction of eQTLs that pass a certain quantile-score cutoff visualized for all possible thresholds and four eQTL distance categories. The quantile score cutoff which leads to an overall (across distance bins) sign accuracy of 90% is highlighted. At this cutoff, sign accuracy is similar across distance categories, although recall is notably lower for distal variants. (**h**) Scatterplot comparing observed versus predicted effect sizes for 8,678 fine-mapped eQTLs filtered for high AlphaGenome prediction scores (>99th percentile of common variants). (**i**) Relationship between AlphaGenome's quantile score threshold (x-axis; Methods) and the resulting performance (y-axis) for eQTL Sign accuracy and Coefficient Spearman R tasks.

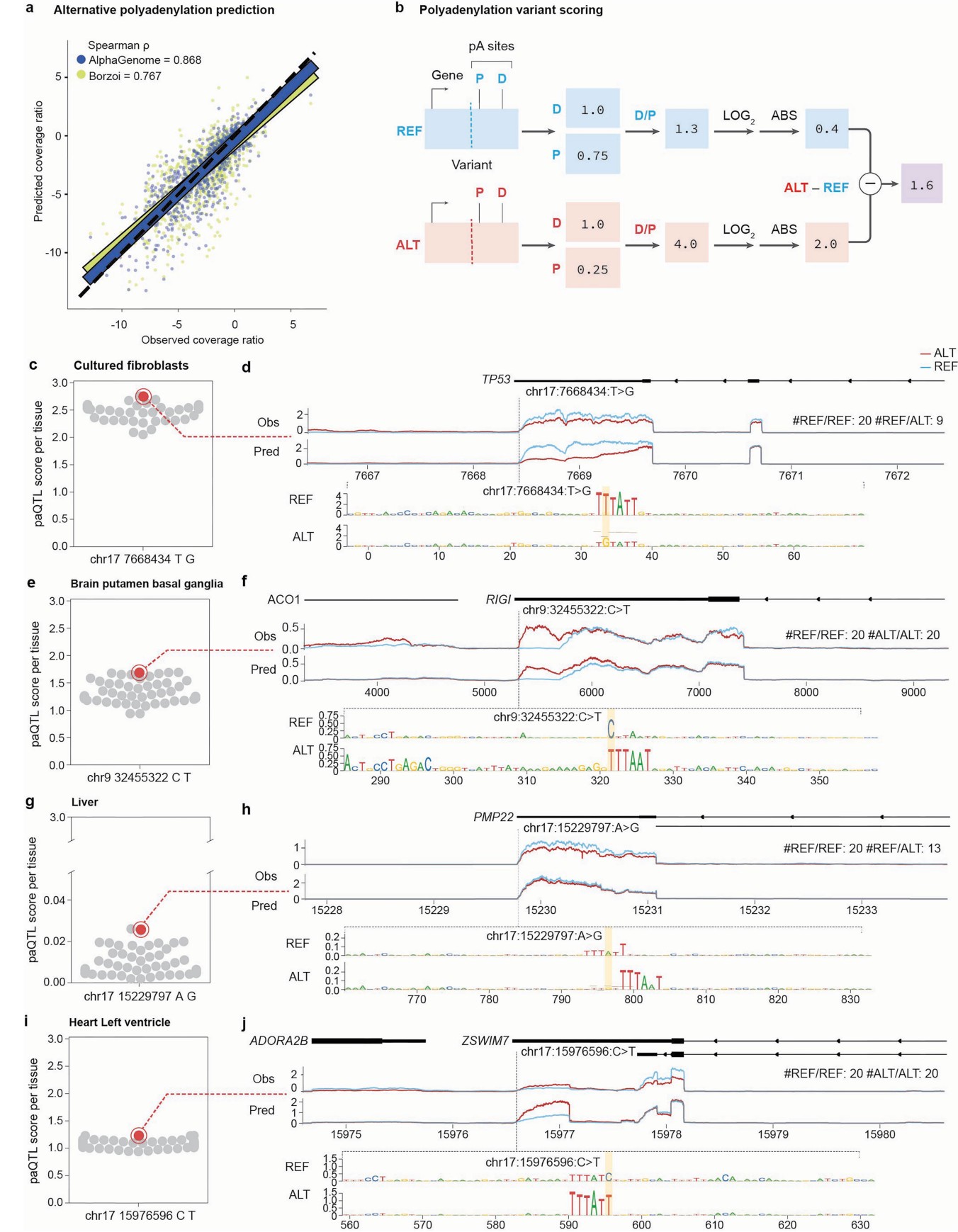

**Extended Data Fig. 6** | See next page for caption.

**Extended Data Fig. 6 | Improved prediction of 3' UTR polyadenylation and paQTLs with AlphaGenome.** (**a**) Polyadenylation predicted vs observed coverage ratio (COVR) for each gene's most distal and proximal PAS for Borzoi and AlphaGenome. (**b**) Polyadenylation variants scoring scheme. (**c**,**d**) Example of variant disrupting a polyadenylation motif in the *TP53* gene. (**c**) Distribution of scores for the indicated variant across GTEx tissues (gray swarmplot) with the highest and lowest scoring tissues and average score highlighted (red dots). (**d**) Observed and predicted RNA-seq for the tissues highlighted in (**c**), and in silico saturation mutagenesis scores (ISM) of the 72 bp flanking the variant.

ISM clearly highlights the relevance of the polyadenylation motif exclusively in the reference background, where the variant does not compromise the motif. (**e**,**f**) As (**c**) and (**d**) but for a variant generating a new polyadenylation motif in the *RIGI* gene. (**g**,**h**) As (**c**) and (**d**) but for a paQTL with a low variant score. The variant disrupts the PAS but a cryptic one emerges nearby with limited effect on gene expression. (**i**,**j**) As (**c**) and (**e**) but for a failure case. AlphaGenome correctly identifies the emergence of a novel PAS but fails to correctly predict the effect on gene expression.

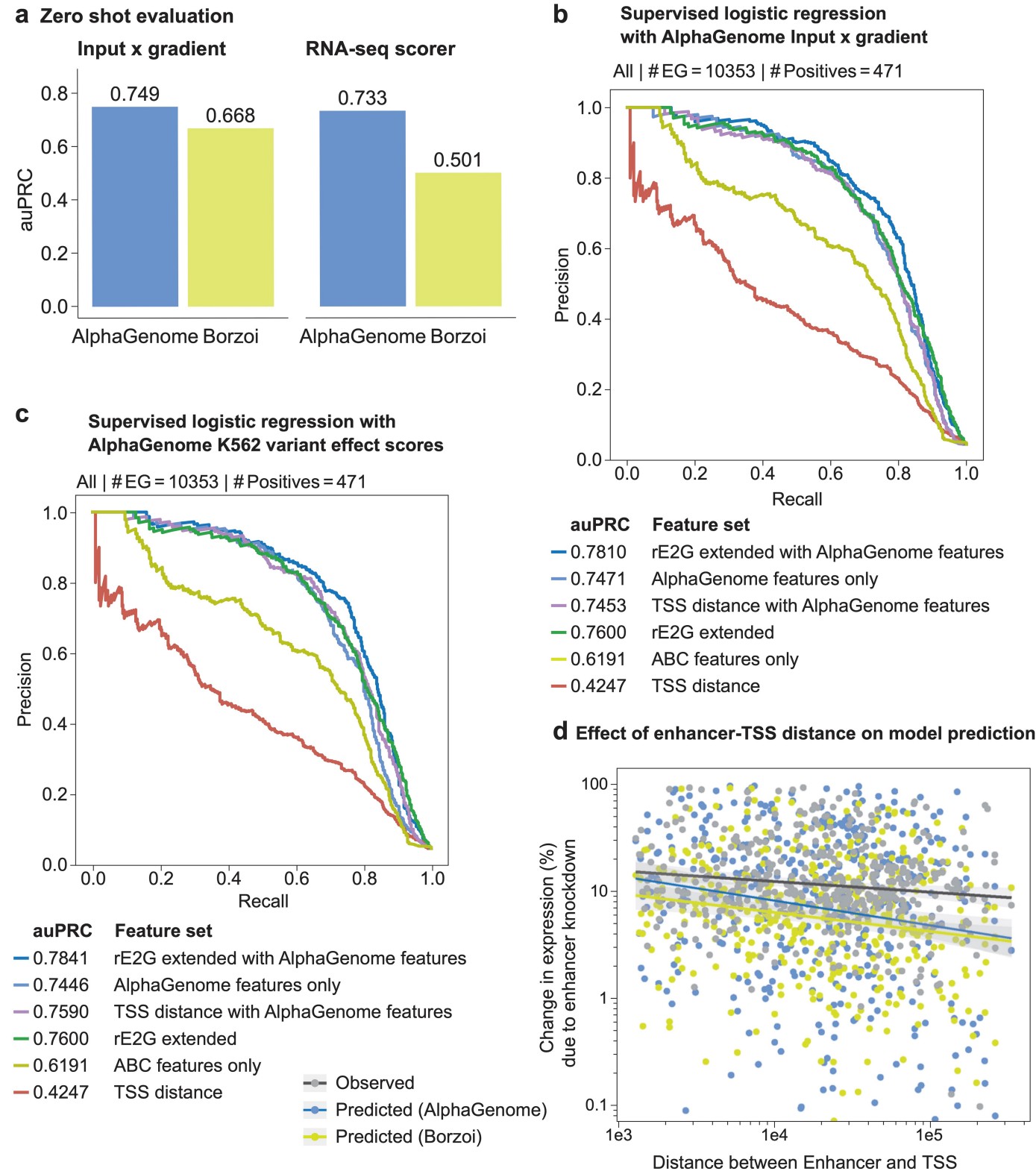

**a** Zero shot evaluation

**b** Supervised logistic regression with AlphaGenome Input x gradient

**c** Supervised logistic regression with AlphaGenome K562 variant effect scores

**d** Effect of enhancer-TSS distance on model predictions

**Extended Data Fig. 7** | See next page for caption.

**Extended Data Fig. 7 | AlphaGenome improves enhancer-gene linking using input gradients and shows enhanced sensitivity to distal enhancers.** (**a**) Zero-shot performance of AlphaGenome and Borzoi on the ENCODE-rE2G benchmark. Bars indicate the area under the precision-recall curve (auPRC) for predicting enhancer-gene links. Two scoring methods derived from each model were evaluated: input gradient scores and RNA-seq variant effect scores. (**b**) Impact of incorporating AlphaGenome's input gradient score as a feature in the ENCODE-rE2G extended logistic regression model, evaluated on the ENCODE-rE2G benchmark. ENCODE-rE2G is a logistic regression model trained to predict enhancer-gene interactions from features[12]. Precision-recall curves are shown, colored by the feature sets used for training the regression model (auPRC values indicated in the legend). Feature sets are: 'rE2G extended with AlphaGenome features': All ENCODE-rE2G extended model features plus a single AlphaGenome's input x gradient score; 'AlphaGenome features only': The AlphaGenome input x gradient score alone; 'TSS distance with AlphaGenome features': AlphaGenome input x gradient score plus the distance to TSS feature; 'rE2G extended': All features from the ENCODE-rE2G extended model[12]; 'TSS distance': Distance to TSS feature from[12]; 'ABC features only': Subset of 'rE2g extended', with only features related to the Activity-By-Contact (ABC) model[12].

(**c**) Precision-recall curves for the ENCODE-rE2G benchmark, similar to panel (**b**), evaluating the ENCODE-rE2G extended regression model with different feature sets. Area under the precision-recall curve (auPRC) values for the different feature sets are indicated in the legend. In this configuration, 'AlphaGenome features' consist of a more comprehensive set of K562 cell line-specific variant effect scores. These include Allele-Specific Activity Scores (AAS) and variant effect scores calculated as the difference between alternate (ALT) and reference (REF) allele predictions (ALT-REF Diff scores). These scores were derived from AlphaGenome for the following genomic assays: RNA-seq of the target gene, ChIP-TF EP300, ChIP-Histone H3K27ac, CAGE, PRO-cap, H1-ESC contact maps. (**d**) Relationship between enhancer perturbation effects (ENCODE-rE2G dataset[12]) and enhancer-promoter distance. The scatter plot shows experimentally observed percentage changes in gene expression upon enhancer knockout (grey points and trend line) versus the genomic distance between the enhancer and the target gene's Transcription Start Site (TSS). Overlaid are trend lines for AlphaGenome's (AG, dark blue) and Borzoi's (green) predictions of these expression changes, derived from their respective model input gradient scores. Each point corresponds to a validated enhancer-gene pair. Error bars (grey) represent 95% confidence intervals from the linear regression.

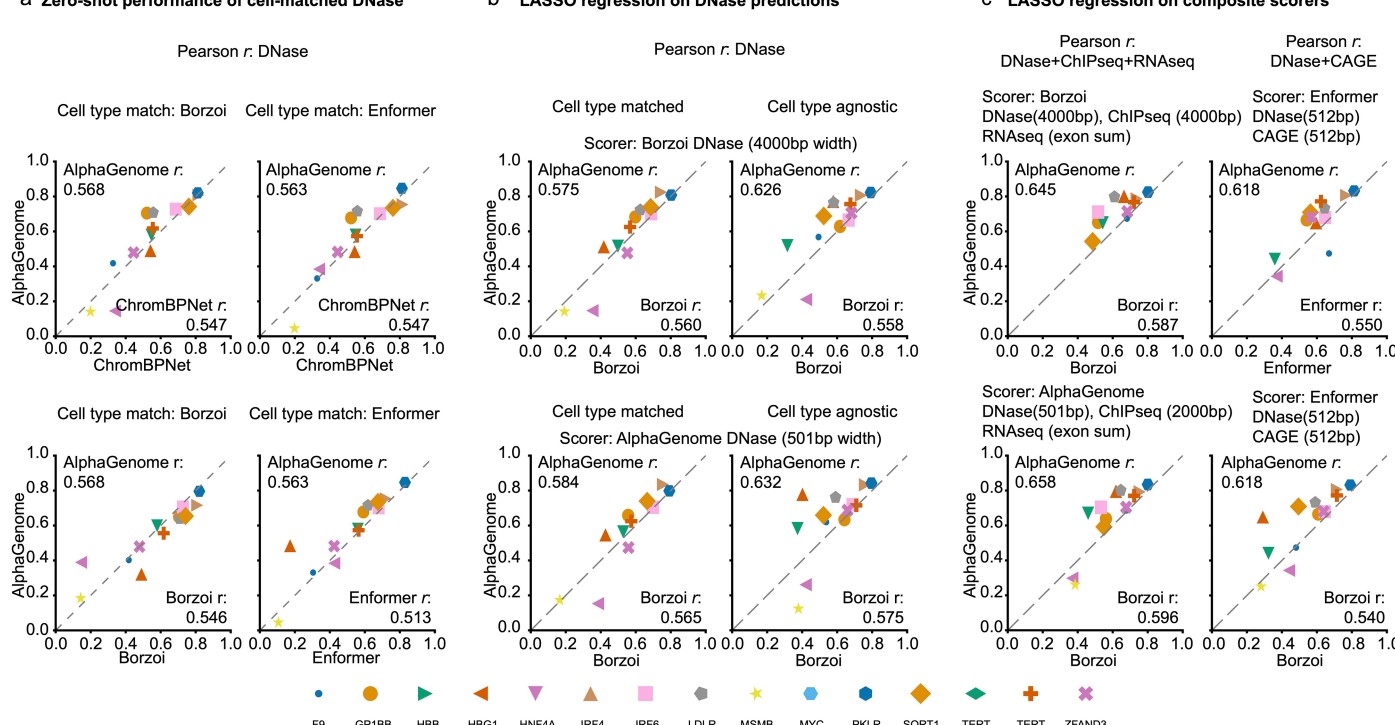

**Extended Data Fig. 8 | AlphaGenome performance on the CAGI5 MPRA challenge.** Comparison of models predicting the effects of regulatory variants on gene expression using CAGI5 MPRA data. (**a**) Pairwise comparisons of zero-shot, cell-type matched performance using raw DNase model outputs. Scatter plots show per-gene or per-context Pearson correlations. AlphaGenome shows comparable or slightly improved performance compared to Enformer, ChromBPNet, and Borzoi Ensemble based on overall correlations across compared loci (details in plot annotations; cell-type matching strategy adapted from Enformer/Borzoi papers, see Methods). Note that ChromBPNet only reports performance for TERT-HEK293T, thus TERT-GBM is excluded in the AlphaGenome vs ChromBPNet comparison. (**b**) Pairwise comparisons of

cell-type matched (see **a**) and cell type agnostic performance using LASSO regression on spatially summed DNase features (Methods). Comparison of the original Borzoi scorer (upper row) and AlphaGenome recommended scoring strategy (bottom row). AlphaGenome outperforms Borzoi in all settings and shows that cell type matching is not required when applying LASSO regression. (**c**) Pairwise comparisons of cell-type agnostic performance using LASSO regression with different input features: left column shows the composite scorer described in Borzoi (DNase + Histone ChIP-seq + RNA), right column shows the composite scorer described in Enformer (DNase + CAGE). AlphaGenome generally outperforms Enformer and Borzoi for all input feature settings.

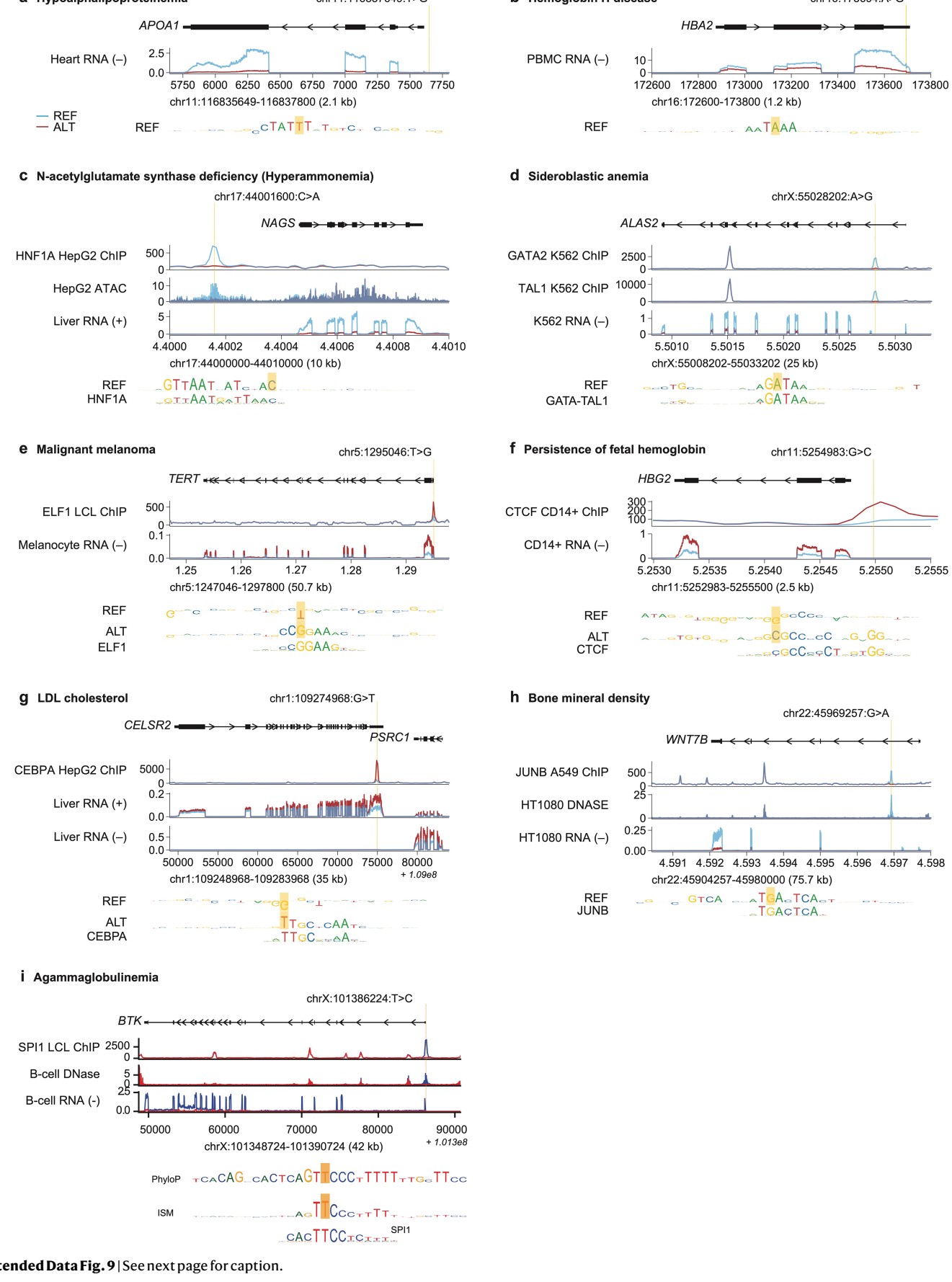

**Extended Data Fig. 9** | See next page for caption.

**Extended Data Fig. 9 | Interpreting trait-affecting variants using AlphaGenome.** (**a**) AlphaGenome predictions for the variant chr11:116837649:T>G, previously reported to result in underexpression of APOA1 and Hypoalphalipoproteinemia[57]. The variant strongly reduces the predicted expression of the *APOA1* transcript in heart tissue. ISM of the reference sequence suggests that the variant disrupts a TATA-like motif. (**b**) AlphaGenome predictions for the variant chr16:173692:A>G, previously reported to cause Hemoglobin H disease[58]. The model predicts that the variant strongly reduces the predicted expression of the *HBA2* transcript. ISM of the reference sequence suggests that the variant disrupts the polyadenylation hexamer. (**c**) AlphaGenome predictions for the variant chr17:44001600:C>A, previously reported to result in reduced binding of HNF1, underexpression of NAGS and ultimately N-acetylglutamate Synthase Deficiency[59]. AlphaGenome predicts reduced binding of HNF1A, local DNA accessibility, as well as a strong reduction in expression of the NAGS transcript in liver and related cell types. ISM of the reference sequence suggests that the variant disrupts a HNF1 motif. (**d**) AlphaGenome predictions for the variant chrX:55028202:A>G, previously reported to result in reduced GATA binding and reduced expression of *ALAS2*, thereby possibly causing Sideroblastic Anemia[60]. The model predicts reduced binding of GATA factors (GATA2 shown), as well as TAL1, and drastically reduced expression of the *ALAS2* transcript in K562. ISM of the reference sequence suggests that the variant disrupts a GATA-TAL composite motif. (**e**) AlphaGenome predictions for the variant chr5:1295046:T>G, previously reported to cause TERT overexpression associated with malignant melanoma[61]. AlphaGenome predicts increased binding of ETS factors (ELF1 shown) and an increase in *TERT* expression in melanocytes. Comparing the ISM around the reference and alternative alleles suggests that the variant creates an ETS factor binding motif. The phyloP[62] score for this position is −3.29, consistent with the fact that gain-of-function variants can occur in fast-evolving parts of the genome. (**f**) AlphaGenome predictions for the variant chr11:5254983:G>C, previously implicated in hereditary persistence of fetal hemoglobin[63]. The model predicts increased CTCF binding, as well as increased *HBG2* expression (shown for CD14+ monocytes). Comparing the ISM around the reference and alternative alleles suggests that the variant creates a CTCF binding motif. The phyloP score for this position is −0.67, indicating that the variant position is under relatively neutral selection (given the low magnitude of the score). (**g**) AlphaGenome predictions for the variant rs12740374, previously found to be associated with low-density lipoprotein cholesterol levels[64]. Additionally, the variant is a plasma pQTL for CELSR2, as well as a GTEx eQTL for *CELSR2* and *PSRC1*[17,65]. Consistent with the observed molecular QTL effects, AlphaGenome predicts increased expression of both of these genes in liver tissue. Moreover the model predicts increased CEBP binding on the alternative allele. Comparing the ISM around the reference and alternative alleles suggests that the variant creates a CEBP binding motif. The phyloP score for this position is −0.20, indicating that the variant position is under relatively neutral selection (given the low magnitude of the score). (**h**) AlphaGenome predictions for the variant rs570639864, previously found to be associated with reduced bone mineral density[66]. AlphaGenome predicts decreased binding of JUN/FOS factors (JUNB shown), as well as a strong reduction in expression of *WNT7B*, a gene implicated in bone formation[67]. ISM of the reference sequence suggests that the variant disrupts a JUN/FOS motif. (**i**) AlphaGenome predictions for the variant chrX:101386224:T>C, previously reported to result in a lack of functional BTK and causing X-linked Agammaglobulinemia (OMIM #300755). The model predicts reduced binding of SPI1 at the *BTK* promoter, diminished local DNA accessibility, as well as a reduction in expression of the *BTK* transcript in B & lymphoblastoid cells. ISM of the reference sequence suggests that the variant disrupts a SPI-like motif (SPI1.H13.0.PB shown). Also shown is the 100-vertebrate PhyloP conservation track, which shows high conservation around the variant.

# Reporting Summary

## Statistics

For all statistical analyses, confirm that the following items are present in the figure legend, table legend, main text, or Methods section.

| n/a | Confirmed | |
|---|---|---|
| ☐ | ☒ | The exact sample size (*n*) for each experimental group/condition, given as a discrete number and unit of measurement |
| ☒ | ☐ | A statement on whether measurements were taken from distinct samples or whether the same sample was measured repeatedly |
| ☐ | ☒ | The statistical test(s) used AND whether they are one- or two-sided<br>*Only common tests should be described solely by name; describe more complex techniques in the Methods section.* |
| ☒ | ☐ | A description of all covariates tested |
| ☒ | ☐ | A description of any assumptions or corrections, such as tests of normality and adjustment for multiple comparisons |
| ☐ | ☒ | A full description of the statistical parameters including central tendency (e.g. means) or other basic estimates (e.g. regression coefficient) AND variation (e.g. standard deviation) or associated estimates of uncertainty (e.g. confidence intervals) |
| ☐ | ☒ | For null hypothesis testing, the test statistic (e.g. *F*, *t*, *r*) with confidence intervals, effect sizes, degrees of freedom and *P* value noted<br>*Give P values as exact values whenever suitable.* |
| ☒ | ☐ | For Bayesian analysis, information on the choice of priors and Markov chain Monte Carlo settings |
| ☒ | ☐ | For hierarchical and complex designs, identification of the appropriate level for tests and full reporting of outcomes |
| ☐ | ☒ | Estimates of effect sizes (e.g. Cohen's *d*, Pearson's *r*), indicating how they were calculated |

*Our web collection on statistics for biologists contains articles on many of the points above.*

## Software and code

Policy information about availability of computer code

| Data collection | No software was used for data collection. This study utilized pre-existing, publicly available datasets. |
|---|---|
| Data analysis | Data analysis used Python v.3.11.8 (https://www.python.org/), NumPy v2.2.5 (https://github.com/numpy/numpy), SciPy v.1.14.1 (https://www.scipy.org/), seaborn v.0.12.2 (https://github.com/mwaskom/seaborn), Matplotlib v.3.9.1 (https://github.com/matplotlib/matplotlib), Pandas v.2.2.3 (https://github.com/pandas-dev/pandas), anndata v0.11.4 (https://github.com/scverse/anndata), Scikit-learn v1.6.1, (https://github.com/scikit-learn/scikit-learn), STAR (version 2.7.11b), Samtools (version 1.21), FRASER 2.085, DROP pipeline (version 1.3.3), and Colab (https://research.google.com/colaboratory; release 2025-06-16).<br><br>AlphaGenome is available for non-commercial use via an online API at http://deepmind.google.com/science/alphagenome, with an accompanying Python software development kit (SDK) provided to interact with the model. The model source code and weights will also be provided upon final publication. We also provide a genome interpretation suite to facilitate the exploration and interpretation of AlphaGenome. In addition to the online API and interpretation suite, we also provide model source code, weights, variant scoring implementations and a selection of variant evaluation datasets and predictions. These can all be found at https://github.com/google-deepmind/alphagenome |

For manuscripts utilizing custom algorithms or software that are central to the research but not yet described in published literature, software must be made available to editors and reviewers. We strongly encourage code deposition in a community repository (e.g. GitHub). See the Nature Portfolio guidelines for submitting code & software for further information.

## Data

Policy information about availability of data

All manuscripts must include a data availability statement. This statement should provide the following information, where applicable:

- Accession codes, unique identifiers, or web links for publicly available datasets
- A description of any restrictions on data availability
- For clinical datasets or third party data, please ensure that the statement adheres to our policy

All primary experimental datasets utilized for the training and evaluation of AlphaGenome in this study were obtained from publicly accessible sources. A comprehensive manifest detailing these data sources – including specific repositories (e.g., ENCODE portal, GTEx portal, 4D Nucleome portal, ClinVar, gnomAD), individual accession numbers, relevant version information, and direct URLs where applicable – is provided in Supplementary Table 2. This study did not generate new primary experimental data requiring deposition.

## Research involving human participants, their data, or biological material

Policy information about studies with human participants or human data. See also policy information about sex, gender (identity/presentation), and sexual orientation and race, ethnicity and racism.

| | |
|---|---|
| Reporting on sex and gender | No human population data was collected as part of this study. Sex is included as clinical covariates for the eQTL mapping. We also report sex-specific track data. |
| Reporting on race, ethnicity, or other socially relevant groupings | There is a description of genetic ancestry in QTL tasks - Ancestries mentioned in the text are European, African, and Yoruba. |
| Population characteristics | No human population data was collected as part of this study. |
| Recruitment | No human data was collected as part of this study. |
| Ethics oversight | No human data was collected as part of this study. |

Note that full information on the approval of the study protocol must also be provided in the manuscript.

# Field-specific reporting

Please select the one below that is the best fit for your research. If you are not sure, read the appropriate sections before making your selection.

☒ Life sciences          ☐ Behavioural & social sciences          ☐ Ecological, evolutionary & environmental sciences

For a reference copy of the document with all sections, see nature.com/documents/nr-reporting-summary-flat.pdf

# Life sciences study design

All studies must disclose on these points even when the disclosure is negative.

| | |
|---|---|
| Sample size | All available data were used for each benchmark and evaluation. No subsampling was performed |
| Data exclusions | Data were binned into splits during model development. No data were excluded from benchmarks. |
| Replication | Code and methods were carefully checked for completeness and replicability. |
| Randomization | Random assignment of subjects was not applicable as this is a computational study utilizing pre-existing, publicly available genomic datasets. However, randomness was employed computationally via random seeds for model initialization and data augmentation. To ensure robust performance estimation and prevent data leakage, data was partitioned using pre-defined genomic interval splits (following established benchmarks) for model training and evaluation. |
| Blinding | Blinding was not necessary as the evaluation relies on automated, quantitative benchmarking rather than subjective scoring. Bias was mitigated by strictly segregating training, validation, and testing data to ensure models were evaluated solely on data they had never seen. |

# Reporting for specific materials, systems and methods

We require information from authors about some types of materials, experimental systems and methods used in many studies. Here, indicate whether each material, system or method listed is relevant to your study. If you are not sure if a list item applies to your research, read the appropriate section before selecting a response.

## Materials & experimental systems

| n/a | Involved in the study |
|-----|----------------------|
| ☒ | ☐ Antibodies |
| ☒ | ☐ Eukaryotic cell lines |
| ☒ | ☐ Palaeontology and archaeology |
| ☒ | ☐ Animals and other organisms |
| ☒ | ☐ Clinical data |
| ☒ | ☐ Dual use research of concern |
| ☒ | ☐ Plants |

## Methods

| n/a | Involved in the study |
|-----|----------------------|
| ☒ | ☐ ChIP-seq |
| ☒ | ☐ Flow cytometry |
| ☒ | ☐ MRI-based neuroimaging |

## Plants

| Seed stocks | None |
|-------------|------|

| Novel plant genotypes | None |
|-----------------------|------|

| Authentication | None |
|----------------|------|

