## [Peer Review File · Nature]

AlphaGenome: improved regulatory variant effect prediction with a unified model

Corresponding Author: Dr Žiga Avsec

Version 0:

Reviewer comments:

Referee #1

(Remarks to the Author)

The authors introduce AlphaGenome, a unified deep learning framework that predicts a multimodal array of functional genomic tracks and, by way of zero-shot assessment of the difference between reference and alternate, variant effects from megabase-scale DNA sequence input. The model combines nucleotide-level resolution, multimodal outputs (including expression, splicing, chromatin accessibility, TF binding, and 3D genome architecture), and generalization across cell types within a single encoder-decoder architecture. It is trained through pretraining and distillation phases, leveraging both experimental data and ensemble teacher predictions. Across a comprehensive set of benchmarks, AlphaGenome consistently matches or exceeds state-of-the-art performance on 24 out of 26 variant effect prediction tasks and 22 out of 24 track prediction benchmarks.

Of particular interest is the model's novel capacity to predict splice junctions, complementing its already strong splice site and usage predictions. AlphaGenome's multimodal scope enables it to engage with a wide range of regulatory mechanisms—including enhancer-promoter interactions and alternative polyadenylation. The manuscript also illustrates the model's capacity to recapitulate known oncogenic effects at the TAL1 locus and shows modest enrichment for trait-altering variants genome-wide. While the model does not solve all problems (nor does it purport to), the work represents an important technical advance with potential for improving our understanding of noncoding variation and its functional consequences. Of note, the authors' commitment to full data and model release is laudable. A few major comments would improve the model's interpretation further, and some minor ones the manuscript's clarity.

Major Comments

Impressively, this model achieves, for the first time, a combination of long-range context, single-base precision, and multimodality, while also improving on most benchmarks. However, the gains reported by the model, while consistent, are sometimes modest in magnitude, understandably as these are major challenges in biology and translational applications. To show how this model could be used in a translational setting, it could be valuable to include compelling case studies—such as refined interpretation of VUS (variants of uncertain significance) or real-world applications in diagnosis or personalized medicine—that would have been difficult to achieve with existing models. The former could be achieved through temporal analysis of ClinVar—do variants that change interpretation from VUS to P/LP have higher AG scores than VUS to B/LB, and does that outperform the current models? Care will need to be taken around variant effects (e.g. removing missense effects), but the authors are doing this already, which I applaud. The latter would be most powerful with a clinical dataset, but in the absence of that, analysis of de novo variants, as is often done in the field, could provide this information: honing to non-coding de novo variants would be an extremely powerful tool.

There is some attempt at this in Fig. 7f, but this is very hard to read: this reviewer is trying to translate it into an ROC curve but it feels tricky. Could some sort of rate ratio or enrichment be used to show this, and compare to a previous method?

Another possibility would be to highlight cases where AlphaGenome enabled the confident interpretation of regulatory variants (e.g., QTLs for splicing, expression, or chromatin accessibility) that were previously ambiguous or lacked support. Are there examples in the current results where AlphaGenome helped reclassify a variant or uncover new regulatory mechanisms?

The authors write: "This makes AlphaGenome a useful complement to conservation-based measures of deleteriousness which are agnostic to a variant's mechanism of action." It would be helpful to show a concrete example where AlphaGenome not only predicts a strong deleterious effect that matches conservation-based metrics, but also suggests a novel mechanism of action. Such an illustration would demonstrate the model's potential to guide novel biological insights.

Minor Comments and Suggestions

In general, many of the statistics shown are ROC or correlations, which is sensible to compare to previous methods. But in many cases, these ROCs are too low to be clinically useful (again, understandably as these are hard tasks). As in the major comment above, are there any cases where more interpretable metrics like rate ratios or enrichments, could be used to more directly highlight the utility in various analytic spaces?

The tools for genome track prediction are appreciated. To enhance usability for the community, would it be possible to release the full predicted genome tracks, particularly for key modalities like splicing? Ideally also with a mechanism to input variant data to see how things change. A browser framework showing this would be incredible but probably out of scope for this manuscript.

Could AlphaGenome be applied to study the spatial relationship between regulatory elements (e.g., TF binding sites) and their target genes?

Can AlphaGenome detect features associated with microRNA loci or their regulatory effects? Even if not explicitly trained for this, it would be helpful to clarify whether the model implicitly learns patterns characteristic of such regions.

A comparison with constraint (e.g., Gnocchi) and conservation scores could offer insights. Is there general overlap with these, and are there regions where AG and conservation metrics diverge?

For the A→C mutation example, how specific is the predicted effect? Would neighboring mutations or different mutations at the same position (e.g., A→T or A→G) produce similar predictions? A short discussion on the specificity of mutational impact would strengthen the mechanistic interpretation.

Across modalities, how do the correlations between predicted and observed tracks compare to cross-species correlations (e.g., human vs. mouse)? Also, was training on mouse tracks beneficial for human track prediction? An ablation study might help clarify this.

In combining AlphaGenome and Borzoi features, is there a measurable gain in predictive power? This could help clarify whether Borzoi retains complementary signals not fully captured by AlphaGenome.

The TAL1 example is compelling. Are these regulatory changes predicted to be tissue-specific? For instance, how does AlphaGenome score the same variants in other tissues beyond CD34+ CMP?

As a very minor note, the authors sometimes mention that performance is "significantly" better but fail to report a p-value, and instead an increased AUC. Could these be removed or better yet, replaced with significance tests?

(Remarks on code availability)

Referee #2

(Remarks to the Author)

This manuscript introduces the AlphaGenome model, a state-of-the-art functional genomic predictor that can predict 11 genomic modalities from the DNA sequence alone. This model can be understood as a very significant improvement over Borzoi (Linder et al., Nature Genet., 2025), and it extends beyond this previous general purpose model in many ways. Broadly speaking it extends the sequence window up to 1 Mb and the resolution down to 1 bp for many but not all genomic tracks. There are many expansions to the architecture and expanded capabilities to predict explicitly additional genomic tracks, taking advantage of additional training datasets modalities. The manuscript is very dense and the technical advances are many. It is not always straightforward to understand the differences between these two models and their respective training datasets. Examples of differences in training data include the inclusion of the PRO-cap start-site assays, 3D chromatin contact maps, explicit splice-junction quantification, and an extended set of GTEx datasets. There are also major advances in the architecture to enable representation at different resolutions and model long-range interactions. Finally, there is also an additional technical improvement in inference time efficiency by training a distilled "student" model that learns from an ensemble of all-folds teachers.

Across almost all modalities tested, AlphaGenome can outperform both the general purpose Borzoi model as well as task-specific models for some of the modalities. For many of the tasks it should be clear that the improvements are modest, but it nevertheless remains an impressive advance given that the task-specific models can often be trained more specifically and

can often outperform more general models.

The evaluation of the model seems convincing with minor concerns about the use of the distilled student model in some evaluations. The authors performed extensive ablation studies, although it could have been interesting to better understand or at least provide and discuss some examples how some modalities benefit from the multi-model joint training.

The model and training is very extensively described and the authors at least claim that the model will be available to use via an API and the code and model weights will be available upon publication. This is critical and should be a requirement for publication.

Overall, the method described represents a new state-of-the-art functional genomics predictor that can enable many future applications (e.g. human variant effect prediction, association of predicted molecular features with phenotypes in large cohorts). This will be of relevance and interest to many.

I only have some minor concerns for clarifications or discussion:

Given the multi-modal data, was there no per-modality gradient weighting or scheduling strategy used? Are there any differences in weights in the loss-functions for the different modalities?

Regarding the training of the student model from the ensemble of teacher models, it was clear that this procedure is very relevant for improving inference time. However, it generally seemed to perform worse in testing. In the ablation testing section, Fig. 7c, why is there no data for the results obtained with the 64 aggregated models?

For augmentation, it isn't stated if there is any specific scheduling of the different introduced genomic alterations made throughout training. Did the authors try different parameters for the introduction of the variation? It would be interesting to have a better sense from the authors of how important this augmentation is for model performance. Presumably, this is part of the reason why there is an improvement between the pretrained single model and the distilled version in Fig. 7c.

The authors do not mention any confidence estimates. Is the distilled student model also learning an estimate of uncertainty from the ensemble? Would this measurement of uncertainty, either directly measured by the ensemble or estimated by the student, be useful in downstream prediction tasks?

For the discussion, what are some current limitations in terms of existing data that should be the focus of future large scale experimental studies? Are there modalities (e.g. chromatin accessibility) where data is not the major bottleneck in modelling? Given the current methodologies, are there specific data-types and modalities that could result in strong advances given additional datasets?

(Remarks on code availability)

The code and API are not yet available to review.

Referee #3

(Remarks to the Author)

The authors present AlphaGenome, a unified DNA sequence-to-function deep learning model that predicts diverse modalities, including gene expression, transcription initiation, and others. Taking 1 megabase of DNA sequence as input and being trained on the human and mouse genomes, AlphaGenome matches or exceeds current state-of-the-art models on genome track and variant effect prediction.

Employing a U-Net-inspired architecture and a two-stage training process (pre-training and distillation), AlphaGenome allows unprecedentedly large input sequences while retaining single-base-pair resolution and simultaneously predicting 5,930 human or 1,128 mouse genome tracks across 11 modalities. The work is clearly a very impressive engineering effort, and the manuscript is extremely clearly written. Of particular note is how complete the methods descriptions are, both in terms of the modeling details and the processing of each data type, and how open the authors are about giving credit to others. On the other hand, there is not much innovation: The model itself and the training loss is very similar to Borzoi (as the authors mention) with an additional loss component from Decima. The work largely involves demonstrating that their unified model outperforms (almost) all of the individual SOTA methods. The use of the model for discovering something new is limited, and no experiments were performed to validate any novel hypotheses. Thus, the advance on novel insights or any individual prediction task appear more incremental, and it is largely the scope and comprehensiveness of AlphaGenome that stands out.

Given the rather incremental improvement of the individual prediction tasks, AlphaGenome will likely not be the AlphaFold for genomics. However, I appreciate very much the amount of engineering effort the authors put in at every stage, and that they credit everyone with where the ideas came from. Moreover, if the authors make the full code accessible to the compbio community and build a powerful and user-friendly interface for the general scientific and medical community for people to query their own variants, AlphaGenome will have the strong impact that warrants a high-profile publication.

Major Comments

The distillation procedure seems conceptually interesting, but it is unclear what the benefit of including the variants (point

mutations and structural variants) was. Specifically, using a teacher model to make predictions for these models does not add new information for the student model compared with training on the original data. Rather, the main benefit seems to be that of classic distillation: faster inference than a larger (ensemble) of models at a small cost to performance. Therefore raising the question whether training using the variants is actually helpful?

The performance is partly evaluated on absolute terms, yet partly only on relative terms, compared to existing models. Fig. 1 for example shows relative improvements, which are informative from a technical point of view (capturing the improvements over previous models) but don't communicate the performance gap to a perfect predictor. Especially for prediction tasks for which the state-of-the-art models perform more poorly (see e.g. Tang, 2023; Sasse, 2023), this additional information seems important. Please add more readily interpretable performance metrics.

Additional comments

I don't understand why quantile normalization and mean-centering the expression values (Fig. 2d, Methods p. 45) brings the PCCs within each track down from 0.82 to 0.52 (compare left to middle). Which feature of the data is distorted to cause this? What does it mean that the models are better in predicting raw log-transformed expression values than their quantile-normalized and/or mean-centered counterparts?

What is the prominent peak in RNA-seq coverage after the APOL4 gene (blue) that is decreased for the ALT allele? Is this part of APOL4, e.g. due to alternative polyadenylation, or a separate downstream gene?

When assessing magnitude of eQTL effects, why is the Spearman CC used rather than the PCC? The agreement between predicted and observed effect sizes looks rather poor (Fig. 4d); please explain and/or comment on this in the main text.

On p. 11 (Fig. 4g,h), the evaluation of the direction of effect seems unnecessarily complicated. Given the binary nature of the data, up vs. down, and the baseline of 50% for balanced datasets, what is the fraction of correctly predicted effects? In Fig. 4h, the legend mentions "with a predicted direction of effect", not indicating if the predicted direction is correct. Please clarify and simplify the analysis and presentation.

On p. 15, the authors might consider revising the introduction to MPRA data, removing the reference to "local chromatin accessibility", which is not tested by plasmid-based MPRA as used in ref. 34.

It is unclear when the authors are referring to the distilled version or some of the other models. Based on the initial description, I assume that sections after L234 would be from the distilled model but some of the figure captions refer explicitly to a distilled model and others do not, e.g. Fig. 4 and specifically Fig. 4j. Please clarify.

It was somewhat unclear why the model made predictions for TF binding at 128-bp resolution, when TFs have been shown by the first author in a previous paper to have important basepair resolution properties. The ablation section mentions that increasing resolution is helpful, and predictions are made for other assays at basepair resolution. Potentially, there are preprocessing issues related to fragment length and single-end vs paired-end differences that may be challenging to overcome, but it would be worth mentioning explicitly why this choice was made.

(Remarks on code availability)

Referee #4

(Remarks to the Author)

I co-reviewed this manuscript with one of the reviewers who provided the listed reports.

(Remarks on code availability)

Version 1:

Reviewer comments:

Referee #1

(Remarks to the Author)

The authors have satisfied all my concerns. My only minor comment would be regarding Fig. 6f - this is still quite hard to read, partially due to the flipped axes (dependent variable on the x rather than y) and partially because it's hard to cross reference exactly what the enrichment corresponds to - is it the red? blue? orange? In addition, a recent preprint from Finucane et al. describes a system for assessing variant effect predictors, and that framework might be useful here to show the impact on human genetics applications.

(Remarks on code availability)

Referee #2

(Remarks to the Author)

The authors have address my previous concerns. I appreciate as well several of the responses to the other reviewers and the effort to correct issues that were found since submission. I also recognize the effort to make their work publicly accessible and reusable.

(Remarks on code availability)

Referee #3

(Remarks to the Author)

The authors have provided substantial revisions and additions to their original manuscript to address the reviewers' concerns and suggestions. In addition, they have communicated to the editor that their code and an API are now available, and that the model source code and weights will be provided upon publication. The revision and the availability of code, API and weights satisfy my original concerns, and I congratulate the authors to their comprehensive work.

(Remarks on code availability)

Referee #4

(Remarks to the Author)

I co-reviewed this manuscript with one of the reviewers who provided the listed reports.

(Remarks on code availability)

Point-by-Point Response to Nature Reviewers' Comments

AlphaGenome: advancing regulatory variant effect prediction with a unified DNA sequence model

Žiga Avsec^{1*} , Natasha Latysheva^{1*}, Jun Cheng^{1*}, Guido Novati^{1*}, Kyle R. Taylor^{1*}, Tom Ward^{1*}, Clare Bycroft^{1*}, Lauren Nicolaisen^{1*}, Eirini Arvaniti^{1*}, Joshua Pan^{1*}, Raina Thomas¹, Vincent Dutordoir¹, Matteo Perino¹, Soham De¹, Alexander Karollus¹, Adam Gayoso¹, Toby Sargeant¹, Anne Mottram¹, Lai Hong Wong¹, Pavol Drotár¹, Adam Kosiorek¹, Andrew Senior¹, Richard Tanburn¹, Taylor Applebaum¹, Souradeep Basu¹, Demis Hassabis¹, Pushmeet Kohli¹ 
Overview

We thank the editor and reviewers for their insightful and thorough comments. In response to their valuable feedback, we have substantially revised the manuscript. Key improvements include new case studies to better demonstrate the model's translational utility, more interpretable performance metrics to improve the clarity of our results, and further exposition on key methodological choices, such as our variant-based distillation strategy.

We believe these changes, reflected in 10 revised figures, 3 new supplementary figures, and 1 new supplementary note, strengthen the manuscript's coverage and impact. To aid the review process, we have also prepared a separate document showing 'Before and After' versions of all revised figures (see “**Before and After Revisions Figure Comparisons**”).

We are grateful for the reviewers' expertise and constructive guidance. Our detailed point-by-point responses follow.

In places where we have changed the text, we have included these changes directly in the response document using the following format:

“...Existing, unchanged manuscript text in italics. Newly introduced text to address the reviewer concerns, in underline. Additional existing, unchanged manuscript text in italics.”

Referees' comments:

Referee #1:

The authors introduce AlphaGenome, a unified deep learning framework that predicts a multimodal array of functional genomic tracks and, by way of zero-shot assessment of the difference between reference and alternate, variant effects from megabase-scale DNA sequence input. The model combines nucleotide-level resolution, multimodal outputs (including expression, splicing, chromatin accessibility, TF binding, and 3D genome architecture), and generalization across cell types within a single encoder-decoder architecture. It is trained through pretraining and distillation phases, leveraging both experimental data and ensemble teacher predictions. Across a comprehensive set of benchmarks, AlphaGenome consistently matches or exceeds state-of-the-art performance on 24 out of 26 variant effect prediction tasks and 22 out of 24 track prediction benchmarks.

Of particular interest is the model's novel capacity to predict splice junctions, complementing its already strong splice site and usage predictions. AlphaGenome's multimodal scope enables it to engage with a wide range of regulatory mechanisms—including enhancer-promoter interactions and alternative polyadenylation. The manuscript also illustrates the model's capacity to recapitulate known oncogenic effects at the TAL1 locus and shows modest enrichment for trait-altering variants genome-wide. While the model does not solve all problems (nor does it purport to), the work represents an important technical advance with potential for improving our understanding of noncoding variation and its functional consequences. Of note, the authors' commitment to full data and model release is laudable. **A few major comments would improve the model's interpretation further**, and some minor ones the manuscript's clarity.

Major Comments

Impressively, this model achieves, for the first time, a combination of long-range context, single-base precision, and multimodality, while also improving on most benchmarks. However, the gains reported by the model, while consistent, are sometimes modest in magnitude, understandably as these are major challenges in biology and translational applications. To show how this model could be used in a translational setting, it could be valuable to 1) **include compelling case studies**—such as **refined interpretation of VUS** (variants of uncertain significance) or 2) **real-world applications in diagnosis or personalized medicine**—that would have been difficult to achieve with existing models. The former could be achieved through temporal **analysis of ClinVar**—do variants that change interpretation from VUS to P/LP have higher AG scores than VUS to B/LB, and does that outperform the current models? Care will need to be taken around variant effects (e.g. removing missense effects), but the authors are doing this already, which I applaud. The latter would be most powerful with a clinical dataset, but in the absence of that, **analysis of de novo variants**, as is often done in the field, could provide this information: honing to non-coding de novo variants would be an extremely powerful tool.

There is some attempt at this in Fig. 7f, but this is very hard to read: this reviewer is trying to translate it into an ROC curve but it feels tricky. Could some sort of rate ratio or enrichment be used to show this, and compare to a previous method?

Another possibility would be to highlight cases where AlphaGenome enabled the confident interpretation of regulatory variants (e.g., QTLs for splicing, expression, or chromatin accessibility) that were previously ambiguous or lacked support. Are there examples in the current results where AlphaGenome helped reclassify a variant or uncover new regulatory mechanisms?

Non-coding variant interpretation is an important potential application of AlphaGenome, and providing 1) compelling case studies and 2) real-world applications would be useful for assessing the model in this context.

To directly address both of these points, we have written an additional **Supplemental Note: Revisiting clinically annotated non-coding variants with ACMG guidelines**, grounded in the recently proposed non-coding variant interpretation guidelines by ACMG (Ellingford *et al.*, 2022). Not only do the authors modify the original 2015 ACMG protein-coding guidelines, but they also provide a table of 30 real-world non-coding variants and their target genes that had been annotated by expert variant curators according to these guidelines.

To perform our analysis, we examined the detailed clinical notes left by the variant annotators for each of the 30 individual variants analysed in the study (Ellingford *et al.* 2022, Table S1, Column R, “Curation notes”, <https://doi.org/10.1186/s13073-022-01073-3>). These variants varied across target genes, molecular mechanisms (promoter, enhancer, splicing, miRNA, 5/3’ UTR, RNA gene), and classification status: 18 Variants of Uncertain Significance (VUS), 7 Likely Pathogenic (LP) variants, 3 Pathogenic (P) variants, and 1 Benign (B) variant.

We assessed each variant using AlphaGenome and assessed whether predictions showed sufficient *in silico* evidence for a clear mechanism. When available, we verified the prediction against the mechanism proposed within the detailed curator notes. A top-line summary of the variant analyses is below: a full case study report detailing all findings is included in the new **Supplemental Note**.

Table 1: Summary of AlphaGenome applied to non-coding variant case studies.

Region type	Variant ID (Gene)	Prior Curation	AlphaGenome Finding	In silico evidence of variant effect	Impact
Splicing	chr1_215867824_G_C (USH2A)	VUS	Precisely predicts the creation of a 129 bp intronic cryptic exon, perfectly	Yes	Corroborates existing

Region type	Variant ID (Gene)	Prior Curation	AlphaGenome Finding	In silico evidence of variant effect	Impact
			matching prior minigene assays.		functional evidence
Splicing	chr15_42387805_C_G (CAPN3)	VUS	Predicts clear exonic skipping, outperforming default SpliceAI settings which miss the effect.	Yes	Corroborates existing functional evidence, see Case study 2
Splicing	chr11_66519596_A_T (BBS1)	Pathogenic	Predicts multiple competing novel splice acceptors, explaining the "complex splicing impact" noted by curators.	Yes	Corroborates existing functional evidence
Splicing	chr15_48430805_G_C (FBN1)	Likely Pathogenic	Accurately predicts a 2-nucleotide exon extension, replicating the exact molecular outcome seen in patient RNA-seq.	Yes	Corroborates existing functional evidence
Splicing	chr1_94010795_C_T (ABCA4)	Pathogenic	Predicts a drop in exon coverage and decreased splice site usage, consistent with curator notes on incorrect splicing.	Yes	Corroborates existing functional evidence
Promoter	chr2_127418408_A_G (PROC)	VUS	Predicts major loss of expression, disruption of activating histone marks, and identifies a promoter motif.	Yes	Corroborates existing functional evidence
Promoter	chr8_11703860_G_T (GATA4)	Not Available	Predicts no meaningful change in local activity or gene expression, suggesting the variant is non-functional.	None	Supports benign classification
Promoter	chr11_2171856_C_T (TH)	Likely Pathogenic	Predicts strong decrease in expression and confirms disruption of the known CREB motif via ISM.	Yes	Corroborates existing functional evidence
Promoter	chr13_48303720_T_G (RB1)	VUS	Predicts decreased expression and DNase	Yes	Corroborates existing

Region type	Variant ID (Gene)	Prior Curation	AlphaGenome Finding	In silico evidence of variant effect	Impact
			accessibility; ISM pinpoints the key functional motif within a known CRE.		functional evidence
Promoter	chrX_71223179_T_C (GJB1)	VUS	Predicts strong negative impact on expression; ISM identifies the critical SOX10 binding motif within an ENCODE CRE.	Yes	Extends existing evidence (base pair resolution motif within CRE)
Enhancer	chr1_209816135_A_AA (IRF6)	VUS	Predicts decreased expression and identifies a functional TF binding motif within a VISTA CRE.	Yes	Extends existing evidence (base pair resolution motif within CRE)
Enhancer	chr7_156791413_A_C (SHH)	Likely Pathogenic	Predicts decreased local enhancer marks but misses the effect on gene expression ~1MB away.	Partial	None
Enhancer	chr11_31664397_C_A (PAX6)	Likely Pathogenic	Predicts decreased local DNase accessibility but misses the effect on gene expression ~128kb away.	Partial	None
Enhancer	chr17_70680162_T_C (SOX9)	Benign	Limitation: Cannot predict; variant is 1.5Mb from the gene, exceeding the model's input length.	None	None
3' UTR	chr11_5225488_A_T (HBB)	VUS	Predicts strong expression decrease by disrupting a canonical polyadenylation motif.	Yes	Changes classification VUS → Likely Pathogenic , see Case study 1
3' UTR	chrX_49250456_T_C (FOXP3)	VUS	Predicts a clear drop in expression by identifying a disrupted polyA motif.	Yes	Corroborates existing functional evidence

Region type	Variant ID (Gene)	Prior Curation	AlphaGenome Finding	In silico evidence of variant effect	Impact
3' UTR	chr1_25816825_T_C (SEPN1)	VUS	Limitation: Predicts weak effect; true mechanism is disruption of mRNA secondary structure (out of scope).	None	None
3' UTR	chr1_100195897_T_G (DBT)	Likely Pathogenic	Correctly predicts decreased expression but underestimates the magnitude.	None	None
3' UTR	chr14_75958692_G_A (TGFB3)	VUS	Predicts no significant changes to gene expression.	None	None
5' UTR	chr5_1295046_T_G (TERT)	VUS	Predicts upregulation, providing in silico evidence that clarifies prior contradictory functional data.	Yes	Resolves ambiguity, supports Pathogenic
5' UTR	chr1_21509427_C_T (ALPL)	VUS	Predicts inconsistent and weak effects on expression at the TSS.	None	None
5' UTR	chr8_22130633_C_T (HR)	VUS	Limitation: No transcriptional changes predicted; mechanism is a missense change in a uORF (out of scope).	None	None
5' UTR	chr10_27100453_T_C (ANKRD26)	VUS	Predicts slight upregulation but with low confidence and some discordance.	None	None
5' UTR	chr19_11089410_CT_C (LDLR)	VUS	Predictions for expression changes are inconsistent across different readouts.	None	None
RNA gene	chr2_121530927_G_A (RNU4ATAC)	Likely Pathogenic	Limitation: No effect predicted; known mechanism (RNA secondary structure) is out of scope for the model.	None	None
RNA gene	chr3_169764963_CAG_C (TERC)	VUS	Limitation: No transcriptional effect predicted; known	None	None

Region type	Variant ID (Gene)	Prior Curation	AlphaGenome Finding	In silico evidence of variant effect	Impact
			mechanism (catalytic activity) is a non-transcriptional outcome.		
RNA gene	chr9_35657920_G_GCA (RMRP)	Pathogenic	Limitation: No effect predicted; mechanism is post-transcriptional (mRNA cleavage impairment).	None	None
RNA gene	chr3_169764918_C_T (TERC)	VUS	Predicts only a slight effect on the short transcript, with a quantile score that may be artificially high.	None	None
miRNA	chr7_129774757_C_T (MIR96)	VUS	Limitation: Fails to predict miRNA expression, a limitation of training on total/polyA+ RNA-seq data.	None	None
miRNA	chr15_79209844_C_T (MIR184)	Likely Pathogenic	Limitation: Fails to predict miRNA expression, a limitation of training on total/polyA+ RNA-seq data.	None	None

Overview of results

Overall, AlphaGenome was able to recapitulate the curators' hypothesis in all splicing and promoter examples (10/10). With splicing variants, the junction predictions were useful corroborations for molecular events such as exon skipping. For example, AlphaGenome was able to predict that a protein-coding missense variant on the CAPN3 protein (chr15_42387805_C_G) actually has an impact on splicing as well, aligning with functional evidence mentioned by the curators.

With promoter variants in particular, AlphaGenome was able to recapitulate functional evidence cited by the curators, including the direction of effect of the variant on the target gene. In instances where the variant curators mentioned ENCODE *cis* regulatory elements that overlap the variant (300 bp), AlphaGenome was able to provide additional evidence of specific base-pair resolution motifs that enhance the interpretation of variant function.

Out of the four enhancer variants, AlphaGenome provided a clear molecular hypothesis for one enhancer nearby its gene (*IRF6*). For two variants further away, AlphaGenome predicted a local change in accessibility at the variant position but failed to predict the corresponding gene

expression change. The distance from the final variant to its target gene exceeded our input sequence length and could not be scored.

On 3' UTR variants, AlphaGenome was helpful in classifying two polyA tract mutations (more on this below), but was unable to inform the other three variants. Similarly, for the five 5' UTR variants, AlphaGenome was helpful in resolving one near-promoter variant, but unhelpful for the other three, and could not score the remaining one due to an out-of-scope mechanism (missense variant in upstream ORF).

Finally, AlphaGenome showed a particular weakness for interpreting variants affecting non-coding RNAs (e.g., miRNAs). This is a direct consequence of a known bias in our training set, which consists of total and polyA-enriched RNA-seq protocols that inefficiently capture small RNA signals.

Case Study 1: Refined interpretation of VUS: NM_000518.5(HBB):c.*110T>A

We wish to highlight a particular variant of interest here. A beta-thalassemia VUS (NM_000518.5(HBB):c.*110T>A, chr11_5225488_A_T) affecting the *HBB* gene was strongly predicted by AlphaGenome to lower the gene expression of the *HBB* target gene (**Supplemental Note: Revisiting clinically annotated non-coding variants with ACMG guidelines**). Additionally, *in silico* mutagenesis indicates that this variant affects a clear polyadenylation motif in the 3' UTR of the gene. Finally, pathogenic mutations at nearby positions, and one at the same position (chr11_5225488_A_T), are all predicted by AlphaGenome to have similar effects in terms of the magnitude and direction of the variant. If this *in silico* evidence was available at the time of the curation in 2022, the curators would have been able to activate the ACMG evidence code PM5_Moderate instead of the weaker PM1_Supporting. In combination with the remaining evidence codes provided by the clinical variant annotators (PM2_Supporting; PP4; PM3), this would resolve the **VUS** to **Likely Pathogenic** according to ACMG guidelines.

We have spoken to both corresponding authors of the 2022 ACMG non-coding guidelines to collaborate in a ClinVar submission. The existing ClinVar annotation for this variant is a zero-star entry from 2004 from the authors of a case study. Depositing an ACMG compliant Likely Pathogenic submission would result in a one-star entry that supersedes this submission and updates this variant with modern, ACMG-compliant evidence.

Case Study 2: Interesting variant mechanism (missense variant with splicing effect)

Another variant of interest that emerged was a variant affecting CAPN3 (CAPN3(NM_000070.3):c.551C>G p.(Thr184Arg), chr15_42387805_C_G), which is a missense mutation with high AlphaMissense (0.9281) and PrimateAI-3D (0.92) scores. However, AlphaGenome predicted clear skipping of the exon, as illustrated both in the predicted RNA-seq coverage and the splice junction predictions (**Supplemental Note: Revisiting clinically annotated non-coding variants with ACMG guidelines**).

Interestingly, the clinicians in Ellingford *et al.* 2022 report a low SpliceAI score for this variant (0.02). We surmise that they used precomputed SpliceAI scores through the VEP plugin, which uses a default max_distance parameter of +/-50 bp. As the splice acceptor affected by this variant is 52 bp away, the effect of this variant is missed using default settings. This is a known limitation of the VEP SpliceAI plugin as discussed by a recent preprint (Using SpliceAI to triage splice-altering variants in 7,220 individuals with rare conditions highlights limitations of the precomputed scores | medRxiv). As suggested in this reference, setting max_distance to larger values in tools such as SpliceAI Lookup results in a more accurate classification (<https://spliceailookup.broadinstitute.org/#variant=chr15-42387805-C-G&hg=38&bc=basic&distance=500&mask=0&ra=0>, delta acceptor loss score = 0.22).

Summary and takeaways

In short, we hope that this substantial addition to the publication (**Supplemental Note: Revisiting clinically annotated non-coding variants with ACMG guidelines**), novel analysis of real-world variants, comparison to other tools, and direct improvement to the ClinVar database satisfies the reviewer's suggestion of "compelling case studies—such as refined interpretation of VUS (variants of uncertain significance) or real-world applications in diagnosis or personalized medicine—that would have been difficult to achieve with existing models".

Note on ClinVar analysis

Finally, as suggested by the reviewer, we explored using reclassified variants from the ClinVar database for our analysis, but found that the data was unsuitable due to a substantial bias. We discovered that variants that were reclassified from VUS to "Pathogenic" were predominantly splicing errors, while those that were reclassified from VUS to "Benign" were typically synonymous or intronic variants. This bias would make the classification task too simple, as a model would only need to learn to separate these easily distinguishable categories rather than identify the complex signals of pathogenicity. This motivated our search for a more class-balanced set of variants that had additional ACMG-compliant curation, which we found in the Ellingford *et al.* 2022 list discussed above.

Note on "rate ratio or enrichment" for Fig 6f

This is addressed in the response to a similar comment below.

Addition to main text section titled "AlphaGenome enables a multimodal analysis of variants" describing this analysis:

"To further assess AlphaGenome's real-world applicability, we analyzed 30 clinically annotated non-coding variants from recent ACMG guidelines. The model successfully recapitulated the molecular mechanisms for splicing and promoter variants, including cryptic exon creation, exon skipping, and transcription factor motif disruption. Furthermore, AlphaGenome provided *in silico* evidence that two polyadenylation motif variants of uncertain significance may affect transcript abundance levels of their corresponding disease genes. While challenges remain, particularly for variants affecting non-coding RNAs or acting over long distances, these case studies

demonstrate the model's utility in corroborating functional data and refining the interpretation of clinically relevant variants."

The authors write: "This makes AlphaGenome a useful complement to conservation-based measures of deleteriousness which are agnostic to a variant's mechanism of action." It would be helpful to show a concrete example where AlphaGenome not only predicts a strong deleterious effect that matches conservation-based metrics, but also suggests a novel mechanism of action. Such an illustration would demonstrate the model's potential to guide novel biological insights.

To demonstrate AlphaGenome's ability to provide mechanistic insights beyond the information provided by conservation, we have added a few illustrative examples to the manuscript.

1. Providing basepair-resolution functional information within broad regions of conservation

As a first concrete example, consider the TraitGym variant chrX:101386224:T>C, which is linked to X-linked Agammaglobulinemia (10.1542/peds.101.2.276, <https://doi.org/10.1016/j.ajhg.2016.07.005>) and is caused by a lack of functional BTK. The variant occurs in a region with a long stretch of highly conserved positions. While the high conservation flags this region as functionally important, it is agnostic as to the mechanism. In contrast, AlphaGenome's *in silico* mutagenesis (ISM) pinpoints the disruption of a SPI motif that causes a reduction in SPI1 localization and leads to decreased BTK expression. We have added this observation as a new panel (i) to the **Extended Data Fig. 9**:

Extended Data Fig. 9i:

We have also added this caption to describe the new panel:

Extended Data Fig. 9i caption: “(i) AlphaGenome predictions for the variant chrX:101386224:T>C, previously reported to result in a lack of functional BTK and causing X-linked Agammaglobulinemia (OMIM #300755). The model predicts reduced binding of SPI1 at the BTK promoter, diminished local DNA accessibility, as well as a reduction in expression of the BTK transcript in B & lymphoblastoid cells. ISM of the reference sequence suggests that the variant disrupts a SPI-like motif (SPI1.H13.0.PB shown). Also shown is the 100-vertebrate PhyloP conservation track, which shows high conservation around the variant.”

2. Gain-of-function variants within poorly conserved regions

Another example of AlphaGenome’s utility over pure conservation measures is the ability to predict gain-of-function variants, which may occur in neutral or fast-evolving genome positions (as opposed to loss-of-function variants, which tend to be in conserved regions). For example, the frequently mutated site upstream of *TAL1* (chr1:47239296, *MUTE* locus) has a weak negative phyloP score of -0.215, indicating slightly faster evolution than neutral. Similarly, in **Extended Data Fig. 9 e, f and g**, we also illustrate variants which cause the creation of a binding motif at neutral or fast-evolving positions in the genome (phyloP scores of -3.29, -0.67, -0.20, respectively). We have updated the captions for the **Extended Data Fig. 9** with these numbers as appropriate.

Extended Data Fig. 9 caption:

“(e) AlphaGenome predictions for the variant chr5:1295046:T>G, previously reported to cause TERT overexpression associated with malignant melanoma 7. AlphaGenome predicts increased binding of ETS factors (ELF1 shown) and a significant increase in TERT expression in melanocytes. Comparing the ISM around the reference and alternative alleles suggests that the variant creates an ETS factor binding motif. The phyloP score for this position is -3.29, consistent with the fact that gain-of-function variants can occur in fast-evolving parts of the genome.”

“(f) AlphaGenome predictions for the variant chr11:5254983:G>C, previously implicated in hereditary persistence of fetal hemoglobin. The model predicts increased CTCF binding, as well as increased HBG2 expression (shown for CD14+ monocytes). Comparing the ISM around the reference and alternative alleles suggests that the variant creates a CTCF binding motif. The phyloP score for this position is -0.67, indicating that the variant position is under relatively neutral selection (given the low magnitude of the score).”

“(g) AlphaGenome predictions for the variant rs12740374, previously found to be associated with low-density lipoprotein cholesterol levels. Additionally, the variant is a plasma pQTL for CELSR2, as well as GTEx eQTL for CELSR2 and PSRC110,11. Consistent with the observed molecular QTL effects, AlphaGenome predicts increased expression of both of these genes in liver tissue. Moreover the model predicts increased CEBP binding on the alternative allele. Comparing the ISM around the reference and alternative alleles suggests that the variant creates a CEBP binding motif. The phyloP score for this position is -0.20, indicating that the variant position is under relatively neutral selection (given the low magnitude of the score).”

3. AlphaGenome complements PhyloP scores on the TraitGym benchmark

To test whether AlphaGenome provides information that is complementary to evolutionary conservation, we created a simple combined score by summing the AlphaGenome and PhyloP conservation scores. On the Mendelian traits benchmark, this combined score achieved an auPRC of 0.66, substantially outperforming both PhyloP alone (0.55) and AlphaGenome alone (0.49). The improvement was even more pronounced on the complex traits benchmark, where the combined score's auPRC of 0.65 was a substantial leap over both PhyloP (0.17) and AlphaGenome (0.24). This result demonstrates that AlphaGenome captures functional information that is orthogonal to evolutionary conservation, and that combining these two approaches yields a more powerful predictor of disease relevant variant effects.

Minor Comments and Suggestions

In general, many of the statistics shown are ROC or correlations, which is sensible to compare to previous methods. But in many cases, these ROCs are too low to be clinically useful (again, understandably as these are hard tasks). As in the major comment above, are there any cases where more interpretable metrics like rate ratios or enrichments, could be used to more directly highlight the utility in various analytic spaces?

[Consolidating with a similar comment, repasted here from above: There is some attempt at this in Fig. 6f, but this is very hard to read: this reviewer is trying to translate it into an ROC curve but it feels tricky. Could some sort of rate ratio or enrichment be used to show this, and compare to a previous method?]

We thank the reviewer for this excellent suggestion. We agree that interpretable metrics are crucial for clinical utility. We have incorporated more readily interpretable metrics in two key instances in the manuscript: prioritizing disease-causing variants (**Fig. 6f**) and predicting the direction of expression-altering variants (**Fig. 4g**).

Figure 6f directly addresses this point of interpretable metrics for variant prioritization by providing an enrichment ratio to help users interpret AlphaGenome scores in a practical setting. This figure concerns using AG to prioritize disease-causing variants for Mendelian and complex traits. The plot clearly shows the trade-off between score thresholds, recall, and enrichment. For example, it demonstrates that while enrichment is modest at lenient thresholds, causal Mendelian variants are **10.7-fold more likely** than matched controls to have a top 0.5% AlphaGenome score. This provides a tangible guide for prioritizing candidate variants. The plot also allows users to estimate the false-positive rate (from the left-hand side) at any given threshold. We have improved the caption in the manuscript to make these points clearer (see below).

We acknowledge that enrichment values are highly dependent on the control set. For this reason, our analysis uses the carefully matched control variants from the TraitGym benchmark, which account for confounding features like distance-to-TSS, to provide a more realistic and robust estimate of performance. While we use standard auPRC metrics for direct model comparisons in the supplementary material, **Figure 6f** is designed specifically for the interpretability you suggest.

f Multimodality in trait-altering non-coding variants

We have updated the caption for **Fig. 6f** for increased clarity:

(f) Fraction of trait-affecting variants (338 for Mendelian and 1140 for complex traits), as well as matched control variants (3042 and 10260, respectively), which exceed varying AlphaGenome quantile-score thresholds in at least one predicted track. Variants are categorized depending on the tracks where the threshold was passed: ‘Local Regulation’ (ChIP/DNase/ATAC), ‘Expression only’ (RNA/CAGE) and ‘Multimodal’ (combination of the above). Numbers in the center indicate the relative enrichment of detected variants amongst candidate causal variants compared to the control variants. Here, we define a variant candidate as ‘detected’ if there is at least one track for which AlphaGenome’s variant effect score exceeds the stated percentile (e.g. 1.0 means the ‘quantile score’ > 0.99; see **Methods**). The enrichment increases with increasingly stricter thresholds, with a reduction in recall (x-axis). For instance, causal variants for Mendelian traits are 10.7-fold more likely to have an AG score in the top 0.5% compared to control variants.

Additionally, in **Fig. 4g**, which covers predicting an eQTL's direction of effect, we use a precision-recall curve that is more informative than a single auROC value. This allows a user to select a score threshold that fits their experimental goals. For instance, a researcher can choose a threshold that yields 90% precision (i.e., 90% of predictions have the correct sign) and see that this will recover 41% of all true eQTLs (recall).

g Fraction correct sign

(The x-axis label “Fraction with correct sign” was previously “Target metric value”; see reviewer 3 response below)

We also provide a slightly different view of these data in **Extended Data Figure 5i**, where we show, for a variety of score thresholds, what the performance is on different eQTL tasks. For instance, we estimate that variant/gene pairs with AG scores in the top 99th percentile will have the correct sign 80% of the time, and compute that 49% of eQTLs have scores at least this high.

Similarly, we also provide full ROC curves for several accessibility variant evals in **Supplementary Figures 9 a, c, e, h**.

Together, these figures are designed to provide the kind of intuitive, practical guidance requested by the reviewer, moving beyond standard aggregate metrics to highlight the model's utility in common analytic scenarios.

The tools for genome track prediction are appreciated. To enhance usability for the community, would it be possible to release the full predicted genome tracks, particularly for key modalities like splicing? Ideally also with a mechanism to input variant data to see how things change. A browser framework showing this would be incredible but probably out of scope for this manuscript.

Since submission, we have now released the AlphaGenome API (<https://deepmind.google.com/science/alphagenome/>), which allows users to freely make full predictions of all AlphaGenome genome tracks, including splicing tracks. We have included user-friendly documentation on the specific use case requested by the reviewer (i.e. seeing how predicted genome tracks change across variants) here: https://www.alphagenomedocs.com/colabs/variant_scoring_ui.html

Upon formal publication, we will also release the model parameters, allowing researchers to make predictions using the model on their own hardware.

Could AlphaGenome be applied to study the spatial relationship between regulatory elements (e.g., TF binding sites) and their target genes?

Yes, AlphaGenome is well-suited for this type of *in silico* experimentation to examine the syntax of TF binding site placement. For example, a user could computationally insert a specific TF binding motif at varying distances from a transcription start site (TSS) and use the model to observe the predicted changes in gene expression. This provides a direct method of systematically mapping the optimal spacing and context for a given TF's predicted activity. We provide the public AlphaGenome API as a framework to enable the community to generate and test such hypotheses (specifically, the `predict_sequence` function is most relevant for this idea).

We have previously applied similar ideas to models such as BPNNet in the past (<https://doi.org/10.1038/s41588-021-00782-6>), but have yet to explore this analysis with

AlphaGenome. Progress in this space using deep learning models has been nicely summarized in a recent review, <https://doi.org/10.1038/s41576-025-00841-2> in their Fig. 4d.

Can AlphaGenome detect features associated with microRNA loci or their regulatory effects? Even if not explicitly trained for this, it would be helpful to clarify whether the model implicitly learns patterns characteristic of such regions.

As shown in the response to the first Major Comment, AlphaGenome exhibited a weakness in predicting RNA-seq coverage of the miRNAs themselves for the two miRNA loci that we explored from the Ellingford *et al.* 2022 list of variants.

To test AlphaGenome's understanding of miRNA regulation, we selected 6 well-validated miRNA-target gene interactions (let7b-LIN28, mir125a/b-LIN28B, mir204-5p/REEP1, mir451a/CAB39, mir223-3p/IGF1R, mir9/HES1) and performed *in silico* mutagenesis across all the bases of each of the miRNA seed binding regions on the 3' UTRs of the target genes. We then examined the resulting contribution scores for the RNA-seq prediction head for multiple ontologies. AlphaGenome does not seem able to predict the effect of mutations in miRNA binding sites. While we do not exclude that AlphaGenome will be able to correctly predict miRNA regulation in simpler cases (e.g. single-isoform genes or well-defined cell types with high-quality training data), we do not currently recommend using AlphaGenome for miRNA-related predictions.

We have added these limitations to the appropriate section in the discussion:

"...Moreover, although the model predicts tissue and cell type-specific genome tracks with some success, accurately recapitulating tissue-specific patterns across cellular contexts and predicting condition-specific variant effects remains challenging. Both our training data and evaluations are heavily focused on protein-coding genes, and future work can be done to improve coverage of non-coding genes such as microRNAs."

A comparison with constraint (e.g., Gnocchi) and conservation scores could offer insights. Is there general overlap with these, and are there regions where AG and conservation metrics diverge?

We have responded to a similar question on conservation asked by Reviewer 1 above (see their second Major Concern). We hope that this response addresses the reviewers' questions on where conservation overlaps AlphaGenome scores and where these two measures diverge.

On the question of constraint (e.g. Gnocchi) – we have not performed an explicit analysis, but we expect the conclusions to be similar to those regarding conservation metrics such as PhyloP. One particular detail about Gnocchi is that its constraint scores are at 1kb resolution, so our first point above ("Providing basepair-resolution functional information within broad regions of conservation") would be particularly relevant.

For the A→C mutation example, how specific is the predicted effect? Would neighboring mutations or different mutations at the same position (e.g., A→T or A→G) produce similar predictions? A short discussion on the specificity of mutational impact would strengthen the mechanistic interpretation.

For the eQTL at chr22:36201698:A>C, we investigated both aspects of mutational specificity raised by the reviewer.

First, we tested the effect of other mutations at the same position. We scored the A→T and A→G mutations using AlphaGenome for their predicted effect on *APOL4* expression. As shown in the table below, the predicted scores are nearly identical for all three alternative alleles:

Variant	Raw score	Quantile score
chr22:36201698:A>C	-1.533253	-0.999980
chr22:36201698:A>T	-1.533307	-0.999980
chr22:36201698:A>G	-1.533484	-0.999980

This data confirms that the model's predicted effect is not specific to the C allele; rather, it predicts a strong negative impact for any mutation that disrupts the reference "A" nucleotide at this position. However, in general, we have observed that the model can be quite sensitive to the exact nucleotide swap at a given location (although in this particular example, it is not).

Second, we assessed the impact of neighboring mutations using the existing *in silico* mutagenesis figure (**Fig. 4b**, inset). In the figure, the height of the sequence logo indicates the contribution of that position to the expression of *APOL4*. This is calculated as the mean variant effect of the three possible alternate alleles at that position. The fact that the variant is one of the nucleotides in a 6 nucleotide segment (ACTCAC) that has high scores indicates that any single nucleotide mutation in that segment is predicted to have a positive contribution to *APOL4* expression (and therefore a negative variant effect score when mutated).

To interpret this further, *APOL4* is transcribed from the negative strand, and the reverse complement of the 6 nucleotide sequence (GTGAGT) matches very well with the GT-AG splice site motif characteristic of splice donors (<https://www.nature.com/articles/nrg.2016.46>).

We have amended our discussion of this variant in the Main Text to reflect this:

"An example at a known eQTL/sQTL locus (rs9610445, chr22:36201698:A>C, posterior inclusion probability (PIP)=0.90) illustrates the model's predictions aligning with observed allele-specific expression differences from GTEx data (Fig. 4b). Furthermore, ISM around the variant suggests a possible mechanism: disruption of a predicted splicing motif by the alternative 'C' allele, potentially 6 nucleotide splice donor sequence motif which is disrupted by the variant, thereby leading to an aberrant transcript and reduced expression (Fig. 4b, inset)."

Across modalities, how do the correlations between predicted and observed tracks compare to cross-species correlations (e.g., human vs. mouse)? Also, was training on mouse tracks beneficial for human track prediction? An ablation study might help clarify this.

Cross-species correlations

To explore how correlations between predicted and observed tracks compare to cross-species correlations (e.g., human vs. mouse), we started with the gene-based RNA-seq evaluation setup in **Figure 2d**. To allow for a valid comparison, we restricted the dataset to one-to-one human mouse orthologous gene pairs, which resulted in 986 gene pairs present in the held-out intervals. Furthermore, we matched RNA-seq tissue metadata across organisms by using UBERON IDs (which are organism agnostic), resulting in 23 shared tissues. The results on the cross-gene, cross-gene (normalized) and cross-tissue (normalized) evaluations are shown below, with the mean of each distribution shown above the violin plot:

In general, we observe that the cross-organism metrics fall below the observed vs. predicted metrics.

The remaining genome track modalities are evaluated directly on held-out intervals. As such, a cross-organism analysis on these modalities was not possible because there are no inventories of 1 MB genome regions that are 100% identical between human and mouse, such that the observed human vs. observed mouse genome tracks could be compared. To achieve this would either require drastically reducing the input sequence length (which would change the nature of the evaluation), or finding a liftover method that could handle 1 MB sized intervals.

Benefit of training on mouse tracks

This benefit has been established in the literature by David Kelly at Calico, who first showed that cross-species training was a key way to boost model performance by adding additional sequence diversity to the training corpus, resulting in “improved test set accuracy for 94% of human CAGE and 98% of mouse CAGE datasets” (Cross-species regulatory sequence activity prediction | PLOS Computational Biology). In our research on Enformer (<https://www.nature.com/articles/s41592-021-01252-x>), we also noticed clear and substantial drops in performance when mouse tracks were withheld from training. Given our experience from this past work, we did not do an explicit data ablation of mouse data, but would expect it to yield similar results.

In combining AlphaGenome and Borzoi features, is there a measurable gain in predictive power? This could help clarify whether Borzoi retains complementary signals not fully captured by AlphaGenome.

While we have not explored the combination of AlphaGenome and Borzoi features in all evaluations where this is possible, during our implementation of the ENCODE enhancer to gene linking (E2G) benchmark, we did observe the result below. Our evaluation tested whether or not adding ‘input x gradient’ features derived from AlphaGenome to the state-of-the-art ENCODE E2G model (<https://www.biorxiv.org/content/10.1101/2023.11.09.563812v1>) would boost performance on the task.

When combined with the SOTA rE2G features, we observed the following auPRC values on the task:

- rE2G extended with AlphaGenome features - 0.7813 (as reported in **Extended Data Fig. 7b**)
- rE2G extended with Borzoi features - 0.7648
- rE2G extended with both AlphaGenome and Borzoi features - 0.7804

In this setting, adding Borzoi and AlphaGenome features to the model performed slightly worse than including the AlphaGenome features alone.

The TAL1 example is compelling. Are these regulatory changes predicted to be tissue-specific? For instance, how does AlphaGenome score the same variants in other tissues beyond CD34+ CMP?

We performed the suggested analysis by comparing the effect of oncogenic variants versus shuffled background variants across all RNA-seq samples available. Within each sample, we used a two-sided Mann-Whitney U statistic to assess the group difference between predicted *TAL1* expression for oncogenic and shuffled background variants.

Below is a plot where each sample is a datapoint, with each facet representing a different biosample type: *in vitro* differentiated cells, primary cells, GTEx tissues, and cell lines. Within each group, each sample is ranked in the x-axis by the magnitude of the Mann-Whitney U statistic (on the y axis). Within each biosample type, there is a single labeled red data point,

highlighting the sample that most closely matches the T-ALL tissue of origin within that biosample type. For primary cells, this is the 'common myeloid progenitor, CD34-positive' biosample, which is what was used in the original figure. For *in vitro* differentiated cells, this was 'hematopoietic multipotent progenitor cell'. For GTEx tissues, this was the thymus. For cell lines, this was Jurkat cells (Jurkat cells are a human T lymphocyte cell line that harbors the MUTE site).

In 3 of the 4 biosample types, the appropriate tissue/context appears as the top ranked sample. For cell lines, the chosen biosample (Jurkat cells) is in the top 5. This suggests that the regulatory changes predicted by AlphaGenome can be highly tissue-specific. We have added this figure to the manuscript as **Supplementary Fig. 10** to complement **Figure 6**:

This new figure has the following caption:

“Supplementary Figure 10: Tissue specificity of *TAL1* upregulating oncogenic variants. Each data point is a human RNA-seq track. For each track, the group difference between predicted *TAL1* expression for oncogenic and shuffled background variants defined in Figure 6 was assessed using a two-sided Mann-Whitney U statistic. The plot is split across four biosample types: *in vitro* differentiated cells, primary cells, GTEx tissues, and cell lines. Within each group, each sample is ranked in the x-axis by the magnitude of the Mann-Whitney U statistic (on the y axis). A single labeled red data point highlights the sample that most closely matches the T-ALL tissue of origin within that biosample type.”

We have also added the following sentence to the Main Text:

“Across modalities, oncogenic variants exhibited a distinct predicted mechanism compared to shuffled controls, as shown via unsupervised clustering (Fig. 6d). Finally, these observations are tissue-specific. By reproducing this analysis across all RNA-seq tracks, we identified that the difference in oncogenic versus shuffled mutations is most pronounced in T-ALL relevant tracks such as ‘Thymus’, ‘CMP’, and ‘hematopoietic multipotent progenitor cells’.”

As a very minor note, the authors sometimes mention that performance is "significantly" better but fail to report a p-value, and instead an increased AUC. Could these be removed or better yet, replaced with significance tests?

Thank you for this important suggestion. We have systematically removed all mentions of significance from the manuscript when used in regards to model comparisons. Those changes are as follows:

*“Using fine-mapped GTEx eQTLs as ground truth, we benchmarked AlphaGenome against the SOTA models Borzoi and Enformer. AlphaGenome demonstrated **significantly** improved prediction of ...”*

*“However, leveraging AlphaGenome’s predictions in a supervised framework **significantly** boosted performance on the causality task: training a random forest model using AlphaGenome’s scores from multiple modalities increased the mean auROC from 0.68 to 0.75, also surpassing previous SOTA performance (mean auROC 0.71 for Borzoi, Fig. 4i). Notably, using features derived from variant scores across all predicted modalities provided a **significant** performance uplift in this supervised setting compared to using RNA-seq derived scores alone (Extended Data Fig. 5e), highlighting the practical benefit of AlphaGenome’s multimodal predictions for identifying causal expression-modulating variants.”*

*“First, models trained on longer DNA sequences (up to the full 1 Mb) **significantly** outperformed those trained on shorter sequences (32kb or less)...”*

*“Overall, distillation offers a way to achieve strong performance with **significantly** reduced inference cost compared to evaluating large ensembles, facilitated by scalable training strategies like assigning one unique teacher per training device (Methods).”*

*“It **significantly** advances our ability to predict molecular functions and variant effects from DNA, offering valuable tools for biological discovery and applications in biotechnology.”*

“AlphaGenome predicts increased binding of ETS factors (ELF1 shown) and an significant increase in TERT expression in melanocytes.”

“(h) Failure cases: ~~Significant~~ underprediction of RNA-seq signal for liver-specific genes ALB (Albumin) and TF (Transferrin).”

“(i) Failure cases: ~~Significant~~ overprediction of MRPL23 (Mitochondrial Ribosomal Protein L23) and NEFL (Neurofilament Light Chain) RNA-seq signal in astrocytes.”

Referee #2:

This manuscript introduces the AlphaGenome model, a state-of-the-art functional genomic predictor that can predict 11 genomic modalities from the DNA sequence alone. This model can be understood as a very significant improvement over Borzoi (Linder et al., Nature Genet., 2025), and it extends beyond this previous general purpose model in many ways. Broadly speaking it extends the sequence window up to 1 Mb and the resolution down to 1 bp for many but not all genomic tracks. There are many expansions to the architecture and expanded capabilities to predict explicitly additional genomic tracks, taking advantage of additional training datasets modalities. The manuscript is very dense and the technical advances are many. It is not always straightforward to understand the differences between these two models and their respective training datasets. Examples of differences in training data include the inclusion of the PRO-cap start-site assays, 3D chromatin contact maps, explicit splice-junction quantification, and an extended set of GTEx datasets. There are also major advances in the architecture to enable representation at different resolutions and model long-range interactions. Finally, there is also an additional technical improvement in inference time efficiency by training a distilled “student” model that learns from an ensemble of all-folds teachers.

Across almost all modalities tested, AlphaGenome can outperform both the general purpose Borzoi model as well as task-specific models for some of the modalities. For many of the tasks it should be clear that the improvements are modest, but it nevertheless remains an impressive advance given that the task-specific models can often be trained more specifically and can often outperform more general models.

The evaluation of the model seems convincing with minor concerns about the use of the distilled student model in some evaluations. The authors performed extensive ablation studies, although it could have been interesting to better understand or at least provide and discuss some examples how some modalities benefit from the multi-model joint training.

To address the reviewer’s interest in multi-modal joint training, we have performed a new set of experiments in which modalities were sequentially added to the training data. This allows us to measure the marginal contribution of each modality relative to the previous training iteration.

We have described these changes in the **Methods**:

“Cumulative modality addition experiments

To determine the marginal performance contribution of each data modality, we trained a series of cumulative models. This ablation study started with a baseline model trained only on RNA-seq data. Subsequent models were trained by progressively adding data modalities in a fixed order, allowing us to quantify the performance gain at each step. The cumulative order of

addition was: 1) RNA-seq, 2) + ATAC-seq, 3) + DNase-seq, 4) + Histone Modifications, 5) + TF
ChIP-seq, 6) + Contact Maps, and 7) + CAGE/PRO-cap (the full model)."

These results are shown in our updated **Supplementary Figure 11** (rightmost column).

We also modified the **Supplementary Figure 12** legend to reflect this change:

“Supplementary Figure 12 | Impact of excluding, isolating, and cumulatively adding modalities on track and variant tasks.

Comparison of AlphaGenome performance on variant and track prediction tasks when training excludes a single output/modality group (left), when training only a single modality group (middle), and when cumulatively adding modalities (right). The x-axis indicates the specific output head (e.g., 'CAGE'; regular size label) or the broader modality group (Expression, Accessibility + contact maps, and Splicing; bolded label) targeted by the ablation. Each dot represents an independent training run (n=4 for 'Exclude Modalities' and n=8 for 'Train only Specific Modalities' random seeds); black dashed lines indicate the mean performance of the full multi-task model for reference; green dots indicate when the task is related to the ablated modality and blue is the default color.”

Generally, the cumulative addition experiments show that there are gains on track evaluations for specific modalities once the relevant training data is added to the mix (as expected), but there may be diminishing returns from adding modalities beyond RNA-seq and DNA accessibility for most of the variant evaluations.

We have added the following sentence to the **Results** section to summarise this observation:

“Complementary experiments showed that excluding single modality groups during training typically resulted in only modest performance decreases, suggesting model robustness likely aided by informative redundancy between modalities (Supplementary Fig. 12, left panel). Training on single modalities was often detrimental to variant effect prediction performance (middle panel). Cumulative addition experiments showed that while track predictions improved most when their relevant data types were added, variant effect tasks benefited most from the initial inclusion of expression and accessibility data, with diminishing returns from subsequent additions (Supplementary Fig. 12, right panel). Together, these results demonstrate the value of the unified multimodal approach for achieving broad, high performance, particularly for tasks integrating diverse regulatory signals.”

The model and training is very extensively described and the authors at least claim that the model will be available to use via an API and the code and model weights will be available upon publication. This is critical and should be a requirement for publication.

I only have some minor concerns for clarifications or discussion:

Given the multi-modal data, was there no per-modality gradient weighting or scheduling strategy used? Are there any differences in weights in the loss-functions for the different modalities?

The total training loss is largely an (unweighted) sum of the losses from each modality's prediction head. Beyond this, we did not use a complex explicit per-modality gradient weighting or scheduling strategy. However, we do use a small number of fixed, internal weighting coefficients to balance different objectives *within* a single modality. Specifically:

- The auxiliary gene-level expression loss, which encourages tissue-specificity, is added to the main RNA-seq loss with a small, fixed weight of 0.1.
- The splice junction loss combines several terms, with the final loss being $0.2 * \text{ratios_loss} + 0.04 * \text{counts_loss}$ to balance the learning of splicing ratios and absolute counts.

These internal weights were determined during model development. We found this simpler overall approach to be robust and effective without requiring an extensive hyperparameter search for cross-modality loss weights.

We have added a sentence to the Methods (“Output Heads” section) saying that we did not explore head-specific loss weighting strategies:

“The subsequent sections detail the specific parameterization and loss function defined for each prediction modality. The total training loss for AlphaGenome is the sum of these losses defined by each head with no additional weighting coefficients.”

Regarding the training of the student model from the ensemble of teacher models, it was clear that this procedure is very relevant for improving inference time. However, it generally seemed to perform worse in testing. In the ablation testing section, Fig. 7c, why is there no data for the results obtained with the 64 aggregated models?

This observation is accurate and highlights a trade-off in model distillation. The teacher ensemble's predictive power stems from averaging out the errors of diverse models. A single student model, by design, has lower capacity and might not perfectly capture the average of the collective. However, it is possible that the distillation procedure we developed is not optimal and could be improved in the future to fully close the gap with model ensembling. Nevertheless, the performance gap to the ensemble is much smaller than the uplift from the single non-distilled model. We considered this to be an acceptable trade-off for the substantial gains in inference speed and model accessibility.

On that point, the ensemble of 64 teacher models was not possible to perform due to hardware memory limitations in our implementation, especially for genome track prediction benchmarks. Our framework can load a maximum of 16 models into GPU memory simultaneously. In contrast, the use of 64 unique teachers during distillation was possible because for every input sample, we use the predictions of only a single teacher to compute each loss sample. This is why in **Fig. 7c**, as the reviewer astutely observed, there is no data for the 64 aggregated models.

We will clarify these points in the relevant Methods section (“Impact of Ensembling and Distillation”) in the revised manuscript:

“To assess the benefits of ensembling and distillation, 64 independent models were pre-trained using the Fold 0 data partition. Single student models (with the same architecture) were produced by distilling knowledge from ensembles of these 64 Fold 0 pre-trained teacher

models, using 1, 4, or 64 unique teachers in the distillation process, following the procedure described in the main "Distillation" methods. The performance of mean ensembles was evaluated by averaging the predictions of randomly selected subsets of these 64 pre-trained models. The maximum ensemble size possible was 16 due to memory limitations of our implementation."

For augmentation, it isn't stated if there is any specific scheduling of the different introduced genomic alterations made throughout training. Did the authors try different parameters for the introduction of the variation? It would be interesting to have a better sense from the authors of how important this augmentation is for model performance. Presumably, this is part of the reason why there is an improvement between the pretrained single model and the distilled version in Fig. 7c.

We assume that the first part of this comment refers entirely to the distillation procedure. As a side note, during teacher training, we are using reverse complement augmentation and sampling genomic intervals at slightly different offsets.

During distillation, we used a fixed, uniform sequence replacement rate of 4% throughout and did not employ a specific scheduling curriculum. This 4% rate was chosen based on initial experiments during model development where we examined the effects of different rates. We found that while higher rates (e.g., >4%) allowed the model to train stably, they led to a drop in performance on key tasks like splicing variant prediction, suggesting that the model was learning the reference genome less effectively. Lower mutation rates also led to slightly lower performance, likely because the distillation set had lower sequence diversity.

As a demonstration of the importance of the sequence augmentation, we have performed an ablation study where we repeated the distillation procedure without the input sequence perturbation (point mutations and indels). The resulting student model distilled in the absence of sequence perturbations showed a drop in performance relative to the student model reported in the paper (which has a replacement rate of 4%).

We have added the following statement to the manuscript to reflect these results:

"...Interestingly, distillation of even just a single teacher model was beneficial for some variant effect prediction tasks such as caQTLs, splicing outliers or eQTL sign prediction. Finally, distillation without randomly mutating the input sequence resulted in a drop in student model performance (eQTL sign: -0.06, eQTL causality: -0.01, sQTL causality: -0.01, splicing outlier -0.015), affirming the impact of our input perturbation strategy during distillation. ..."

We think that the improvement of the single distilled model compared to the single teacher model is mostly stemming from reverse complement augmentation, which acts similarly to test time augmentation (TTA).

The authors do not mention any confidence estimates. Is the distilled student model also learning an estimate of uncertainty from the ensemble? Would this measurement of uncertainty,

either directly measured by the ensemble or estimated by the student, be useful in downstream prediction tasks?

While our final distilled student model does not directly learn an uncertainty estimate, perhaps the variance of predictions across the teacher ensemble may provide a measure of model uncertainty. This being said, the teacher models are trained on the same underlying data and thus may exhibit correlated errors. We agree that investigating this is an interesting area for future research and have noted this in the discussion:

“Moreover, integrating AlphaGenome predictions with other measures of variant effect, such as conservation-based scores, as well as existing data on gene functions and pathways could prove useful in advancing common and rare-variant analysis. Finally, estimates of model certainty would be useful for better interpreting predictions.”

For the discussion, what are some current limitations in terms of existing data that should be the focus of future large scale experimental studies? Are there modalities (e.g. chromatin accessibility) where data is not the major bottleneck in modelling? Given the current methodologies, are there specific data-types and modalities that could result in strong advances given additional datasets?

We think it is useful to distinguish between two types of data bottleneck:

- Lack of output diversity: certain combinations of modality and cellular context are missing and can therefore not be trained on.
- Lack of input diversity: all our outputs are measurements of the same underlying reference genomes.

We agree that, for certain modalities such as chromatin accessibility and RNA-seq, a large amount of high-quality data is available for a large number of cellular contexts. Here, the main bottleneck likely is the lack of the underlying genome diversity, as argued by Boer and Taipale (<https://pubmed.ncbi.nlm.nih.gov/38093018/>). Certain regulatory elements - distal enhancers possibly among them - may simply occur too infrequently in the human and mouse genomes for a model to accurately learn the mechanisms that determine their function and their impact on gene expression from observational data alone.

We see two data generation efforts that could help to increase input genome diversity:

- Assaying additional species: generating ENCODE-like compendia for additional vertebrate species could allow models to learn from both conservation and divergence between species.
- Perturbational data targeting non-coding regulatory elements: provide direct, causal evidence of a variant's effect on gene expression and are invaluable for rigorously validating and training the next generation of variant effect prediction models.

For other modalities, such as CHIP-TF or 3D contact maps, there is currently a severe lack of cellular diversity, with most of the data coming from a limited number of cell lines. Generating TF binding or contact map data for a larger number of cell lines may allow the model to better grasp cell-type specific regulatory mechanisms, as well as provide useful additional outputs when interpreting variant effects.

For contact maps in particular, efforts to generate higher resolution data may prove useful for better resolving the precise connections between distal enhancers and their target genes (the contact maps in this paper were at 2048bp resolution). Having said that, we did see that there were diminishing returns from adding modalities beyond RNA-seq and DNA accessibility (see **Supplementary Figure 11**, rightmost column). However, it's unclear whether this is because most of the remaining measurements are less helpful or whether because they are mostly from experiments done in major cell lines.

We have added a sentence to our Discussion reflecting these points:

“Addressing these limitations motivates several future research directions. Data generation efforts could increase input genome diversity by assaying additional species, or by perturbing non-coding regulatory elements at scale to help build the next generation of variant effect prediction models.”

Referee #2 (Remarks on code availability):

The code and API are not yet available to review.

We are happy to report that the API is now live and free to use at <https://deepmind.google.com/science/alphagenome/>, and the model weights and open source code will be available at the time of formal publication.

Referee #3:

The authors present AlphaGenome, a unified DNA sequence-to-function deep learning model that predicts diverse modalities, including gene expression, transcription initiation, and others. Taking 1 megabase of DNA sequence as input and being trained on the human and mouse genomes, AlphaGenome matches or exceeds current state-of-the-art models on genome track and variant effect prediction.

Employing a U-Net-inspired architecture and a two-stage training process (pre-training and distillation), AlphaGenome allows unprecedentedly large input sequences while retaining single-base-pair resolution and simultaneously predicting 5,930 human or 1,128 mouse genome tracks across 11 modalities. The work is clearly a very impressive engineering effort, and the manuscript is extremely clearly written. Of particular note is how complete the methods

descriptions are, both in terms of the modeling details and the processing of each data type, and how open the authors are about giving credit to others. On the other hand, there is not much innovation: The model itself and the training loss is very similar to Borzoi (as the authors mention) with an additional loss component from Decima. The work largely involves demonstrating that their unified model outperforms (almost) all of the individual SOTA methods. The use of the model for discovering something new is limited, and no experiments were performed to validate any novel hypotheses. Thus, the advance on novel insights or any individual prediction task appear more incremental, and it is largely the scope and comprehensiveness of AlphaGenome that stands out.

Given the rather incremental improvement of the individual prediction tasks, AlphaGenome will likely not be the AlphaFold for genomics. However, I appreciate very much the amount of engineering effort the authors put in at every stage, and that they credit everyone with where the ideas came from. Moreover, if the authors make the full code accessible to the compbio community and build a powerful and user-friendly interface for the general scientific and medical community for people to query their own variants, AlphaGenome will have the strong impact that warrants a high-profile publication.

Major Comments

The distillation procedure seems conceptually interesting, but it is unclear what the benefit of including the variants (point mutations and structural variants) was. Specifically, using a teacher model to make predictions for these models does not add new information for the student model compared with training on the original data. Rather, the main benefit seems to be that of classic distillation: faster inference than a larger (ensemble) of models at a small cost to performance. Therefore raising the question whether training using the variants is actually helpful?

Reviewer 2 asked a similar question: we share the answer again here:

As a demonstration of the importance of the sequence augmentation, we have performed an ablation study where we repeated the distillation procedure without the input sequence perturbation (point mutations and indels). The resulting student model distilled in the absence of sequence perturbations showed a drop in performance relative to the student model reported in the paper (which has a replacement rate of 4%).

We have added the following statement to the manuscript to reflect these results:

“...Interestingly, distillation of even just a single teacher model was beneficial for some variant effect prediction tasks such as caQTLs, splicing outliers or eQTL sign prediction. Finally, distillation without randomly mutating the input sequence resulted in a drop in student model performance (eQTL sign: -0.06, eQTL causality: - 0.01, sQTL causality: -0.01, splicing outlier - 0.015), affirming the impact of our input perturbation strategy during distillation. ...”

The performance is partly evaluated on absolute terms, yet partly only on relative terms, compared to existing models. Fig. 1 for example shows relative improvements, which are informative from a technical point of view (capturing the improvements over previous models) but don't communicate the performance gap to a perfect predictor. Especially for prediction tasks for which the state-of-the-art models perform more poorly (see e.g. Tang, 2023 <https://pubmed.ncbi.nlm.nih.gov/38036789/>; Sasse, 2023 <https://pubmed.ncbi.nlm.nih.gov/38036778/>), this additional information seems important. Please add more readily interpretable performance metrics.

We have now revised **Figure 1d** and **e** to include the absolute performance values for AlphaGenome. This provides the necessary context for the relative improvement percentages and allows readers to more clearly see the raw performance on each task.

Furthermore, to improve interpretability as suggested, we have also updated the figure captions to make the scale and perfect performance value for each metric (e.g., a correlation of 1.0) more explicit.

This is captured by the following caption additions:

“(d) Relative performance improvement (%) of AlphaGenome over the best competing model for a selection of genome track prediction tasks across different modalities and resolutions (Supplementary Table 3). The 'Value' column represents AlphaGenome's absolute performance. For all tasks shown, a value of 1.0 indicates perfect performance, with the exception of 'Profile JSD', for which the ideal value is 0. Both competing models and AlphaGenome pre-trained, fold split models were evaluated on held-out genome regions unseen during model training. For classification tasks, we adjust the relative improvement to account for the performance of a random classifier (Methods). PA: polyadenylation. JSD: Jensen-Shannon divergence. (e) Relative performance improvement of AlphaGenome over the best competing model for a subset of variant effect prediction tasks (Supplementary Table 4). The distilled student AlphaGenome model is used for these evaluations. The ds/caQTL direction (causality) rows represent the average relative improvement across multiple similar datasets (Methods).”

Additional comments

I don't understand why quantile normalization and mean-centering the expression values (Fig. 2d, Methods p. 45) brings the PCCs within each track down from 0.82 to 0.52 (compare left to middle). Which feature of the data is distorted to cause this? What does it mean that the models are better in predicting raw log-transformed expression values than their quantile-normalized and/or mean-centered counterparts?

This disparity indicates that while the model is effective at capturing the average expression levels of genes across tissues, it is less proficient at predicting tissue-specific deviations in expression.

In the first evaluation, variance in the raw log-transformed data is driven by two primary components: (1) the mean, tissue-averaged expression of a gene and (2) its tissue-specific variation. Because of the presence of housekeeping genes (highly expressed in all tissues) and lowly expressed genes, the first component dominates the second, and therefore the high 0.82 PCC on raw data largely reflects the model's success in predicting the first component – the strong, global signal distinguishing highly- versus lowly-expressed genes.

In the normalized evaluation, mean-centering the data removes this dominant, tissue-averaged component, leaving only tissue-specific variation. This is a much more challenging task, resulting in the lower 0.52 PCC.

What is the prominent peak in RNA-seq coverage after the APOL4 gene (blue) that is decreased for the ALT allele? Is this part of APOL4, e.g. due to alternative polyadenylation, or a separate downstream gene?

In the originally submitted figure, the RNA-seq peak was misaligned due to an error during the final figure formatting. We have corrected this in the revised **Figure 4b**. The prominent RNA-seq peak is now correctly aligned with an exon of the *APOL4* gene itself and is not a separate downstream feature. We are grateful to the reviewer for helping us improve the clarity and accuracy of the figure.

When assessing magnitude of eQTL effects, why is the Spearman CC used rather than the PCC? The agreement between predicted and observed effect sizes looks rather poor (Fig. 4d); please explain and/or comment on this in the main text.

We initially used Spearman correlation for eQTL effect size to follow the established convention from prior work (Enformer, Borzoi), which focuses on correctly ranking the relative effects of variants. We thank the reviewer for this comment, and have now computed and added the Pearson correlation to the **Figure 4d** caption to assess the direct linear agreement. We note for the reviewer that while Pearson R (0.39) is, as expected, lower than the Spearman R (0.5), the improvement over Borzoi was still substantial (0.39 compared to 0.33), irrespective of the metric used.

“d) Observed versus AlphaGenome-predicted eQTL effect sizes. Scatterplot comparing observed SuSiE ‘beta posterior’ values to AlphaGenome (distilled) predicted effect sizes for 17,675 fine-mapped GTEx eQTLs (SNVs). Each point is a unique variant/gene/tissue combination. Spearman’s ρ (signed) = 0.50; Spearman’s ρ (unsigned, absolute values) = 0.10. Pearson’s r (signed) = 0.39; Pearson’s r (unsigned, absolute values) = 0.20.”

We agree that the overall correlation in the scatterplot (**Fig. 4d**) appears modest. This reflects a known challenge in the field: predicting the precise quantitative magnitude of a variant's effect from sequence alone is a very difficult problem. We find this is particularly true for distal eQTLs, whose effects models often underpredict (see **Extended Data Figure 5f**):

f Effect of eQTL-TSS distance

Notably, we have shown that if we restrict to variants that have high AlphaGenome scores (e.g. in the 99th percentile), the Spearman correlation is much higher (0.73). This is shown in **ED Fig. 5f** and **ED Fig. 5h**:

Extended Data Figure 5f

Extended Data Figure 5h

h eQTLs with predicted scores larger than 99% of common variants

On p. 11 (**Fig. 4g,h**), the evaluation of the direction of effect seems unnecessarily complicated. Given the binary nature of the data, up vs. down, and the baseline of 50% for balanced

datasets, what is the fraction of correctly predicted effects? In Fig. 4h, the legend mentions “with a predicted direction of effect”, not indicating if the predicted direction is correct. Please clarify and simplify the analysis and presentation.

On Fig. 4g:

We acknowledge the confusion around the fraction of correctly predicted effects. This is exactly what is shown in the x-axis of **Fig 4g**. Accordingly, we have replaced the x axis label “target metric value” to “Fraction with correct sign” for clarity:

g Fraction correct sign

On Fig. 4h:

We have also changed the wording in the caption to describe the panel with greater clarity around how we defined a credible set as having a resolved sign prediction:

*“(h) GWAS credible set direction of effect (Sign) prediction coverage. Fraction of Open Targets [31] GWAS loci with a predicted direction of effect for a plausible target gene, comparing AlphaGenome predictions to eQTL colocalization approach. Top: Each bar segment represents a different strategy for summarizing AlphaGenome scores (PIP-weighted, PIP-max, and Any variant) and two different score thresholds that yielded a given sign accuracy on eQTLs (80% and 90%, see **Methods**). For COLOC, we counted a credible set as resolved if the $COLOC H4 > 0.95$ [18]. Bar segments indicate loci resolved by: COLOC only (blue), AlphaGenome only (red), both methods (purple), or neither (grey). Bottom: Using the AlphaGenome strategy of PIP-weighted (80%), credible sets were further stratified by different properties (see **Methods**).”*

On p. 15, the authors might consider revising the introduction to MPRA data, removing the reference to “local chromatin accessibility”, which is not tested by plasmid-based MPRA as used in ref. 34.

We thank the reviewer for this helpful suggestion. We have revised the introduction to MPRA data, which now reads as follows:

“... Beyond population-level QTLs, understanding the impact of local sequence context on gene regulation is important. We assessed AlphaGenome chromatin features in the CAGI5 saturation mutagenesis MPRA challenge, which aims to predict the target gene expression driven by short DNA sequences reflecting promoter or enhancer elements. We evaluated this benchmark using DNase, RNA-seq, and ChIP output types and compared to Enformer, Borzoi, and ChromBPNet predictions for the same sequences.”

It is unclear when the authors are referring to the distilled version or some of the other models. Based on the initial description, I assume that sections after L234 would be from the distilled model but some of the figure captions refer explicitly to a distilled model and others do not, e.g. Fig. 4 and specifically Fig. 4j. Please clarify.

Thank you for this helpful feedback. The reviewer’s assumption about the usage of the distilled model from L234 onwards is correct. The error is in our inconsistent labeling: for the captions in **Figure 4**, we repeatedly emphasize that the model is “(distilled)”, which is inconsistent with the remainder of the manuscript. To address the confusion, we have done the following:

- Added additional model descriptions to **Figure 1** captions:

Figure 1 captions:

“(d) Relative performance improvement (%) of AlphaGenome over the best competing model for a selection of genome track prediction tasks across different modalities and resolutions (Supplementary Table 3). The 'Value' column represents AlphaGenome's absolute performance. For all tasks shown, a value of 1.0 indicates perfect performance, with the exception of JSD, for which the ideal value is 0. Both competing models and AlphaGenome pre-trained, fold split models were evaluated on held-out genome regions unseen during model training. For classification tasks, we adjust the relative improvement to account for the performance of a random classifier (Methods). PA: polyadenylation. JSD: Jensen-Shannon divergence (e) Relative performance improvement of AlphaGenome over the best competing model for a subset of variant effect prediction tasks (Supplementary Table 4). The distilled student AlphaGenome model is used for these evaluations. The ds/caQTL direction (causality) rows represent the average relative improvement across multiple similar datasets (Methods)”

- Removed the repeated emphasis of “AlphaGenome (distilled)” and changed this to simply “AlphaGenome” in both the **Figure 4** model labels and the figure captions:

Figure 4 captions:

“(c) eQTL effect size ('Coefficient') prediction performance. Bar plot comparing tissue-weighted mean Spearman's ρ for AlphaGenome (~~distilled~~) against Borzoi (ensemble) and Enformer for

predicting eQTL effect magnitudes for single nucleotide variants (SNVs; 'Coefficient') and insertions/deletions ('Coefficient (indels)') across 49 GTEx tissues.

(d) Observed versus AlphaGenome-predicted eQTL effect sizes. Scatterplot comparing observed SuSiE 'beta posterior' values to AlphaGenome-~~(distilled)~~ predicted effect sizes for 17,675 fine-mapped GTEx eQTLs (SNVs). Each point is a unique variant/gene/tissue combination). Spearman's ρ (signed) = 0.50; Spearman's ρ (unsigned, absolute values) = 0.10.

(e) eQTL direction of effect ('Sign') prediction performance. Bar plots comparing tissue-weighted mean Area Under the ROC Curve (auROC) for AlphaGenome-~~(distilled)~~ against Borzoi (ensemble) and Enformer for SNVs and indels.

(f) eQTL Sign prediction performance stratified by variant-TSS distance for SNVs. Bar plots show tissue-weighted mean auROC per model across distance bins.

(g) Sign prediction accuracy versus eQTL recovery. Plot relating target Sign accuracy (achieved by applying different AlphaGenome or Borzoi variant score thresholds) to the fraction of GTEx eQTLs recovered for AlphaGenome-~~(distilled)~~ versus Borzoi (ensemble). Dashed lines indicate eQTL recovery rates at score thresholds yielding 90% Sign accuracy for each model.

(h) GWAS credible set direction of effect (Sign) prediction coverage. Fraction of GWAS loci (from Open Targets31) with a predicted direction of effect for a plausible target gene, comparing AlphaGenome-~~(distilled)~~ predictions at different accuracy thresholds (80% or 90%) against an eQTL colocalization approach (COLOC $H4 > 0.95$ [18]), stratified by variant/locus properties (Methods). Bar segments indicate loci resolved by: COLOC only (blue), AlphaGenome only (red), both methods (purple), or neither (grey).

(i) eQTL causality prediction ('Causality', tissue-weighted mean auROC). The bar plot compares zero-shot performance of AlphaGenome-~~(distilled)~~ against baselines and supervised performance using a Random Forest on model variant scores ('Causality (RF)').

(j) Enhancer-gene linking performance (ENCODE-rE2G CRISPRi dataset11). Zero-shot evaluation: Performance (auPRC) comparison stratified by enhancer-TSS distance for AlphaGenome-~~(distilled)~~ vs Borzoi vs TSS distance baseline. Supervised evaluation: AlphaGenome input gradient score integrated into ENCODE-rE2G-extended vs ENCODE-rE2G models."

Figure 4 panels:

- Figure 4f - ~~(distilled)~~
- Figure 4j ~~(distilled)~~
- Figure 4k ~~(distilled)~~

It was somewhat unclear why the model made predictions for TF binding at 128-bp resolution, when TFs have been shown by the first author in a previous paper to have important basepair resolution properties. The ablation section mentions that increasing resolution is helpful, and predictions are made for other assays at basepair resolution. Potentially, there are preprocessing issues related to fragment length and single-end vs paired-end differences that may be challenging to overcome, but it would be worth mentioning explicitly why this choice was made.

In our past work, we have indeed observed that modeling TF binding events using ChIP-nexus (or ChIP-exo) data at base resolution is important. In contrast to ChIP-nexus data, ChIP-seq data are inherently much noisier in terms of spatial resolution, with broad peaks spanning 100-300 bp. However, given that we were already including ATAC and DNase-seq data at base resolution, we anticipated that the benefits of modeling TF ChIP-seq data at base-resolution would be relatively small compared to the substantial effort of ingesting and pre-processing the data. Additionally, ChIP-exo and ChIP-nexus data are relatively rare in ENCODE. We have therefore left including higher-resolution TF binding data as an extension for future work.

We have now added this rationale to the manuscript in the Methods section describing ChIP-Seq data acquisition:

“...Following these steps, selected ChIP-seq experiments were grouped by their ontology CURIE, strand, and specific ChIP-seq target (TF or histone modification, Supplementary Table 2). Within each group, the 'fold change over control' signal was summed. This signal was then binned to a 128-bp resolution since the signal ENCODE-provided ChIP-seq bigWig files were already smoothed and provided at coarser ~100bp resolution. These resulting ChIP-seq tracks were used without further normalization, as fold-change values inherently account for control signals. We note that while higher resolution ChIP-based assays such as ChIP-exo and ChIP-nexus yield reliable basepair resolution signal, these tracks are rare and the vast majority of ENCODE chromatin immunoprecipitation was done using ChIP-seq, which is not inherently base-resolution. Given the substantial data processing effort that would be required to curate non-ENCODE base pair resolution ChIP data, we deferred base-resolution modeling for TF binding to future work.”

Referee #4:

I co-reviewed this manuscript with one of the reviewers who provided the listed reports.

We thank Referee #4 for their contribution to the review process.

Additional changes

In addition to these reviewer-requested comments, we have also made the following manuscript improvements:

(1) Test time augmentation (TTA) for aggregated track evaluations

Previously, we had not implemented test-time augmentation for aggregated track evaluations (such as aggregations over the gene body, the polyA-centric coverage ratios, etc). We have updated the following evaluations:

- Gene-centric RNA-seq evals for human and mouse

- COVR evaluation (additionally, see below for an update on the PolyADB processing step)

The following paper elements are now updated as a result:

- **Fig. 2d**
- **Extended Data Fig. 6a**
- **Fig. 1d**
- **Supplementary Table 3**
- **Extended Data Fig. 3c**

The relative rankings between AlphaGenome and Borzoi remain unchanged after the TTA updates.

Accordingly, we have updated the **Methods** (“Test-time augmentation” section) as follows:

“A test-time augmentation (TTA) strategy involving strand averaging was employed when evaluating performance on variant, unaggregated, and aggregated track evaluations.”

(2) COVR evaluation

In the COVR evaluation, we updated the set of sites used in the coverage ratios evaluation to include only gene and polyadenylation site pairs annotated by PolyADB and **within 50bp of the gene boundary**.

The results remain similar to what was reported previously. We have updated **Extended Data Fig. 6a**. The main text has been updated to read:

“We found that AlphaGenome achieves SOTA performance in predicting APA (Spearman $\rho = 0.868$ vs. Borzoi’s 0.767 ; Extended Data Fig. 6a; Methods)”.

We have updated the **Methods** (“Alternative Polyadenylation” section) to reflect this:

“... These COVR values quantify the relative usage of the distal versus proximal polyadenylation sites for a gene as annotated by PolyADB, restricted to sites no further than 50bp from the gene boundary.”

(3) eQTL causality evaluation

Upon investigation of our implementation of the eQTL Causality task, we found that our approach of “double rebalancing” (first for class, then for distance) unintentionally skewed the dataset heavily towards distal variants. This distal skew made the task artificially challenging and thereby compressed model performance to a narrow range.

To fix this, we regenerated the benchmark from the eQTL Catalogue (using the thresholds $PIP \geq 0.9$ for causal variants and ≤ 0.01 for noncausal variants) using a single, simultaneous

balancing for both class and distance. This new dataset preserves the native distance distribution of causal variants. On this corrected, more representative evaluation, the model rankings are now consistent with our previous eQTL coefficient/sign metrics:

Category	AlphaGenome Before	AlphaGenome After	Borzoi Before	Borzoi After	Relative Improvements Before	Relative Improvements After
eQTL causality (zero shot)	0.683	0.713	0.685	0.702	-1.08	5.41
eQTL causality (supervised)	0.745	0.800	0.713	0.759	15.02	15.64

We have updated all relevant methods, figures, and results as follows:

Abstract and results - AlphaGenome now outperforms leading models on 25 out of 26 variant effect evaluations.

Figure 1e - updated barplot numbers.

Supplemental Table 4 - updated variant scoring numbers.

Figure 4i - updated panel.

Extended Data Fig. 5a, b, c, e - updated figures.

Figure 7 / Supplementary Figure 11 - updated numbers for eQTL causality benchmark for the ablation studies.

Methods revisions:

“...1. eQTL Dataset Preparation. For SNP eQTL effect size and sign evaluations, we used a dataset of eQTLs from GTEx v823, fine-mapped using the SuSiE method³³. For these evaluations, only variants with a posterior inclusion probability (PIP) ≥ 0.9 were retained. Variants with a posterior inclusion probability (PIP) ≥ 0.9 are labeled as causal, while those with $PIP < 0.01$ are deemed noncausal. For all reported metrics and counts of eQTLs we used variants only in the ‘test’ set chromosomes (see ‘Chromosome splits for variant benchmarks’ above), which correspond to approximately one third of the fine-mapped variants.”

We have also updated this section to reflect the new dataset construction approach:

“5. Zero-Shot eQTL Causality Prediction. We evaluated the model’s zero-shot ability to differentiate causal from non-causal eQTLs. For this purpose, we downloaded the reprocessed and SuSiE fine-mapped GTEx data provided via the EMBL-EBI eQTL catalogue. We removed all non-SNV variants. We then designated variant-gene pairs with $PIP \geq 0.9$ as putatively causal, whereas variant-gene pairs with $PIP \leq 0.01$ were designated as non-causal. We excluded variant-gene pairs where the eGene was not annotated in the GTF used throughout our analyses (Gencode V46).

To account for the inherent distance bias of this data (most causal eQTL are close to their target genes, whereas negatives show a more uniform distribution), we performed a distance balancing. Specifically, we computed the log of the absolute distance (bp) between the eQTL variant and the TSS of its target gene (using the median TSS across all transcripts of that gene in the GENCODE V46 GTF). We then divided the range of distances into 10 equal-width bins, and within each bin, randomly downsampled the variant/gene/tissue tuples belonging to the non-causal class to match the number in the causal class. This resulted in 21,948 variant/gene/tissue tuples in the test set, with a class balance of exactly 50%. We measured model performance at distinguishing causal from non-causal eQTLs using the auROC metric, averaged across GTEx tissues weighted by the fraction of eQTLs in each tissue. We also evaluated distance-to-TSS as a predictor using the same metric, and confirmed that it was no longer predictive in the distance-balanced causality dataset.”

(4) Additional ablations

(i) Tissue specific gene expression bug

After all model training runs had finished, we identified an implementation error affecting the RNA-seq output head’s tissue-specific gene expression loss (detailed in the 'Output Heads' section of the **Methods**). This error resulted in incorrectly flipped gene boundary strands for half of the training intervals. We have since verified that, due to the low relative weight and magnitude of this auxiliary loss term, the error’s impact on final model performance was within the range of expected noise, as measured by the Pearson correlation of log-fold change in gene expression.

We have added the following sentences to the **Methods** (“Output Heads” section) in order to document that this bug was present during training:

“... The Multinomial term within this calculation is weighted by a factor of 5.0. This auxiliary gene-level loss contributes to the model’s total training loss with an overall weight of 0.1.”

After all model training runs had finished, we identified an implementation error affecting the RNA-seq output head’s tissue-specific gene expression loss. This error resulted in incorrectly flipped gene boundary strands for half of the training intervals. We have since verified that, due to the low relative weight and magnitude of this auxiliary loss term, the error’s impact on final

model performance was within the range of expected noise, as measured by the Pearson correlation of log-fold change in gene expression.

(ii) Interval sampling in ablations

We have improved the way that we sampled and resized intervals in the ablation study for **Fig. 7b**, and have made the following addition to the **Methods** “Model Ablations” section:

“Matched Training and Evaluation Length (Green Series): Models were trained and evaluated using the same matched input sequence length. In all evaluation scenarios, we uniformly tile the holdout set by the model's target interval (the specific sub-sequence of model predictions that we evaluate) to ensure a consistent and unbiased comparison across different sequence lengths.”

We have also added the following clarifying text to the “Model Ablations” Methods section:

“To ensure a fair comparison across all model configurations (especially those that vary the input sequence length), we implemented a specialized evaluation protocol for all ablation results. While in our primary analyses use fixed target intervals defined by Borzoi (see Dataset splitting and cross-validation), here we uniformly tiled the validation set genomic regions with non-overlapping target intervals. For track prediction, the size of these intervals was dynamically set to 3/16 of the input sequence length. This formula maintains the same proportional evaluation window across all tested sequence lengths and corresponds to the standard 196kb for 1 Mb input. For contact map evaluations we tiled the validation by the full input sequence length. Contact map evaluations were also performed at AlphaGenome's native 2048 bp resolution, without interpolation, and averaged over all 28 human datasets.”

We have also slightly reworded the “Impact of Target Resolution” section in the **Methods** to better explain how the ablations work, and clarify how the natively lower-resolution tracks are handled:

“In this study, we systematically varied the output resolution for our natively high-resolution targets, including gene expression (RNA-Seq, CAGE-seq, PRO-cap), DNA accessibility (ATAC-seq, DNase-seq), and splicing. Separate models were trained where these tracks were predicted at 1 bp, 2 bp, 8 bp, 32 bp, and 128 bp. The model's inherently lower-resolution targets – ChIP-seq and contact maps – were held constant and trained at their native resolutions (128 bp and 2048 bp, respectively) in all experiments. This design allowed us to isolate the effect of changing the resolution of one set of tasks on the performance across the entire model.”

For these ablations, the core model architecture, including the decoder structure up to the point of embedding extraction, remained unchanged. The decoder embeddings are extracted at the resolution of the target tracks, and used to compute the output embeddings with the output embedder module described in Output Heads. The specific methods used to downsample the high-resolution ground truth data were as follows:

- For continuous tracks (RNA-Seq, ATAC-seq, etc.), 1 bp resolution counts were summed into consecutive bins after augmentation was applied at base resolution.
- For per-base classification (splice sites), 1 bp resolution ground truth labels (positive classes) were max-pooled to the target resolution.
- *For splice junction count prediction, which relies on base-resolution positions of donor and acceptor sites, lower resolution embeddings from the Output Embedder were upsampled by repeating along the sequence axis (e.g., 4 times for 4 bp embedding), and used by the junction prediction head as if at base-resolution.”*

(5) Miscellaneous typos and corrections

We corrected a typo in the units of ChIP-seq read coverage in the **Methods**:

“Following these steps, selected ChIP-seq experiments were grouped by their ontology CURIE, strand, and specific ChIP-seq target (TF or histone modification, Supplementary Table 2). Within each group, the ‘fold change over control’ signal was ~~averaged~~ summed.”

We updated the legend in **Extended Data Fig. 5A** with new colors for slight visibility improvements.

AlphaGenome Before and After Revisions Figure Comparisons

While these figure changes are detailed in our point-by-point responses, this document provides a consolidated, side-by-side comparison of the original and revised figures for your convenience.

Figure 1 - Barplot update

Changes:

- A "Value" column has been added to display AlphaGenome's absolute metric value, making it easier to assess both absolute and relative performance of the model.
- In **Figure 1d**, the following entries have been updated:
 - RNA-seq coverage
 - RNA-seq gene expr. LFC Pearson r
 - Alternative PA
 - DNase-seq 1 bp
 - ATAC-seq 1bp
- In **Figure 1d**, we have switched to using Profile JSD as the metric for DNase-seq and ATAC-seq tracks as this is more appropriate for the tasks and is at base-resolution.
- In **Figure 1e**, the eQTL causality value has been updated.

Figure 1d and Figure 1e Before Revision:

Figure 1d and Figure 1e After Revision:

d Track prediction: Pre-trained model, fold split

e Variant effect prediction: Distilled model, all-folds

Figure 2 - Gene-level RNA-seq update

Changes:

- We have updated the RNA-seq gene-level performance numbers, which are now slightly higher due to the usage of TTA (test-time augmentation).

Figure 2d Before Revision:

Figure 2d After Revision:

Figure 4 - eQTL updates

Changes:

- In **Figure 4b**, we corrected the placement of the *APOL4* transcript.
- In **Figure 4i**, we corrected a flaw in our eQTL causality dataset and now report updated results.
- In **Figure 4d**, we added a Pearson r value to the scatter plot.

Figure 4b Before Revision:

b Predictions for a known eQTL

Figure 4b After Revision:

b Predictions for a known eQTL

Figure 4i Before Revision:

i eQTL causality prediction

Figure 4i After Revision:

i eQTL causality prediction

Figure 4d Before Revision:

d Predicted versus observed

Figure 4d After Revision:

d Predicted versus observed

Figure 6 - Adding label to multimodal variant analysis figure

Changes:

- In **Figure 6f**, we have added the word 'Enrichment' to improve the clarity of the figure.

Figure 6f Before Revision:

f Multimodality in trait-altering non-coding variants

Figure 6f After Revision:

f Multimodality in trait-altering non-coding variants

Figure 7 - Ablations update

Changes:

- The “eQTL Causality auROC” row has been updated (other values may shift slightly due to the experiment being rerun but are within noise).

Figure 7 Before Revision:

Figure 7 After Revision:

Extended Data Figure 3C - Updated due to TTA

Extended Data Figure 3C Before Revision:

c Borzoi at base-pair resolution

Extended Data Figure 3C After Revision:

c Borzoi at base-pair resolution

d Borzoi at 32 bp resolution

Extended Data Figure 5: eQTL updates

Changes:

- In **Figure ED 5a**, we have improved the legend for readability and updated the two scatter plots concerning eQTL causality.
- In **Figure ED 5b**, we have updated the “eQTL causality” panel.
- In **Figure ED 5c**, we have updated the “eQTL causality” panel.

Extended Data Figure 5A Before Revision:

a eQTLs stratified by tissue

Extended Data Figure 5A After Revision:

Extended Data Figure 5B Before Revision:

Extended Data Figure 5B After Revision:

b eQTLs stratified by distance

Extended Data Figure 5C Before Revision:

c eQTLs stratified by functional class

Extended Data Figure 5C After Revision:

Extended Data Figure 5E Before Revision:

Extended Data Figure 5E After Revision:

e Impact of individual variant scorers on eQTL causality RF performance

Extended Data Figure 6: Polyadenylation update

Changes:

- For ED 6A, new polyadenylation coverage ratio scatter plots were generated.

Extended Data Figure 6A Before Revision:

a Alternative polyadenylation prediction

Figure ED 6A After Revision:

a Alternative polyadenylation prediction

Extended Data Figure 9: PhyloP variant addition

Changes:

- Added a new panel i to accompany the new PhyloP analysis.

Supplemental Figure for TAL1 - added as SI Figure 10

Supplementary Figure 10 | **Tissue specificity of *TAL1* upregulating oncogenic variants.** Each data point is a human RNA-seq track. For each track, the group difference between predicted *TAL1* expression for oncogenic and shuffled background variants defined in Fig. 6 was assessed using a two-sided Mann-Whitney U statistic. The plot is split across four biosample types: *in vitro* differentiated cells, primary cells, GTEx tissues, and cell lines. Within each group, each sample is ranked in the x-axis by the magnitude of the Mann-Whitney U statistic (on the y-axis). A single labeled red data point highlights the sample that most closely matches the T-ALL tissue of origin within that biosample type.

Figure SI 11: Modality ablations update

Changes:

- Added third new column to show the results of the new cumulative ablation experiments.
- The “eQTL Causality auROC” row has been updated.

Figure S11 Before Revision:

Figure S11 After Revision (now called Figure S12):

Point-by-Point Response to Nature Reviewers' Comments

AlphaGenome: advancing regulatory variant effect prediction with a unified DNA sequence model

Žiga Avsec^{1*} ✉, Natasha Latysheva^{1*}, Jun Cheng^{1*}, Guido Novati^{1*}, Kyle R. Taylor^{1*}, Tom Ward^{1*}, Clare Bycroft^{1*}, Lauren Nicolaisen^{1*}, Eirini Arvaniti^{1*}, Joshua Pan^{1*}, Raina Thomas¹, Vincent Dutordoir¹, Matteo Perino¹, Soham De¹, Alexander Karollus¹, Adam Gayoso¹, Toby Sargeant¹, Anne Mottram¹, Lai Hong Wong¹, Pavol Drotár¹, Adam Kosiorek¹, Andrew Senior¹, Richard Tanburn¹, Taylor Applebaum¹, Souradeep Basu¹, Demis Hassabis¹, Pushmeet Kohli¹ ✉

Overview

We are delighted to learn that our manuscript has been accepted in principle for publication. We are grateful for the constructive feedback provided by the reviewers throughout this process, which has significantly strengthened the clarity and robustness of our work. We believe the revised manuscript now presents a rigorous resource for the genomics community.

Our detailed point-by-point responses follow.

In places where we have changed the text, we have included these changes directly in the response document using the following format:

“...Existing, unchanged manuscript text in italics. Newly introduced text to address the reviewer concerns, in underline. Additional existing, unchanged manuscript text in italics.”

Referees' comments:

Referee #1:

The authors have satisfied all my concerns. My only minor comment would be regarding Fig. 6f - this is still quite hard to read, partially due to the flipped axes (dependent variable on the x rather than y) and partially because it's hard to cross reference exactly what the enrichment corresponds to - is it the red? blue? orange? In addition, a recent preprint from Finucane et al. describes a system for assessing variant effect predictors, and that framework might be useful here to show the impact on human genetics applications.

We thank the reviewer for pointing out the readability issue with Fig. 6f. We have explored ways to improve the figure and its description. Firstly, we have amended the caption for further clarity, specifically to improve the understanding of the relationship between the categories and the enrichment score:

(f) Fraction of trait-affecting variants⁴² (“candidate Causal”, 338 for mendelian and 1140 for complex traits), as well as matched control variants⁴² (“Control”, 3042 and 10260, respectively), which exceed varying quantile-score thresholds in at least one predicted track. Here, surpassing a quantile-score threshold of e.g. 1.0 implies a predicted effect in excess of 99% of common variants (Methods). Variants are categorized depending on the tracks where the threshold was passed: ‘Local Regulation’ (ChIP/DNase/ATAC), ‘Expression only’ (RNA/CAGE) and ‘Multimodal’ (combination of the above). Numbers in the center indicate the relative enrichment of detected variants among candidate causal variants compared to the control variants. Here, we define a variant as ‘detected’ if there is at least one track for which AlphaGenome’s variant effect score exceeds the stated percentile (e.g. 1.0 means the ‘quantile score’ > 0.99; see Methods). Numbers above the bars indicate the relative enrichment of detected variants (sum of the three categories) among candidate causal variants compared to the control variants. The enrichment increases with increasingly stricter thresholds, with a reduction in recall (x-axis). For instance, causal variants for Mendelian traits are 10.7-fold more likely to have an AG score in the top 0.005% compared to control variants.

Additionally, we explored transposing the axes so that the dependent variable is now plotted on the y-axis for more intuitive interpretation. The resulting figure is this:

We agree with the reviewer that this makes visual comparisons easier and enhances the readability of the figure, and have therefore replaced the version in the manuscript with this new arrangement.

We also thank the reviewer for directing us to the recent work by Finucane *et al.* (2024). We agree that distinguishing between variant-to-gene (molecular impact) and gene-to-disease (gene essentiality/relevance) components is critical for evaluating variant scoring methods.

AlphaGenome is designed to excel at the 'variant-to-gene' component by predicting molecular phenotypes (expression, splicing, chromatin state) directly from sequence with high resolution. We acknowledge that for clinical applications, these molecular predictions must often be integrated with 'gene-to-disease' priors (such as gene constraints or disease associations) to achieve maximal performance on pathogenic classification tasks. We have added a snippet to our Discussion to explicitly cite Finucane *et al.* and call out this distinction:

... Finally, application to complex trait analysis is limited given that AlphaGenome predicts molecular consequences of variants, while these phenotypes involve broader biological processes (including gene function, development, and environmental factors) and gene-to-disease effects (Finucane et al. 2024) beyond the model's direct sequence-to-function scope.

Referee #2:

The authors have address my previous concerns. I appreciate as well several of the responses to the other reviewers and the effort to correct issues that were found since submission. I also recognize the effort to make their work publicly accessible and reusable.

We thank the reviewer for their positive feedback and for their constructive engagement throughout the review process. We are glad that our efforts to address the previous concerns and to ensure the accessibility and reusability of our work have been satisfactory.

Referee #3:

The authors have provided substantial revisions and additions to their original manuscript to address the reviewers' concerns and suggestions. In addition, they have communicated to the editor that their code and an API are now available, and that the model source code and weights will be provided upon publication. The revision and the availability of code, API and weights satisfy my original concerns, and I congratulate the authors to their comprehensive work.

We thank the reviewer for their kind words and congratulations. We appreciate their recognition of our efforts to make the AlphaGenome code, API, and weights available to the community, and we are pleased that the revisions have satisfied their original concerns.

Referee #4:

I co-reviewed this manuscript with one of the reviewers who provided the listed reports.

We thank the reviewer for their time and contribution to the assessment of our manuscript.